# 🍅 TOMATO: Assessing Visual Temporal Reasoning Capabilities in Multimodal Foundation Models

**Ziyao Shangguan**[*1]   **Chuhan Li**[*1]   **Yuxuan Ding**[1]   **Yanan Zheng**[1]
**Yilun Zhao**[1]   **Tesca Fitzgerald**[1]   **Arman Cohan**[12]
[1]Yale University   [2]Allen Institute for AI
{ziyao.shangguan, chuhan.li.cl2575}@yale.edu
https://github.com/yale-nlp/TOMATO

## Abstract

Existing benchmarks often highlight the remarkable performance achieved by state-of-the-art Multimodal Foundation Models (MFMs) in leveraging temporal context for video understanding. However, *how well do the models truly perform visual temporal reasoning?* Our study of existing benchmarks shows that this capability of MFMs is likely overestimated as many questions can be solved by using a single, few, or out-of-order frames. To systematically examine current visual temporal reasoning tasks, we propose three principles with corresponding metrics: (1) *Multi-Frame Gain*, (2) *Frame Order Sensitivity*, and (3) *Frame Information Disparity*. Following these principles, we introduce TOMATO, **T**emp**O**ral Reasoning **M**ultimod**A**l Evalua**Ti**O**n, a novel benchmark crafted to rigorously assess MFMs' temporal reasoning capabilities in video understanding. TOMATO comprises 1,484 carefully curated, *human-annotated* questions spanning *six* tasks (*i.e., action count*, *direction*, *rotation*, *shape & trend*, *velocity & frequency*, and *visual cues*), applied to 1,417 videos, including 805 self-recorded and -generated videos, that encompass human-centric, real-world, and simulated scenarios. Our comprehensive evaluation reveals a human-model performance gap of 57.3% with the best-performing model. Moreover, our in-depth analysis uncovers more fundamental limitations beyond this gap in current MFMs. While they can accurately recognize events in isolated frames, they fail to interpret these frames as a continuous sequence. We believe TOMATO will serve as a crucial testbed for evaluating the next-generation MFMs and as a call to the community to develop AI systems capable of comprehending the human world dynamics through the video modality.

## 1 Introduction

Visual temporal reasoning, an important aspect of human perception, refers to the cognitive process of understanding and interpreting sequences of visual information over time, such as recognizing patterns of motions, detecting changes in scenery, and following the progression of events (Kahneman et al., 1992). Currently, state-of-the-art methods for addressing visual temporal reasoning are centered on the use of Multimodal Foundation Models (MFMs) (OpenAI, 2024; Anthropic, 2024b; Wang et al., 2024a; Fei et al., 2024), which have shown remarkable performance across numerous temporal reasoning video benchmarks (Li et al., 2023; Liu et al., 2024; Li et al., 2024b; Chen et al., 2024a). However, despite these impressive performances, our study in §5 has shown that the models' true capabilities in visual temporal reasoning are likely overestimated.

This study examines four existing widely-used temporal reasoning video benchmarks (Li et al., 2023; 2024b; Liu et al., 2024; Chen et al., 2024a), as illustrated in Figure 1. We identify patterns in their question formulation that allow models to exploit shortcuts, enabling them to answer correctly using a single, few, or out-of-order frames. To rigorously evaluate whether a benchmark effectively assesses MFMs' visual temporal reasoning ability, we propose three principles with corresponding metrics:

---

*Equal Contribution.

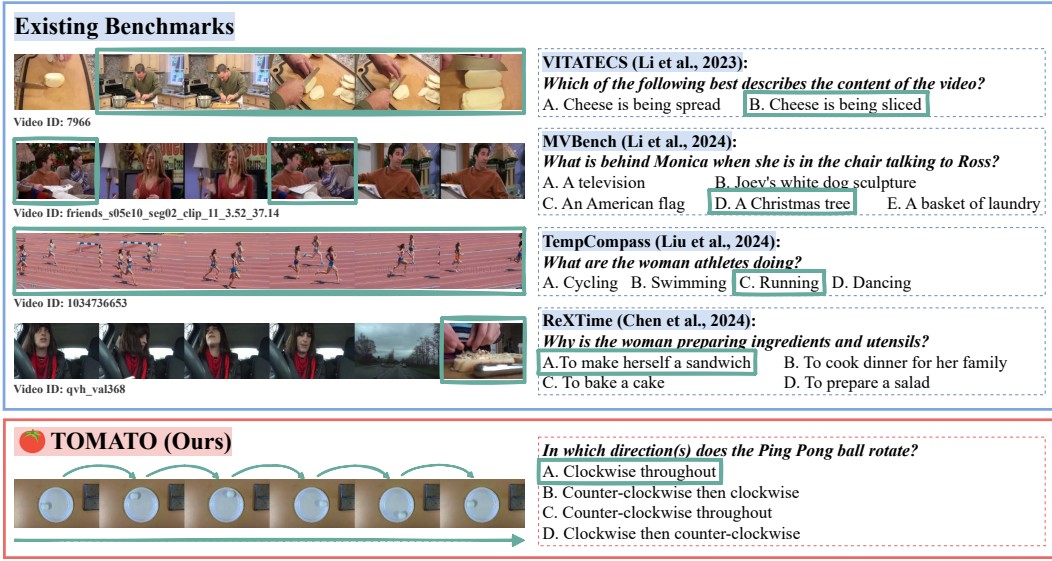

Figure 1: Examples of several VideoQA benchmarks. We examine four existing representative benchmarks and ours: VITATECS, MVBench, TempCompass, ReXTime, and TOMATO. In the four existing benchmarks, we use bounding boxes to highlight frames that models can exploit as shortcuts, allowing questions to be answered using *any one* of highlighted frames. In TOMATO, however, models must reason both between consecutive frames and across all frames as a continuous sequence.

(1) *Multi-Frame Gain*, where a *higher* value indicates the task is less solvable by a single frame (§3.1), (2) *Frame Order Sensitivity*, where a *higher* value indicates the task is more reliant on the correct order of frames (§3.2), and (3) *Frame Information Disparity*, where a *lower* value indicates information is more evenly distributed across the frames (§3.3). Our analysis reveals that, on average, existing benchmark exhibit a *Multi-Frame Gain* of less than 5%, compared to 66.3% for ours (§5.1); a *Frame Order Sensitivity* of less than 8%, compared to 34.1% for ours (§5.2); and a *Frame Information Disparity* of over 27%, while ours is only 4.6% (§5.3). These results suggest that tasks from existing benchmarks are relatively more solvable by a single frame, rely less on the order of the frames, and allow for more unevenly distributed information across the frames. Therefore, models' visual temporal reasoning capabilities, crucial to MFMs' video comprehension, are likely overestimated.

Following the three aforementioned principles for building a more effective visual temporal reasoning benchmark, we introduce TOMATO, **T**emp**O**ral Reasoning **M**ultimod**A**l Evalua**Ti**O**n, a novel video understanding benchmark designed to explicitly assess MFMs' temporal reasoning capabilities. TOMATO comprises 1,484 carefully curated, *human-annotated* multiple-choice questions spanning *six* distinct temporal reasoning tasks (*action count*, *direction*, *rotation*, *shape & trend*, *velocity & frequency*, and *visual cues*) applied to 1,417 videos encompassing human-centric, real-world, and simulated scenarios (§4.1). TOMATO features a diverse collection of videos, sourcing from YouTube, four existing video datasets (Jang et al., 2017; Yi et al., 2020; Li et al., 2022; Pătrăucean et al., 2023), as well as self-recorded and -generated videos (§4.2). To enhance diversity, we carefully edit the sourced YouTube and self-recorded videos, incorporating various characteristics such as counter-factual scenarios, composite motions, and zoomed-in views. In creation of the 805 original videos, we explicitly include both human-centric and simulated scenarios to address the lack of existing videos that capture human interactions or generated synthetic scenes. The question-answering (QA) pairs are meticulously designed to ensure that models need to reason about the transitions across all frames, making visual temporal reasoning essential for solving the tasks.

We conduct a comprehensive evaluation of 21 open-source models and 10 proprietary models. Notably, the best-performing open-source model, Qwen2-VL-72B, which achieves 37.9% overall accuracy, outperforms all proprietary models, including GPT-4o, which achieves 37.7% overall accuracy. However, both types of models remain significantly below human-level performance, which reaches 95.2% using full videos and 79.7% with 16 frames. Moreover, our analysis goes beyond highlighting these performance gap, and it exposes deeper, more fundamental limitations in current

MFMs' capabilities (§6.3). Specifically, we show that these models: (1) lack the basic ability to interpret the frames as a continuous sequence, (2) fail to truthfully leverage the visual input while being over-reliant on common sense, and (3) are highly susceptible to noisy information. We hope that our findings can provide useful insights for future work in guiding development of improved MFMs.

In summary, our contributions are:

- We identify an inflated performance of MFMs on existing visual temporal reasoning benchmarks and establish *three* principles with metrics to evaluate the effectiveness of benchmarks in assessing models' visual temporal reasoning capabilities.
- We introduce TOMATO, a novel benchmark for assessing MFMs' capabilities on visual temporal reasoning tasks, spanning *six* reasoning types and *three* video scenarios, including 805 self-created and -generated videos.
- We present a comprehensive evaluation of 21 open-source models and 10 proprietary models on TOMATO, revealing a substantial gap between human-level and MFM-enabled visual temporal reasoning capabilities.
- We provide in-depth error case analysis, uncovering more fundamental shortcomings in MFMs' visual temporal reasoning capabilities that go beyond the human-model performance gap.

## 2 RELATED WORK

**General video understanding benchmarks.** Video understanding capability plays a pivotal role in multimodal learning and is a key step toward achieving artificial general intelligence. Prior to the era of MFMs, early benchmarks (Heilbron et al., 2015; Goyal et al., 2017; Sigurdsson et al., 2018) focus primarily on action recognition. However, more recent benchmarks (Jang et al., 2017; Yi et al., 2020) represent a shift towards evaluating models' ability to reason about temporal dynamics and causal events. The rise of MFMs has further propelled the field toward more complex, human-like understanding tasks. These tasks include (1) long-form video understanding (Zhang et al., 2023; Rawal et al., 2024; Zhou et al., 2024; Nagrani et al., 2024), (2) multi-disciplinary video understanding (He et al., 2024), and (3) comprehensive evaluation across various tasks (Ning et al., 2023; Li et al., 2024d;c; Fu et al., 2024). Building upon this remarkable progress, MMBench-Video (Fang et al., 2024) advocates the needs for more temporal questions since many questions in existing benchmarks are rendered rather "static," but its reasoning dimension such as Attribute Recognition, Object Recognition, and OCR still remain static. This notion motivates the creation of TOMATO, a benchmark specifically designed to evaluate MFMs' visual temporal reasoning capabilities.

**Visual temporal reasoning benchmarks.** Several benchmarks have been developed to specifically evaluate models' visual temporal reasoning capabilities. For instance, VITATECS (Li et al., 2023) introduces six temporal reasoning tasks (*e.g., "A man is putting on a tie or putting off a tie?"*) and asks models to distinguish between the correct and counter-factual caption. Addressing the lack of tasks diversity in VITATECS, TempCompass (Liu et al., 2024) expands tasks types to include multiple-choice QA, yes/no QA, caption matching, and caption generation tasks. Aiming to cover a wide-range of temporal-sensitive videos, MVBench (Li et al., 2024b) defines nine core temporal tasks with 20 subtasks, each of which cannot be solved using a single frame. Similarly, ReXTime (Chen et al., 2024a) targets comprehensive temporal reasoning tasks and puts a special emphasis on cause and effect samples. However, despite these efforts, as illustrated in Figure 1, we observe that many questions in these benchmarks can be answered correctly using single, few, or out-of-order frames (§5), limiting their effectiveness to evaluate models' true visual temporal reasoning capabilities. To address these shortcomings, we introduce TOMATO, a benchmark designed to provide a more rigorous evaluation of visual temporal reasoning.

## 3 BENCHMARKING PRINCIPLES FOR VISUAL TEMPORAL REASONING TASKS

In this section, we define three key principles and corresponding metrics for assessing how rigorously a benchmark targets visual temporal reasoning rather than static image understanding.

### 3.1 MULTI-FRAME GAIN

**Key principle:** A visual temporal reasoning task should require reasoning across multiple frames, making it impossible for models to solve the task using 1 frame.

Requiring models to reason across multiple frames ensures that the tasks are distinguished from static image recognitions. This requirement aligns with prior works (Lei et al., 2022; Fang et al., 2024), which highlight that many video understanding tasks rely heavily on static visual information.

To assess this principle, we define *Multi Frame Gain* $\kappa$, which measures the relative performance gain from using a single frame to $m$ frames as input. When considering a single frame, we examine both settings: (1) a random frame and (2) a handpicked, highly informative frame specific to the question. The $m$ frames are uniformly sampled at equal intervals from the videos. Using $Acc(m \text{ frames})$ to denote the model's accuracy in solving benchmark tasks using $m$ frames, and $\epsilon$ a small positive constant, we define $\kappa$ as:

$$\kappa = \frac{Acc(m \text{ frames})}{Acc(1 \text{ frame}) + \epsilon} - 1$$

A lower $\kappa$ value indicates that the question can be more accurately answered using a single frame, while a higher $\kappa$ value indicates a necessity to reason across multiple frames.

### 3.2 FRAME ORDER SENSITIVITY

**Key principle:** A visual temporal reasoning task, when given multiple frames, should enforce constraints on maintaining the correct order of frames.

Shuffling video frames disrupts the temporal information throughout the video, as demonstrated in the prior works (Misra et al., 2016; Dorkenwald et al., 2022; Hao et al., 2022). If a task is solvable by shuffled frames, time dependencies across frames are absent, and reasoning along the time dimension is unnecessary, which disqualifies the task from being temporal.

To quantify this principle, we introduce *Frame Order Sensitivity* $\tau$, which measures the relative performance gain from using the shuffled $m$ frames to the ordered $m$ frames. Using $Acc(m \text{ frames})$ to denote the model's accuracy in solving benchmark tasks using $m$ frames, and $\epsilon$ a small positive constant, we define $\tau$ as:

$$\tau = \frac{Acc(m \text{ frames})}{Acc(\text{shuffled } m \text{ frames}) + \epsilon} - 1$$

A lower $\tau$ value indicates the question can be answered more accurately using out-of-order frames, while a higher $\tau$ value suggests a stronger reliance on the original order of the frames.

### 3.3 FRAME INFORMATION DISPARITY

**Key principle:** A visual temporal reasoning task, when given multiple frames, should ensure that each frame contributes relatively evenly to solving the task.

Even contribution suggests that no single frame provides disproportionately more information. Even in tasks involving sequential events, where the number of events aligns with the number of frames necessary to answer the question accurately, models should not achieve significantly higher accuracy by relying on a handpicked single frame over any random single frame. This principle aligns with prior works (Huang et al., 2018; Liu et al., 2021) in which they emphasize the necessity of leverage temporal information from all frames to ensure an accurate and robust temporal video understanding.

To quantitatively evaluate this principle, we introduce *Frame Information Disparity* $\rho$, which measures the relative performance gain achieved by switching from a random single frame to a handpicked single frame. Using $Acc(m \text{ frames})$ to denote the model's accuracy in solving benchmark tasks using $m$ frames, and $\epsilon$ a small positive constant, we define $\rho$ as:

$$\rho = \frac{Acc(\text{handpicked 1 frame})}{Acc(\text{random-sampled 1 frame}) + \epsilon} - 1$$

A higher $\rho$ value indicates that the question can be more accurately answered by a handpicked single frame compared to a random single frame, while a lower $\rho$ value indicates a more even distribution of informativeness across the multiple frames. In other words, ideally, a benchmark with perfectly even distribution of informativeness among all the frames should achieve a $\rho$ value of 0.

# 4 TOMATO: A VISUAL TEMPORAL REASONING BENCHMARK

We introduce TOMATO, a new visual temporal reasoning benchmark that satisfies all three afore-mentioned principles, addressing issues where tasks from existing benchmarks are relatively more solvable by a single frame, rely less on the order of frames, and allow for more unevenly distributed information across the frames (§5). TOMATO comprises 1,484 carefully curated, *human-annotated* questions spanning *six* visual temporal reasoning tasks, applied to 1,417 videos, including 805 self-recorded and -generated videos, that encompass *human-centric*, *real-world*, and *simulated scenarios*. In the following sections, we describe temporal tasks in TOMATO (§4.1), video collection (§4.2), and question annotation §4.3. The main statistics of TOMATO are presented in Table 9.

## 4.1 TEMPORAL TASKS IN TOMATO

We introduce *six* visual temporal reasoning tasks, each of them requiring multi-frame visual temporal reasoning: (1) *Rotation*: Determine the rotational direction of of the subject; (2) *Direction*: Identify the direction of the subject's movement; (3) *Velocity & Frequency*: Detect changes in the subject's movement speed or variations in the frequency of repeated actions; (4) *Shape & Trend*: Analyze the subject's trajectory, such as the shape or general trend of its movement; (5) *Visual Cues*: Discern key visual signals to determine the sequence or timing of actions without audio; and (6) *Action Count*: Count how many times a specific action has been performed. Examples are provided in Table 6.

To ensure comprehensive coverage across various scenarios, we categorized each video into one of three *demonstration categories*: (1) *Human-centric*: Involving human interactions, where actions or intentions are observed; (2) *Real-world*: Focuses on actions involving objects in various real-world scenes; and (3) *Simulated*: Depicting simplified, simulated environments representing temporal actions. The distribution of demonstration categories across each task is shown in Figure 3 in §E.2.

## 4.2 VIDEO COLLECTION

TOMATO features a diverse range of videos from three distinct sources: *YouTube*, *existing video datasets*, and *self-recorded and -generated benchmark-specific videos*. To enhance diversity, we collect and create videos of three scenarios: *real-world*, *human-centric*, and *simulated*. Additionally, we edit videos to incorporate *counterfactual scenes*, *composite motions*, and *zoomed-in views*, aiming to investigate the impact of these characteristics on the performance of MFMs (§6.3). License information for all videos is detailed in §F.

**YouTube videos.** We chose YouTube as our primary source to ensure diverse and representative real-world scenarios in our benchmark. Given a specific visual temporal reasoning task, we task human annotators with searching for YouTube videos that best represented the corresponding task definition (§4.1). The selected videos cover a wide array of topics, including science experiments, outdoor activities, educational tutorials, and artistic performances. To prevent models from relying on commonsense knowledge, such as assuming *the second hand of a watch rotates clockwise*, we edit videos to create counterfactual scenarios. These edits, including reversing, concatenating, adjusting speed, and mirroring, ensure that models must fully analyze video content to answer questions correctly. In total, we collect 171 source videos before editing and 398 videos after editing.

**Existing video datasets.** It is essential for AI agents to comprehend the human world through detecting the point of change in actions (*Visual Cues*) and discern the number of actions occurred (*Action Count*). Therefore, to address the lack of diversity in YouTube videos for these reasoning types, we incorporate four established datasets, each contributing to unique domains and scenarios: (1) *Music-AVQA* (Li et al., 2022) featuring multi-instrument performances; (2) *CLEVRER* (Yi et al., 2020) presenting synthetic scenes with multiple moving objects; (3) *TGIF-QA* (Jang et al., 2017) focusing on action counting in varied scenes; and (4) *Perception Test* (Pătrăucean et al., 2023)

addressing indoor action counting tasks, such as clapping and object moving. For *Music-AVQA* and *CLEVER*, we re-annotate and re-write questions to specifically inquire the temporal contexts of these videos (§4.3). In total, we incorporate 214 videos: 70 from Music-AVQA, 50 from CLEVRER, 50 from TGIF-QA, and 44 from Perception Test.

**Self-recorded and -generated benchmark-specific videos.** For an AI agent to truly comprehend the human world dynamics, it is essential for it to understand scenarios where a person is actively interacting with the agent, treating the agent as the "other" participant of an interaction. However, YouTube videos and existing datasets lack such human-centric interactive scenes for various reasoning types in TOMATO (§4.1). To address this limitation, we record videos such as a human conveying "turn-around" by twirling their wrist while maintaining a pointing gesture, or drawing a shape in the air using their arm. Furthermore, to aid data collection and study the effect of simulated videos, we generate more simple, abstract representations of both real-world objects and human. Thus, we expand our dataset by (1) creating simulated videos using Keynote[1], featuring objects moving in different patterns; and (2) generating 3D human model videos through the VIBE (Kocabas et al., 2019) and SMPL (Bogo et al., 2016) frameworks. To study the effect of different video characteristics, we edit videos to incorporate counter-factual scenes, composite motions, and zoomed-in views. In total, we create 298 videos before editing and 805 videos after editing, spanning all *six* tasks.

## 4.3 QUESTION-ANSWER ANNOTATION AND QUALITY CHECK.

**QA annotation.** To address the limitation that tasks from existing benchmarks are solvable by a single, a few, or out-of-order frames (discussed further in §5), TOMATO focuses on crafting questions that require reasoning about transitions across all frames. Our annotation process varied by video sources. For YouTube videos, self-created benchmark-specific videos, and their edits, human annotators composed QA pairs targeting specific temporal reasoning tasks (§4.1).

**Quality check.** To ensure high-quality QA annotations, we implemented a three-stage process: initial annotation by annotators, followed by cross-checking and verification among annotators, and finally, collective resolution of disagreements with a final review (see §G). This rigorous approach ensured consistency and accuracy across all annotated QAs.

## 5 COMPARISONS AMONG VISUAL TEMPORAL REASONING BENCHMARKS

In §3, we defined three key principles with corresponding metrics to assess how effectively a benchmark addresses visual temporal reasoning. Using these metrics, we compare TOMATO with four recent visual temporal reasoning benchmarks: VITATECS (Li et al., 2023), MVBench (Li et al., 2024b), TempCompass (Liu et al., 2024), and ReXTime (Chen et al., 2024a). To conduct this comparison, we randomly sampled approximately 200 QAs[2] from these benchmarks, using two state-of-the-art MFMs: GPT-4o (OpenAI, 2024) and Qwen2-VL-72B (Wang et al., 2024a). For metrics that requires handpicking frames, we present annotators with the full video and corresponding question, and ask them to select the most informative frame for each benchmark. For metrics requiring multiple frames, we set $m = 16$, as our study across $m = 1, 8, 16, 32$ demonstrates that 16 frames provide a sufficient window for effective analysis (§E.3).

### 5.1 MULTI-FRAME GAIN

We present results on *Multi-frame Gain* for both (1) a random frame (Table 1) and (2) a handpicked, highly informative frame (Table 2). As shown in the two tables, TOMATO achieves a significantly higher $\kappa$ value in using both random single frame and handpicked single frame. This significant relative performance gain from the single frame input setting to the 16 frames input shows the necessity in our tasks to reason using multiple frames. In comparison of the two tables, we observe an expected decrease in the $\kappa$ value in using handpicked single frame compared to random single frame on existing benchmarks, indicating the relative ease in answering the benchmark questions

---

[1] https://support.apple.com/keynote

[2] Sampling varies slightly: TempCompass: 200 samples. MVBench: 204 samples (excluding overlapping reasoning types). VITATECS: 203 samples. ReXTime: 180 samples from QVHighlight's validation set.

Table 1: Performance using single-frame and 16-frame inputs.

| # Frames | VITATECS | | | MVBench | | | TempCompass | | | ReXTime | | | TOMATO | | |
|---|---|---|---|---|---|---|---|---|---|---|---|---|---|---|---|
| | 1 | 16 | $\kappa \uparrow$ | 1 | 16 | $\kappa \uparrow$ | 1 | 16 | $\kappa \uparrow$ | 1 | 16 | $\kappa \uparrow$ | 1 | 16 | $\kappa \uparrow$ |
| GPT-4o | 70.0 | 88.2 | 26.1 | 47.1 | 62.3 | 32.3 | 52.5 | 71.5 | 36.2 | 61.7 | 78.3 | 26.9 | 21.2 | 37.7 | 78.0 |
| Qwen2-VL | 71.4 | 86.2 | 20.7 | 47.1 | 63.2 | 34.4 | 50.0 | 79.0 | 58.0 | 63.9 | 81.7 | 27.8 | 20.6 | 37.9 | 84.0 |
| Average | 70.7 | 87.2 | 23.4 | 47.1 | 62.7 | 33.3 | 51.3 | 75.3 | 47.1 | 62.8 | 80.0 | 27.4 | 20.9 | 37.8 | **81.0** |

Table 2: Performance using handpicked single-frame (denoted by **1[H]**) and 16-frame inputs.

| # Frames | VITATECS | | | MVBench | | | TempCompass | | | ReXTime | | | TOMATO (200) | | |
|---|---|---|---|---|---|---|---|---|---|---|---|---|---|---|---|
| | 1[H] | 16 | $\kappa \uparrow$ | 1[H] | 16 | $\kappa \uparrow$ | 1[H] | 16 | $\kappa \uparrow$ | 1[H] | 16 | $\kappa \uparrow$ | 1[H] | 16 | $\kappa \uparrow$ |
| GPT-4o | 84.2 | 88.2 | 4.7 | 59.8 | 62.3 | 4.1 | 64.5 | 71.5 | 10.9 | 86.7 | 78.3 | -9.7 | 21.5 | 37.0 | 72.1 |
| Qwen2-VL | 87.7 | 86.2 | -1.7 | 57.8 | 63.2 | 9.3 | 63.5 | 79.0 | 24.4 | 85.6 | 81.7 | -4.6 | 24.0 | 38.5 | 60.4 |
| Average | 86.0 | 87.2 | 1.5 | 58.8 | 62.7 | 6.7 | 64.0 | 75.3 | 17.6 | 86.1 | 80.0 | -7.1 | 22.8 | 37.8 | **66.3** |

Table 3: Performance using shuffled 16-frame (denoted by **16[S]**) and 16-frame inputs.

| # Frames | VITATECS | | | MVBench | | | TempCompass | | | ReXTime | | | TOMATO | | |
|---|---|---|---|---|---|---|---|---|---|---|---|---|---|---|---|
| | 16[S] | 16 | $\tau \uparrow$ | 16[S] | 16 | $\tau \uparrow$ | 16[S] | 16 | $\tau \uparrow$ | 16[S] | 16 | $\tau \uparrow$ | 16[S] | 16 | $\tau \uparrow$ |
| GPT-4o | 85.7 | 88.2 | 2.9 | 59.8 | 62.3 | 4.1 | 59.0 | 71.5 | 21.2 | 77.8 | 78.3 | 0.6 | 25.8 | 37.7 | 46.2 |
| Qwen2-VL | 83.3 | 86.2 | 3.6 | 58.8 | 63.2 | 7.5 | 64.0 | 79.0 | 23.4 | 81.7 | 81.7 | 0 | 31.1 | 37.9 | 21.9 |
| Average | 84.5 | 87.2 | 3.2 | 59.3 | 62.7 | 5.8 | 61.5 | 75.3 | 22.3 | 79.7 | 80.0 | 0.3 | 28.5 | 37.8 | **34.1** |

Table 4: Performance using single frame and handpicked single-frame (denoted by **1[H]**) inputs.

| # Frames | VITATECS | | | MVBench | | | TempCompass | | | ReXTime | | | TOMATO (200) | | |
|---|---|---|---|---|---|---|---|---|---|---|---|---|---|---|---|
| | 1 | 1[H] | $\rho \downarrow$ | 1 | 1[H] | $\rho \downarrow$ | 1 | 1[H] | $\rho \downarrow$ | 1 | 1[H] | $\rho \downarrow$ | 1 | 1[H] | $\rho \downarrow$ |
| GPT-4o | 70.0 | 84.2 | 20.4 | 47.1 | 59.8 | 27.1 | 52.5 | 64.5 | 22.9 | 61.7 | 86.7 | 40.5 | 20.5 | 21.5 | 4.9 |
| Qwen2-VL | 71.4 | 87.7 | 22.7 | 47.1 | 57.8 | 22.9 | 50.0 | 63.5 | 27.0 | 63.9 | 85.6 | 33.9 | 23.0 | 24.0 | 4.3 |
| Average | 70.7 | 86.0 | 21.6 | 47.1 | 58.8 | 25.0 | 51.3 | 64.0 | 24.9 | 62.8 | 86.1 | 37.2 | 21.8 | 22.8 | **4.6** |

using handpicked single frame. Interestingly, the $\kappa$ values are negative in the handpicked single frame setting on ReXTime; this negative performance gain in using more frames might stem from noise introduced by additional frames.

## 5.2 FRAME ORDER SENSITIVITY

In our shuffled 16 frames setting, we apply random shuffling on the ordered 16 frames to ensure that the same set of frames are used in both settings. As shown in Table 3, TOMATO achieves significantly higher $\tau$, demonstrating that our benchmark imposes a stricter requirement on maintaining the order of the frames to accurately answer the question.

## 5.3 FRAME INFORMATION DISPARITY

As shown in Table 4, the performance gain from random single frame to handpicked single frame is the lowest on TOMATO, indicating a relatively more consistent informativeness across all frames compared to other existing benchmarks.

## 6 EVALUATING VISUAL TEMPORAL REASONING IN ADVANCED MFMS

In introducing TOMATO, we present a comprehensive evaluation of 31 MFMs, including 10 proprietary models and 21 open-source models, to assess their visual temporal reasoning capabilities. In the following sections, we detail the experimental setup (§6.1), evaluation results (§6.2), and

Table 5: Evaluation results on TOMATO. Unless otherwise specified, each model processes 16 frames. **Bold** and underlined numbers indicate the best and second-best performance in each category, respectively. See Table 8 for detailed model configurations. ‡: Models that can process video directly. *: Models that can only process eight frames. Details for these categories are shown in §4.1

| Model | Rotation | Direction | Velocity & Frequency | Shape & Trend | Visual Cues | Action Count | All |
|---|---|---|---|---|---|---|---|
| | (286) | (403) | (210) | (223) | (70) | (292) | (1,484) |
| *Baselines* | | | | | | | |
| Human (Videos) | 93.5 | 95.4 | 94.1 | 100.0 | 95.0 | 93.6 | 95.2 |
| Human (Frames) | 60.9 | 93.9 | 85.3 | 91.7 | 60.0 | 70.2 | 79.7 |
| Random (GPT-4o) | 16.8 | 17.4 | 35.7 | 29.6 | 32.9 | 20.9 | 23.1 |
| Random Choice (42) | 22.0 | 17.6 | 22.9 | 17.9 | 18.6 | 13.4 | 18.5 |
| Frequent Choice | 21.5 | 18.5 | 24.8 | 20.2 | 33.6 | 18.8 | 21.0 |
| *Proprietary Multimodal Foundation Models (MFMs)* | | | | | | | |
| GPT-4o | 24.5 | **45.2** | 31.9 | **42.6** | **58.6** | 36.0 | **37.7** |
| Gemini 1.5 pro ‡ | 25.9 | 40.7 | **35.2** | 41.3 | 37.1 | **36.3** | 36.1 |
| Gemini 2.0 Flash Exp ‡ | 23.8 | 39.0 | 34.3 | 36.8 | 48.6 | 29.8 | 33.7 |
| GPT-4o-mini | 19.9 | 32.8 | 28.1 | 29.6 | 41.4 | 28.8 | 28.8 |
| Claude 3 Opus | **31.1** | 23.3 | 32.4 | 27.8 | 28.6 | 29.5 | 28.2 |
| Claude 3.5 Sonnet | 27.3 | 25.6 | 26.2 | 27.8 | 32.9 | 31.2 | 27.8 |
| Gemini 1.5 Flash ‡ | 22.4 | 30.3 | 31.0 | 26.9 | 30.0 | 27.7 | 27.8 |
| Claude 3 Haiku | 25.9 | 19.4 | 31.0 | 33.2 | 30.0 | 26.4 | 26.2 |
| Reka Flash ‡ | 19.6 | 26.6 | 10.0 | 21.5 | 32.9 | 16.8 | 20.5 |
| Reka Core ‡ | 14.3 | 17.6 | 9.5 | 18.8 | 22.9 | 12.7 | 15.3 |
| *Open-Source Multimodal Foundation Models (MFMs)* | | | | | | | |
| Qwen2-VL-72B | 26.9 | **38.2** | **43.8** | 36.3 | 48.6 | **42.8** | **37.9** |
| Qwen2-VL-7B | 23.8 | 29.5 | 41.9 | 29.6 | 37.1 | 34.2 | 31.5 |
| Video-CCAM-v1.1 14B | **32.2** | 26.1 | 29.5 | 27.4 | 44.3 | 35.6 | 30.7 |
| InternVL 2 40B | 23.4 | 32.0 | 15.7 | **41.7** | 34.3 | 29.1 | 29.0 |
| LLaVA-Video-72B | 18.9 | 30.0 | 16.7 | 32.3 | **50.0** | 34.6 | 28.2 |
| Video-CCAM-v1.1 9B | 22.4 | 25.6 | 25.7 | 26.0 | 34.3 | 33.6 | 27.0 |
| InternVideo 2 8B | 31.8 | 24.1 | 23.3 | 25.6 | 35.7 | 25.0 | 26.4 |
| Qwen2-VL-2B | 26.9 | 22.8 | 31.4 | 23.8 | 32.9 | 25.7 | 26.0 |
| LLaVA-OneVision 7B | 16.8 | 25.1 | 23.8 | 28.7 | 35.7 | 31.2 | 25.5 |
| Video-CCAM-v1.1 4B | 21.7 | 24.3 | 19.0 | 27.4 | 32.9 | 31.5 | 25.3 |
| LLaVA-Video-7B | 20.1 | 23.8 | 15.2 | 29.1 | 41.4 | 29.8 | 24.9 |
| VILA 13B | 29.0 | 19.6 | 19.0 | 27.4 | 32.9 | 27.7 | 24.7 |
| LLaVA-Video-7B-Video-Only | 15.4 | 24.1 | 19.5 | 31.4 | 38.6 | 25.7 | 23.9 |
| Video LLaVA 7B * | 29.4 | 17.9 | 27.1 | 23.3 | 34.3 | 20.9 | 23.6 |
| VideoLLaMA 2 72B ‡ | 14.3 | 24.6 | 22.4 | 26.5 | 27.1 | 28.8 | 23.5 |
| InternVL 2 26B | 18.5 | 29.3 | 10.5 | 31.4 | 11.4 | 25.7 | 23.3 |
| LLaVA-NeXT-Video-32B | 20.6 | 26.3 | 12.4 | 24.2 | 30.0 | 24.3 | 22.7 |
| InternVL 2 8B | 17.1 | 25.1 | 9.0 | 28.7 | 31.4 | 22.9 | 21.7 |
| Phi 3.5 Vision | 20.3 | 16.6 | 14.3 | 23.3 | 40.0 | 24.7 | 20.7 |
| AuroraCap-7B-VID | 19.6 | 17.1 | 26.7 | 20.6 | 25.7 | 17.1 | 20.6 |
| VideoLLaMA 2 7B ‡ | 10.1 | 22.8 | 15.7 | 18.8 | 31.4 | 19.5 | 18.5 |

a multi-faceted analysis (§6.3), considering factors such as model architectures, reasoning type correlations, frame counts, and video characteristics.

## 6.1 EXPERIMENTAL SETUP

**Models.** We evaluate a diverse set of general-purpose MFMs. For *open-source* MFMs, we test: VILA (Lin et al., 2023b), InternVL 2 (Chen et al., 2024b) Phi 3.5 Vision (Team, 2024), Video LLaVA (Lin et al., 2023a), InternVideo 2 (Wang et al., 2024b), LLaVA-NeXT-Video (Zhang et al., 2024a), LLaVA-OneVision (Li et al., 2024a), VideoLLaMA2 (Cheng et al., 2024), Qwen2-VL (Wang et al., 2024a), VideoCCAM (Fei et al., 2024), AuroraCap (Chai et al., 2024), and LLaVA-Video (Zhang et al., 2024b). We also evaluate the following *proprietary* MFMs: GPT-4o (OpenAI, 2024), Claude (Anthropic, 2024a;b), Reka (Reka, 2024), and Gemini (Gemini, 2024). For all models, we provide generation configuration in Table 8.

**Baselines.** We include a text-only baseline, denoted as Random (GPT-4o), in which we prompt GPT-4o to guess the answer without access to the videos (prompt in §D.3). Additionally, we report results for Random Guess and Frequent Guess baselines. Furthermore, we evaluate human performance on

TOMATO, reporting results for humans using video input, *i.e.,* Human (Video), and 16 frames as input, *i.e.,* Human (Frames). Annotator biographies are provided in Table 10 in §G.1.

## 6.2 EXPERIMENTAL RESULTS

We provide quantitative results on TOMATO for all models in Table 5. To better understand where models fail, we select a set of representative models (OpenAI, 2024; Anthropic, 2024b; Wang et al., 2024a; Fei et al., 2024; Chen et al., 2024b) and present examples of failure cases in §I, §J, §K, §L, §M, and §N. Analysis of video scenarios (counterfactual, zoomed-in, and first-person) is in §B.

**Widespread difficulty in visual temporal reasoning.**   Our evaluation (Table 5) underscores the significant challenges of TOMATO across all tested models. The leading open-source model, Qwen2-VL-72B, achieves 37.9% accuracy, slightly outperforming GPT-4o's 37.7%. However, this still leaves a substantial 57.3% performance gap compared to human accuracy of 95.2%. While this result demonstrates the competitive potential of open-source models in video understanding, many still fall below 30.0%, indicating weaknesses in TOMATO despite their decent performance in existing benchmarks (Pătrăucean et al., 2023; Li et al., 2024b; Fu et al., 2024).

## 6.3 ANALYSIS

**Models lack the basic ability to interpret frames as a continuous sequence.**   While MFMs demonstrate remarkable performance in understanding sequential events in videos (Chen et al., 2024a), our benchmark exposes a more fundamental limitation: models struggle to reason across multiple time steps and to interpret the frames as a continuous sequence. As shown in error case §I.2.1, GPT-4o correctly generates captions for each consecutive change in the moon's movement, showcasing its ability to reason at individual time steps. However, it fails to infer based on the captions that the overall sequence represents a clockwise rotation. This issue is not limited to *rotation* (§I.1.1, §I.2.1); similar shortcomings are observed in *direction* (§J.3.3), *action count* (§N.2.5), *etc.*.

**Models fail to truthfully leverage the visual input while being over-reliant on common sense.**
In our evaluations, despite explicit instructions (§D.1) to rely on the visual input rather than common sense, we find that models frequently hallucinate based on information from single frames rather than utilizing true *visual* reasoning. For instance, in error case §K.2.4, GPT-4o incorrectly concludes that an object is dropping due to the presence of motion blur in some of the frames. However, this is a reversed video where the object is actually only moving upward - a conclusion that can only be reached if the video modality where truthfully utilized. A similar instance of the limitation occurs in error case §J.1.2, where GPT-4o likely assumes the person raises their hand first to reach the posture depicted in the first 8 single frames. In reality, the person's hand remains relatively stationary throughout these frames, but the model fails to make accurate visual comparisons across these frames.

**Models are highly susceptible to noisy information in the input.**   As demonstrated in error case §J.3.3, where a block moves downwards, models are especially vulnerable to noisy information, such as misleading text on the block. In particular, while GPT-4o correctly describes the block's downward motion based on its relative position to the screen, it incorrectly concludes that the block is moving upward, likely influenced by the false information presented in the text written on the block. Similarly, in error case §N.2.5, a butterfly-shaped laser spot is moving in a triangle shape. However, the unique butterfly shape likely causes the models to lose focus on the trajectory of the laser spot, and resort to random guessing in their conclusions.

**Explicitly incorporating time-aware positional encoding can likely enhance visual temporal reasoning.**   The Qwen2-VL family (Wang et al., 2024a), which consistently achieves the highest scores across multiple categories, leverages Multimodal Rotary Positional Encoding (M-RoPE) within its visual encoders. M-RoPE explicitly encodes temporal information into visual tokens, allowing models to retain critical temporal context throughout the entire pipeline. In contrast, models without such temporal-aware positional encoding scheme appear to lose critical time-related context after visual encoding, such as Causal Cross-Attention Masks (CCAM) used in vision-language alignment (Fei et al., 2024), Unmasked Video Token Reconstruction during training (Wang et al., 2024b), and Spatial-Temporal Convolution (STC) connector applied after frozen visual encoder (Cheng et al.,

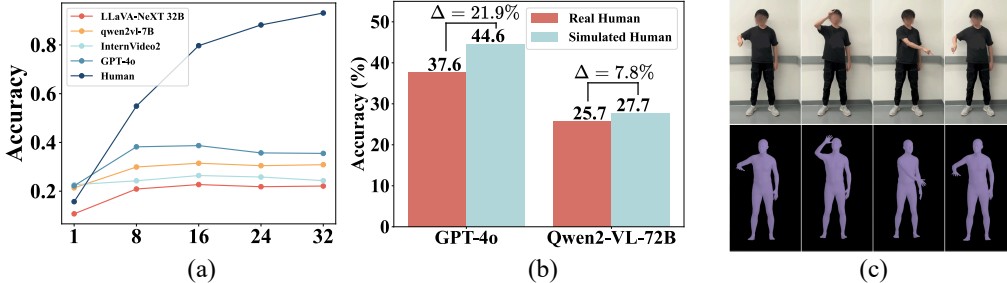

Figure 2: (a) Human and models' performance on TOMATO with different number of frames. (b) Real human and simulated human's performance on TOMATO. (c) Example of real human vs. simulated human.

2024). Performance of Qwen2-VL on TOMATO suggests that explicitly incorporating temporal-aware positional encoding like M-RoPE is likely essential for enhancing MFMs' visual temporal reasoning capabilities. Future improvements in open-source models could benefit from adopting similar strategies to close the gap between the open-source and proprietary models.

**Existing models are limited to understanding events that are interpretable in $\leq 8$ frames.** We assess four MFMs' performance across different number of frames on TOMATO, as illustrated in Figure 2 (a). The consistent improvement in accuracy of human performance suggests that our benchmark rely on the additional frames to convey more temporal information. Notably, although models exhibit performance increase transitioning from 1 frame to 8 frames, the performance plateaus beyond this point. This suggests that even if models are able to reason the transitions between the frames before 8 frames, they cannot utilize the additional temporal information obtained by the added frames. Therefore, we conclude that the overall performance of the four MFMs remain suboptimal in their visual temporal reasoning capabilities, and there is still room for improvements in MFMs' ability to leverage the additional information on the frame transitions introduced by added frames.

**Models perform better on simulated human than real human scenarios.** To investigate the extent to which a cleaner, more abstract representation of video content (Figure 2 (c)) influences models' temporal reasoning abilities, we evaluate two leading MFMs. Specifically, this evaluation contrasts real human scenarios and their corresponding simulated counterparts, across five reasoning types (*i.e., action count*, *direction*, *rotation*, *shape & trend*, and *velocity & frequency*), covering the same 101 QA pairs in both scenarios. As a result, GPT-4o marks a noticeable improvement of 21.9% from real human to simulated human scenarios, underscoring the potential of enhancing models' temporal reasoning capabilities in video understanding through semantic video abstraction. Conversely, Qwen2-VL-72B displays a modest increase of 7.8% transitioning from real human to simulated human. While it marginally outperforms GPT-4o in the overall evaluation on TOMATO, its visual temporal reasoning capabilities in simulated scenarios still show room for improvement (Figure 2 (b)). Future work can target further enhancing models' temporal reasoning abilities for real human videos by exploring their generalization capabilities through leveraging automatically generated simulated 3D human motion data (Guo et al., 2022).

## 7 CONCLUSION

Existing benchmarks likely overestimate the true visual temporal reasoning capabilities of MFMs. In response, we establish three key principles and corresponding metrics to systematically examine visual temporal reasoning tasks. Building upon these principles, we introduce TOMATO, a novel video understanding benchmark to rigorously assess MFMs' true visual temporal reasoning capabilities. Besides revealing a previously underestimated human-model performance gap, our comprehensive evaluation highlights a critical limitation: MFMs fail to interpret videos as continuous sequences, instead resorting to understanding isolated frames, which severely undermines their visual temporal reasoning capabilities. This work sheds the light for developing AI systems capable of comprehending changing scenes in real life through the video modality.

ACKNOWLEDGMENTS

We are grateful for compute credits provided by Google through the TRC program. We thank Professor Alex Wong, Professor Rex Ying, and Professor Dylan McKay for their valuable feedback. We also appreciate the insightful discussions and support from members of the Yale NLP Lab, Yale Inquisitive Robotics Lab, Yale Graph and Geometric Learning Lab, and our peers in the Yale two-year MSCS program.

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

CONTENTS

# A  TASKS EXAMPLES OF TOMATO

For Music-AVQA and CLEVRER, we re-annotated QA pairs to emphasize temporal aspects, such as *"which musical instrument plays first"* for Music-AVQA and *"how many collisions are there in the video"* for CLEVRER. For TGIF-QA and Perception Test, we retained their existing questions but generate additional numerical answer options close to the groundtruth.

Table 6: Task examples of TOMATO. Some videos are collected from existing video datasets, including Music-AVQA (Li et al., 2022), CLEVRER (Yi et al., 2020), TGIF-QA (Jang et al., 2017), and Perecption Test (Pătrăucean et al., 2023). [†]: Tasks requiring re-annotation (§4.3).

| Temporal Tasks | Video Sources | Examples |
|---|---|---|
| **Rotation** (19.3%) | YouTube & Self-created | *In which direction(s) does the object rotate?* (A) Clockwise (B) Counter-clockwise (C) Clockwise then counter-clockwise (D) Counter-clockwise then clockwise (E) No rotation |
| **Direction** (27.2%) | YouTube & Self-created | *In which direction(s) does the person's hand move?* (A) Left (B) Right (C) First to the left then to the right (D) First to the right then to the left (E) No movements |
| **Velocity & Frequency** (14.2%) | YouTube & Self-created | *What is the speed pattern of the train?* (A) Accelerating (B) Decelerating (C) Constant Speed (D) No movement |
| **Shape & Trend** (15.0%) | YouTube & Self-created | *What is the shape of the object that the person draws in the air?* (A) Circle (B) Triangle (C) Square/rectangle (D) Trapezoid (E) Diamond (F) Not drawing at all |
| **Visual Cues** (4.7%) | *Music-AVQA* [†] | *Which musical instrument plays first?* (A) Accordion (B) Saxophone (C) Both instruments play simultaneously (D) Neither instrument produces any sound |
| **Action Count** (19.7%) | *CLEVRER* [†] | *How many collision(s) are there in the video?* (A) 1 (B) 2 (C) 3 (D) 4 (E) 5 (F) 6 |
| | *TGIF-QA* | *How many times does the cat lick the water tap?* (A) 1 (B) 2 (C) 3 (D) 4 (E) 5 |
| | *Perception Test* | *How many times does the person launch the object on the slanted plane?* (A) 1 (B) 2 (C) 3 (D) 4 (E) 5 (F) 6 |
| | Self-created | *How many trapezoid(s) does the person draw in the air?* (A) 1 (B) 2 (C) 3 (D) 4 (E) 5 (F) 6 |

# B   SCENARIO ANALYSIS

Table 7: MFMs performance on Counterfactual, Zoomed-in, and First-Person Perspective QAs. "True" and "False" indicate whether the questions meet the criteria specified in the merged cell above the respective column (*e.g.,* Counterfactual, Zoomed-In, and First Person Perspective).

| | Counterfactual | | | Zoomed-In | | | First-Person Perspective | | |
|---|---|---|---|---|---|---|---|---|---|
| | False | True | $\triangle\%$ | False | True | $\triangle\%$ | False | True | $\triangle\%$ |
| GPT-4o | 38.5 | 24.0 | -37.7 | 37.8 | 40.1 | +6.0 | 38.8 | 47.5 | +22.5 |
| Qwen2-VL-72B | 45.3 | 32.0 | -29.4 | 31.6 | 30.3 | -4.2 | 30.4 | 61.3 | +101.6 |
| Qwen2-VL-7B | 37.1 | 33.0 | -11.0 | 24.9 | 25.4 | +1.9 | 24.7 | 42.5 | +72.1 |
| Video-CCAM-v1.1 14B | 37.5 | 29.0 | -22.6 | 24.7 | 24.4 | -1.0 | 24.1 | 35.0 | +45.2 |

In this section, we present our analysis across various video scenarios. While the videos are not explicitly designed to rigorously validate the findings, we believe these insights are valuable for highlighting general trends observed in the experimental results for future studies.

**More capable models are more reliant on common sense.**   In curating the counterfactual QAs, we employ video editing techniques (*e.g.,* reversing, rotating, cropping) to produce contents that are impossible to observe in real life. As detailed in Table 7, although all four models demonstrate similar performance on non-counterfactual QAs, the shift to counterfactual examples reveals a significant performance drop, particularly for the best general-purpose model, GPT-4o, with a decrease of 37.7%, and the leading open-source model, Qwen2-VL-72B, with a 29.4% decrease. These results suggest that more capable models are more prone to exploit shortcuts within the task's background information and heavily rely on pre-trained knowledge, rather than truthfully understanding the video content, even when explicitly instructed to not rely on commonsense reasoning in solving the tasks.

**Zoomed-in views offer limited performance improvement in challenging human scenarios.** Although performance on standard views varies across different models (Table 7), providing manual zoomed-in views yields only modest performance gains: 6.0% for GPT-4o and 1.9% for Qwen2-VL-7B. Surprisingly, the zoomed-in views even worsen the performance of Video-CCAM-v1.1 14B and Qwen2-VL-72B by 1.0% and 4.2%, respectively. The limited aid provided by the zoomed-in view indicates that the challenges inherent to the tasks can not be tackled through zooming-in, but require deeper temporal understanding of the videos beyond enhancements in visual focus.

**Models excel in first-person over third-person perspective temporal reasoning video understanding.**   In comparing 80 first-person perspective QAs to a larger set of 668 third-person perspective QAs, our goal is to explore the general trends in model performances based on the presence or absence of a main subject in the video. The results suggest that the absence of a main subject does not hinder model performance. In fact, we observe significantly better model performance on the first-person perspective tasks (Table 7). Notably, the Qwen2-VL models achieve remarkable performance gains of 101.6% and 72.1%, respectively, with Qwen2-VL-72B scoring 61.3—outperforming GPT-4o by 28.9% under the same conditions. These findings also highlight the potential of open-source models to surpass more capable general-purpose proprietary models on temporal reasoning video understanding tasks.

## C    Experiment Setup

### C.1    Model Configuration

Table 8: Model configurations for evaluation. Unset values indicate that their default values are being used. ‡: Models that can process video directly. *: Models that can only process eight frames. †: Set `token_kept_ratio` to 0.61. Configurations are based on official model repositories where available. **Temp.**: temperature.

| Model | API Checkpoint / HF Checkpoint | Do Sample | Max New Tokens | Temp. | Top-P | Seed |
|---|---|---|---|---|---|---|
| *Proprietary Multimodal Foundation Models (MFMs)* | | | | | | |
| GPT-4o-mini | gpt-4o-mini-2024-07-18 | | 1024 | 0 | 1 | 215 |
| GPT-4o | gpt-4o-2024-08-06 | | 1024 | 0 | 1 | 215 |
| Claude 3 Opus | claude-3-opus-20240229 | | 1024 | 0 | 1 | |
| Claude 3.5 Sonnet | claude-3-5-sonnet-20240620 | | 1024 | 0 | 1 | |
| Claude 3 Haiku | claude-3-haiku-20240307 | | 1024 | 0 | 1 | |
| Gemini 1.5 Flash | gemini-1.5-flash-001 | | 1024 | 0 | 1 | |
| Gemini 1.5 Pro | gemini-1.5-pro-001 | | 1024 | 0 | 1 | |
| Gemini 2.0 Flash Exp | gemini-2.0-flash-exp | | 1024 | 0 | 1 | |
| Reka Flash | reka-flash-20240226 | | 1024 | 0 | 1 | |
| Reka Core | reka-core-20240501 | | 1024 | 0 | 1 | |
| *Open-Source Multimodal Foundation Models (MFMs)* | | | | | | |
| InternVL 2 8B | OpenGVLab/InternVL2-8B | False | 1024 | | | |
| InternVL 2 26B | OpenGVLab/InternVL2-26B | False | 1024 | | | |
| InternVL 2 40B | OpenGVLab/InternVL2-40B | False | 1024 | | | |
| InternVideo 2 Chat 8B | OpenGVLab/InternVideo2-Chat-8B | False | 1024 | | | |
| Video-CCAM-v1.1 4B | JaronTHU/Video-CCAM-4B-v1.1 | False | 1024 | | | |
| Video-CCAM-v1.1 9B | JaronTHU/Video-CCAM-9B-v1.1 | False | 1024 | | | |
| Video-CCAM-v1.1 14B | JaronTHU/Video-CCAM-14B-v1.1 | False | 1024 | | | |
| LLaVA NeXT Video 32B | lmms-lab/LLaVA-NeXT-Video-32B-Qwen | False | 1024 | | | |
| LLaVA OneVision 7B | lmms-lab/llava-onevision-qwen2-7b-ov | False | 1024 | | | |
| Video LLaVA 7B * | LanguageBind/Video-LLaVA-7B | False | 1024 | | | |
| VILA 13B | Efficient-Large-Model/VILA-13b | False | 1024 | | | |
| Phi 3.5 Vision Instruct | microsoft/Phi-3.5-vision-instruct | False | 1024 | | | |
| Qwen2-VL-7B | Qwen/Qwen2-VL-7B-Instruct | False | 1024 | | | |
| Qwen2-VL-72B | Qwen/Qwen2-VL-72B-Instruct-AWQ | False | 1024 | | | |
| VideoLLaMA2 7B ‡ | DAMO-NLP-SG/VideoLLaMA2-7B | False | 1024 | | | |
| VideoLLaMA2 72B ‡ | DAMO-NLP-SG/VideoLLaMA2-72B | False | 1024 | | | |
| LLaVA-Video-72B | lmms-lab/LLaVA-Video-72B-Qwen2 | False | 1024 | | | |
| LLaVA-Video-7B | lmms-lab/LLaVA-Video-7B-Qwen2 | False | 1024 | | | |
| LLaVA-Video-7B-Video-Only | lmms-lab/LLaVA-Video-7B-Qwen2-Video-Only | False | 1024 | | | |
| AuroraCap-7B-VID † | wchai/AuroraCap-7B-IMG-xtuner | False | 1024 | | | |

### C.2    Implementation Details for Model Inference

All MFMs are evaluated using a zero-shot strategy across all benchmarks, including TOMATO, to ensure fair comparison. Whenever possible, we use the provided code from each model's official page for video preprocessing. Notably, VideoLLaMA 2 7B and VideoLLaMA 2 72B (Cheng et al., 2024) processes video content directly without splitting it into frames, so we input the videos without modification. Gemini 1.5 Flash (Gemini, 2024) also processes video content directly with 1 FPS. As noted by Tam et al. (2024), restricting the output format of MFMs can hinder performance; therefore, we do not impose any output restrictions. However, this approach can complicate answer parsing using simple heuristics. To address this issue, we employ `GPT-4o-mini` to extract multiple-choice answers from each model's response when our standard parser fails, where prompts are provided in §D. We use NVIDIA A100 GPUs for all non-API-based evaluation.

# D  PROMPTS

## D.1  EVALUATION PROMPT

```
You will be provided with {num_frames} separate frames uniformly sampled
    from a video, the frames are provided in chronological order of the
    video. Analyze these frames and provide the answer to the question
    about the video content. Answer the multiple-choice question about
    the video content.

You must use these frames to answer the multiple-choice question; do not
    rely on any external knowledge or commonsense.

<question> {question} </question>

<options> {index2ans} </options>

Even if the information in these separate frames is not enough to answer
    the question, PLEASE TRY YOUR BEST TO GUESS AN ANSWER WHICH YOU THINK
     WOULD BE THE MOST POSSIBLE ONE BASED ON THE QUESTION.

DO NOT GENERATE ANSWER SUCH AS 'NOT POSSIBLE TO DETERMINE.'
```

## D.2  ANSWER EXTRACTING PROMPT

```
You are given a response, a list of multiple-choice options, and a
    index2answer mapping. You are required to extract the letter option
    from the GPT.

<response> {response} </response>

<all_choices> {all_choices} </all_choices>

<index2answer> {index2ans} </index2answer>

Only output the single parsed letter from the response. No other texts
    are needed.

If you think no options can match the index2answer dictionary, randomly
    select one letter.

Your extracted letter is:
```

## D.3 VIDEO-FREE RANDOM GUESS PROMPT

```
Randomly guess a reasonable answer based on the question only.

<question> {question} </question>

<options> {index2ans} </options>

DO NOT GENERATE ANSWER SUCH AS 'NOT POSSIBLE TO DETERMINE.'
```

# E  DATA ANALYSIS

## E.1  STATISTICS

Table 9: TOMATO main statistics.

| Statistics | Value |
|---|---|
| Total Questions | 1,484 |
| Total Videos | 1,417 |
| *Demonstration Type* | |
|     Human | 588 (41.4%) |
|     Object | 596 (42.1%) |
|     Simulated | 233 (16.4%) |
| *Source* | |
|     Self-recorded and -generated | 805 (56.8%) |
|     YouTube | 398 (28.1%) |
|     Existing Video Datasets | 214 (15.1%) |
| Unique Source Videos | 683 |
| Duration (Seconds, avg/max) | 9.21 / 72.74 |
| Resolution (avg/max) | $1332 \times 1076$ / $1080 \times 1920$ |
| Number of Reasoning Tasks | 6 |
| Number of Demonstration Categories | 3 |
| Question Length (avg/max) | 11.71 / 22 |
| Single Choice Length (avg/max) | 3.69 / 10 |
| Choices per Question (avg/max) | 5.19 / 7 |

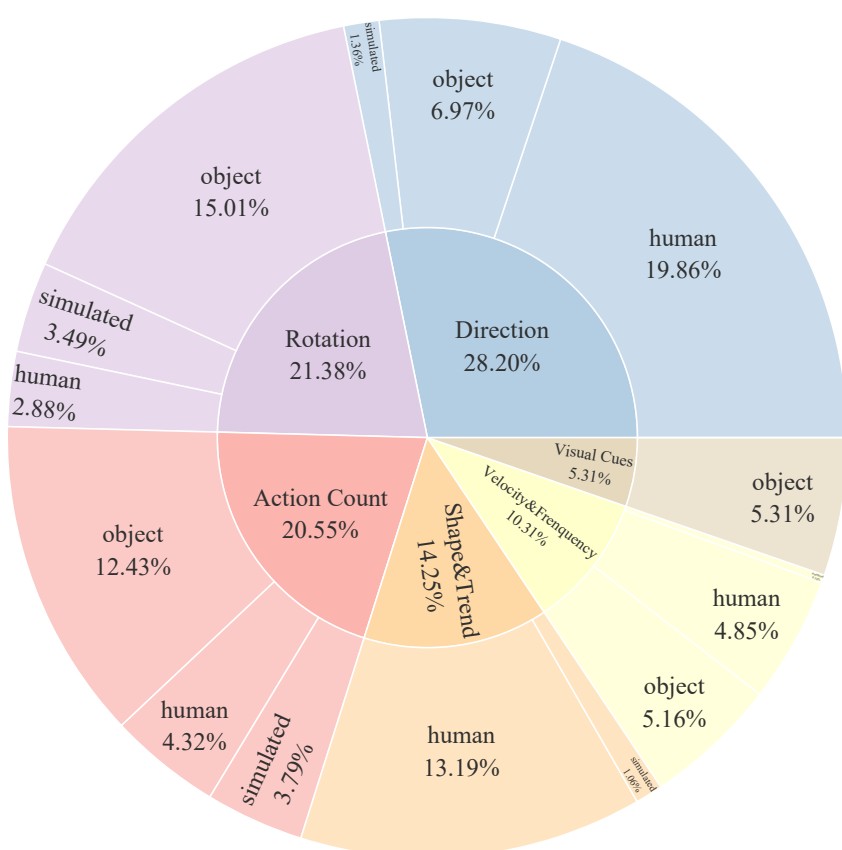

Figure 3: Video source distribution of TOMATO.

## E.3 Performance Comparisons across Different Number of Frames

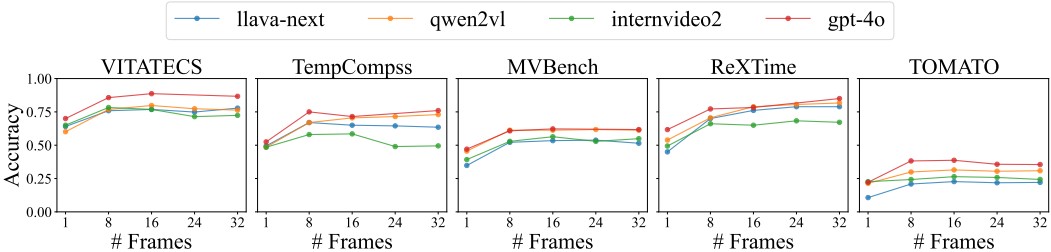

Figure 4: Performance comparisons of VITATECS (Li et al., 2023), TempCompass (Liu et al., 2024), MVBench (Li et al., 2024b), RexTime (Chen et al., 2024a), and TOMATO across different number of frames.

## E.4 Model Performance with ≤ 8 Frames

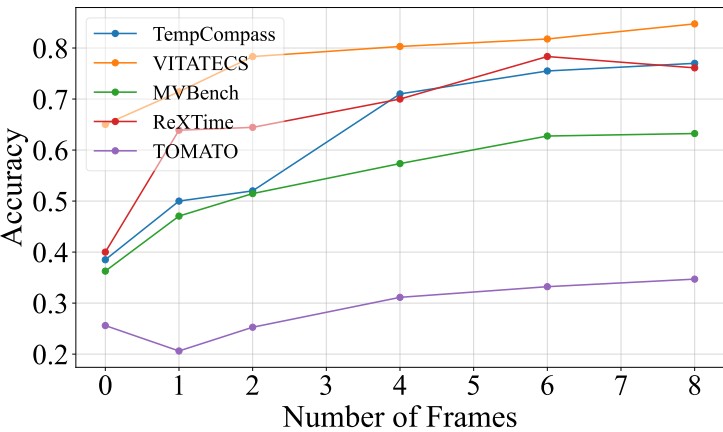

Figure 5: Performance of Qwen2-VL-72B across five benchmarks with input restricted to fewer than 8 frames.

## E.5 Video Duration Distribution

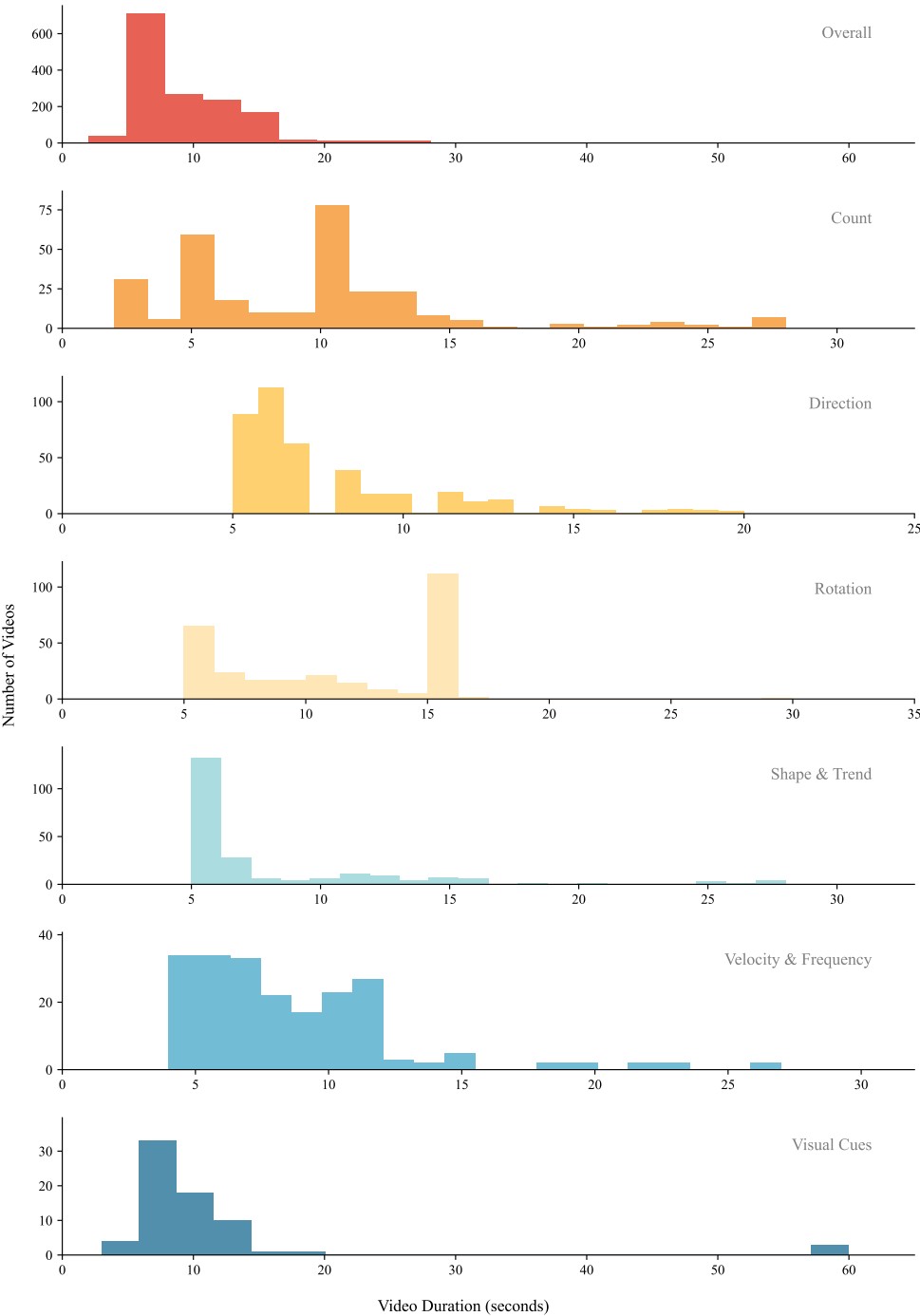

Figure 6: Video duration distribution of TOMATO

## E.6 RESPONSE LENGTH DISTRIBUTION

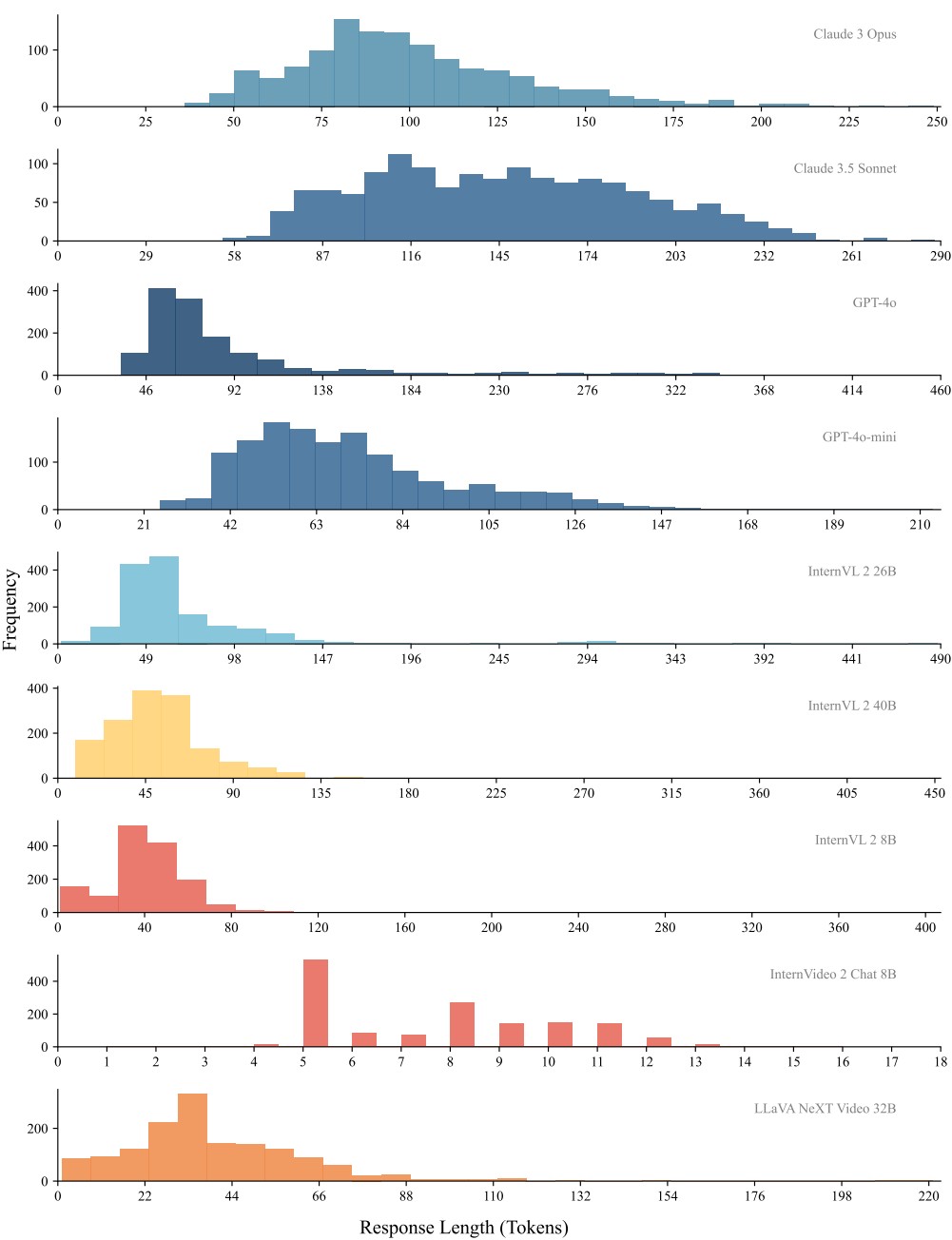

Figure 7: Response length distribution, part 1

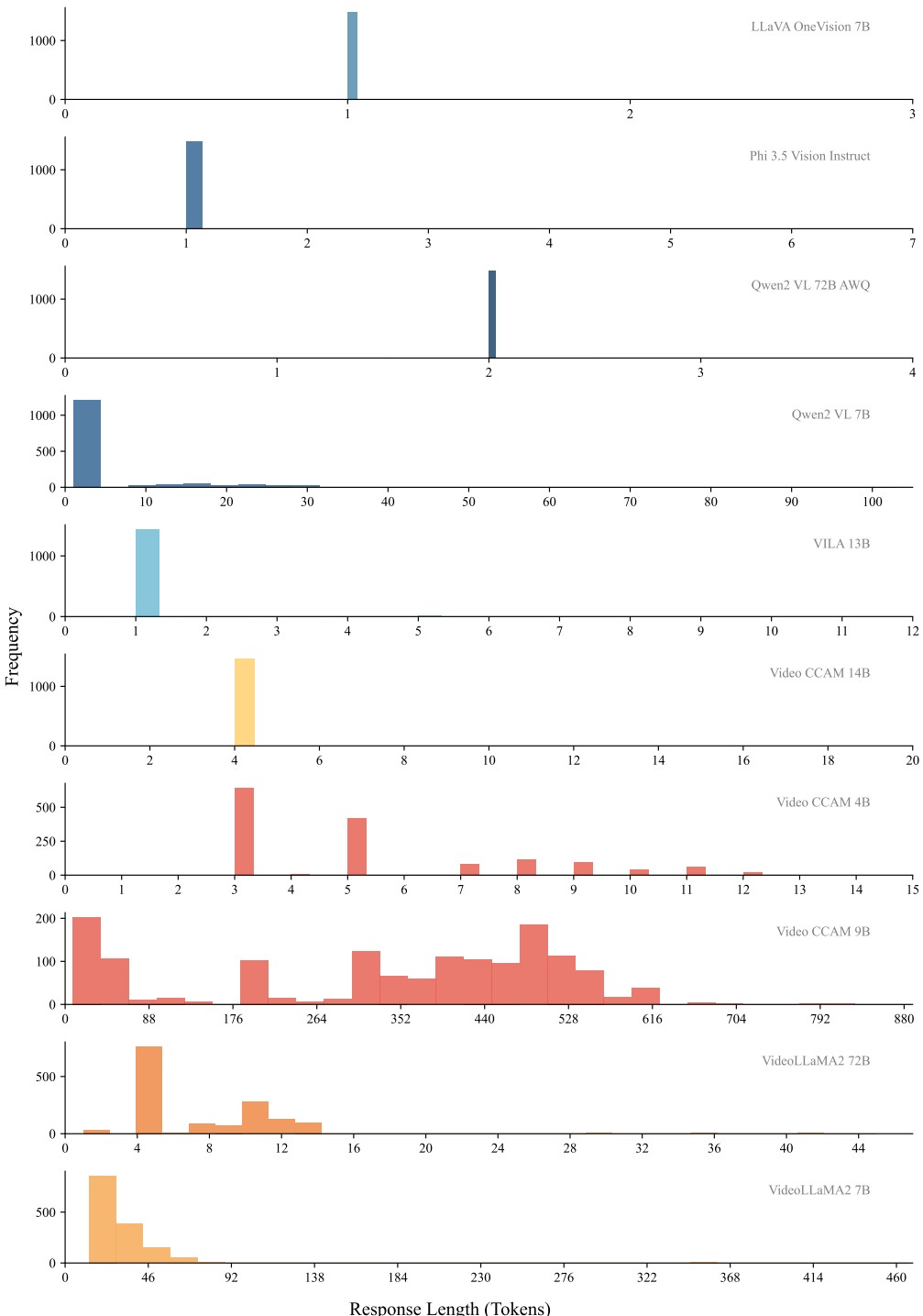

Figure 8: Response length distribution, part 2

# F  LICENSE INFORMATION.

We, the authors, bear all responsibility in case of violation of rights. TOMATO is intended only used for academic research purposes. Commercial use of TOMATO in any form is strictly prohibited. The videos included in TOMATO are subject to the following licensing conditions:

**YouTube videos.**    All videos sourced from YouTube[3] are licensed under Creative Commons[4]. The original YouTube video links are provided in our dataset, and proper attribution is given in accordance with the license terms. We do not hold the copyright to any of the YouTube videos; we ask that you respect the rights of the original video creators. If you believe that any content in TOMATO infringes on your rights, please contact the authors immediately, and we will remove the video accordingly.

**Existing video datasets.**    CLEVRER (Yi et al., 2020) is licensed under CC0[5]. TGIF-QA Jang et al. (2017) is built upon TGIF (Li et al., 2016), which is available for non-commercial only[6]. Perception Test (Pătrăucean et al., 2023) is licensed under Creative Commons Attribution 4.0 International License (CC-BY)[7]. Music-AVQA (Li et al., 2022) is licensed under Creative Commons Attribution-NonCommercial 4.0 International (CC BY-NC 4.0)[8].

**Self-recorded and -generated benchmark-specific videos.**    Videos created specifically for TOMATO are licensed under CC BY-NC-SA[9]. These videos are intended to be freely used for academic purposes, with proper attribution required. We kindly ask that all individuals appearing in our self-recorded videos are respected.

---

[3]https://www.youtube.com/

[4]https://creativecommons.org/

[5]https://creativecommons.org/public-domain/cc0/

[6]https://github.com/raingo/TGIF-Release/blob/master/LICENSE

[7]https://creativecommons.org/licenses/by/4.0/legalcode

[8]https://creativecommons.org/licenses/by-nc/4.0/

[9]https://creativecommons.org/licenses/by-nc-sa/4.0/deed.en

## G  ANNOTATION DETAILS

### G.1  ANNOTATOR BIOGRAPHIES

Table 10: Biographies for human annotators in TOMATO.

| ID | Year | Field of Study |
|----|------|----------------|
| 1 | Second-Year Master's Student | Mathematics |
| 2 | Second-Year Master's Student | Computer Science & Mathematics |
| 3 | Recent Master's Graduate | Mechanical Engineering |
| 4 | Second-Year Ph.D. Student | Computer Science |

### G.2  PILOT ANNOTATION

To determine the visual temporal reasoning types in TOMATO, we conduct a pilot study aimed at exploring and refining task definitions. Each annotator is tasked with brainstorming potential task types relevant to visual temporal reasoning. Once a new reasoning type is proposed, each annotator is assigned to compose 20 question-answer (QA) pairs of this reasoning type. To ensure that the QA pairs of this reasoning type adhere to the principles outlined in §3, we utilize GPT-4o to verify each metrics score. As a result of this pilot process, we identify and finalize *six* distinct visual temporal reasoning tasks: *Rotation*, *Direction*, *Velocity & Frequency*, *Shape & Trend*, *Visual Cues*, and *Action Count*. These task types form the foundation for the full-scale annotation of our benchmark, ensuring comprehensive coverage of visual temporal reasoning.

### G.3  FULL-SCALE ANNOTATION

**Initial annotation.**    For YouTube videos, annotators are tasked with selecting videos that corresponded to a specific reasoning type (as outlined in §4.1). Given the targeted reasoning type, annotators carefully search for videos from YouTube and compose questions that exemplified the temporal reasoning patterns (Table 6). For *TGIF-QA* (Jang et al., 2017), annotators are instructed to select video-question pairs that they could easily answer. This is necessary because many questions in *TGIF-QA* are challenging for humans to answer due to the low resolution or low frame rate of the videos. For *CLEVRER* Yi et al. (2020), annotators are responsible for writing the question (*i.e.,* "How many distinct collision(s) occur throughout the entire video?"), and we extract answers from the meta data of each video. For *Music-AVQA* (Li et al., 2022), annotators are tasked with curating videos that exemplified the *Visual Cues* reasoning types and composing corresponding question-answer pairs. For self-recorded and -generated videos, both annotators and authors are responsible for recording and generating videos on specific reasoning types, as well as composing corresponding question-answer pairs.

**Cross-checking and verification.**    Each annotator reviews the work of another annotator by carefully re-annotating the answer, without being provided with the original annotator's response. At the end of the process, the two sets of annotations are compared and checked for consistency. If discrepancies arise between the two annotators' answers, both annotators and the authors engage in a collective resolution to reconcile the differences. During this stage, out of 1,484 questions-answer pairs, 44 questions require collective resolution during this verification phase.

**Collective resolution.**    In cases where the annotators do not reach consensus during the cross-checking phase, a discussion is held to resolve the disagreement. This phase enables both authors and annotators to determine whether the discrepancy stems from a lack of careful annotation or from poor video quality. Out of the 44 mismatched answers, we replace four videos and correct the answers for the remaining cases.

# H   DATA ANNOTATION PLATFORM

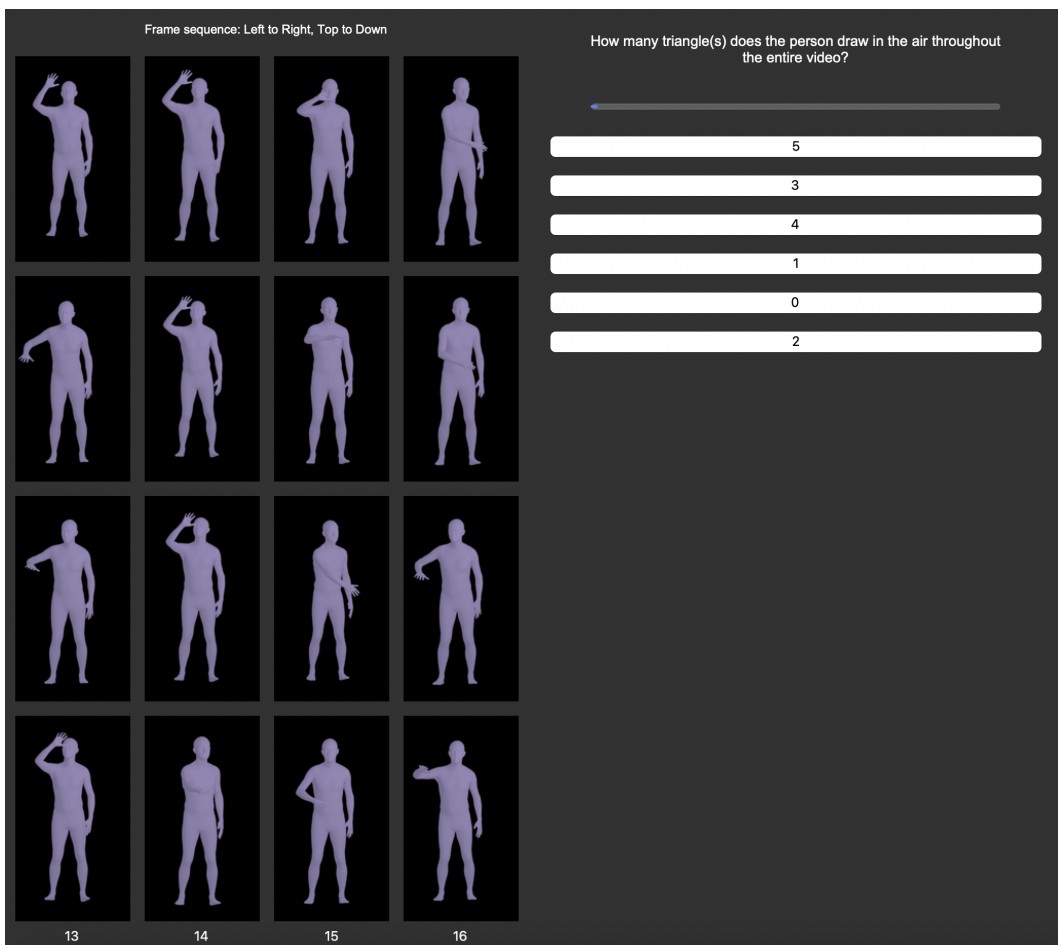

Figure 9: Screenshot of our human evaluation platform. As shown, the annotator is presented with a question (top right) alongside the corresponding video, displayed in a grid of 16 frames ($4 \times 4$). The annotator is then required to select the correct answer from the provided options.

# I  Common Failure Cases of Rotation

## I.1  Human

### I.1.1  Example 1



**rotation 8**

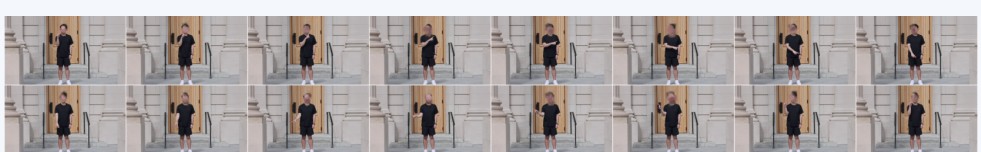

Which direction(s) does the person's hand rotate in?

A. Counter-clockwise throughout.
B. No rotation.
C. Clockwise throughout.
D. Counter-clockwise then clockwise.
E. Clockwise then counter-clockwise.

Answer: C. Clockwise throughout.

</div>

**GPT-4o:** Based on the provided frames, the person's hand appears to be rotating in a circular motion. Initially, the hand is pointing upwards, then it moves to the right, downwards, and finally to the left before returning to the starting position. This suggests a counter-clockwise rotation.

Therefore, the most likely answer is:

**Option A: Counter-clockwise throughout.**

**Claude 3.5 Sonnet:** Based on analyzing the sequence of images provided, the most likely answer to the question "Which direction(s) does the person's hand rotate in?" is:

E: Clockwise then counter-clockwise.

The images show the person's hand and arm movement in a series of frames. At the beginning, the hand is raised with the index finger pointing upward. As the sequence progresses, the hand appears to rotate clockwise, moving down and to the side. Then in later frames, the hand motion reverses, rotating counter-clockwise back up to the initial raised position with the finger pointing up.

While it's difficult to be certain of the full motion from still images alone, the clockwise followed by counter-clockwise rotation seems to best match the arm and hand positions shown across the frame sequence provided.

**Qwen2 VL 72B instruct AWQ:** E

**Qwen2 VL 7B Instruct:** D

**Video-CCAM-v1.1 14B:** Answer is A.

**InternVL 2 40B:** Based on the frames provided, the person's hand appears to rotate in a counter-clockwise direction throughout the video.

Answer: A

## I.2  OBJECT

### I.2.1  EXAMPLE 1

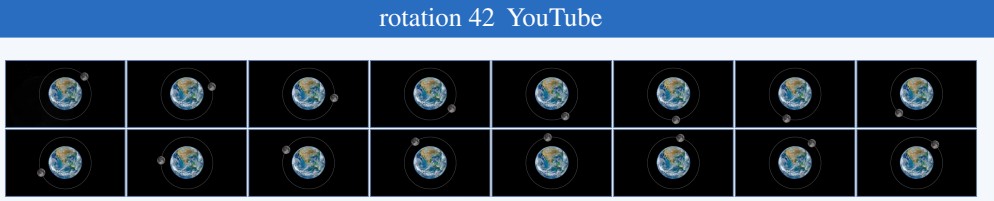

> ### rotation 42  YouTube
>
> https://www.youtube.com/watch?v=MrzWzMUhQfw
>
> What direction(s) does the moon rotate relative to the Earth?
>
> A. Counter-clockwise throughout.
> B. No rotation.
> C. Clockwise throughout.
> D. Clockwise then counter-clockwise.
> E. Counter-clockwise then clockwise.
>
> Answer: C. Clockwise throughout.

**GPT-4o:** Based on the provided frames, the moon appears to be rotating around the Earth in a counter-clockwise direction. This is evident from the sequential positions of the moon as it moves from the right side of the Earth, downwards, to the left side, and then upwards, completing a counter-clockwise orbit.

Therefore, the most possible answer based on the frames is:

**'A': Counter-clockwise throughout.**

**Claude 3.5 Sonnet:** Based on analyzing the sequence of 16 frames provided, the answer to the question "What direction(s) does the moon rotate relative to the Earth?" appears to be:

A. Counter-clockwise throughout.

The images show the moon orbiting around the Earth in a counter-clockwise direction when viewed from above the North Pole. The moon starts on the right side of the Earth and moves counter-clockwise around the circular orbit, completing a full revolution. Throughout the entire sequence, the moon maintains this counter-clockwise motion relative to the Earth.

**Qwen2 VL 72B instruct AWQ:** A

**Qwen2 VL 7B Instruct:** A

**Video-CCAM-v1.1 14B:** Answer is A.

**InternVL 2 40B:** Based on the provided frames, the moon appears to rotate in a counter-clockwise direction relative to the Earth throughout the video.

Therefore, the most possible answer is:

**A: Counter-clockwise throughout.**

I.2.2   EXAMPLE 2

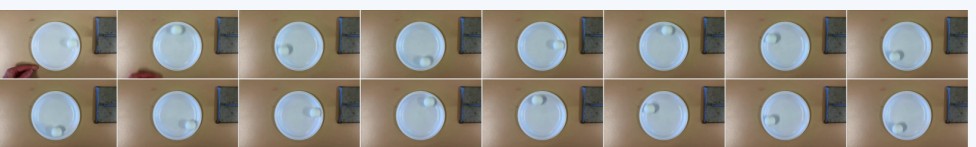

https://www.youtube.com/watch?v=ApO6V_oetVI

What direction(s) does the Ping Pong ball rotate in?

A. Clockwise throughout.
B. No rotation.
C. Clockwise then counter-clockwise.
D. Counter-clockwise throughout.
E. Counter-clockwise then clockwise.

Answer: D. Counter-clockwise throughout.

**GPT-4o:** Based on the provided frames, the ping pong ball appears to be rotating in a consistent direction. The ball moves around the plate in a circular motion, which suggests a rotational movement.

From the sequence of frames, it seems that the ball is rotating in a clockwise direction throughout the video. There is no indication of a change in the direction of rotation.

Therefore, the most likely answer is:

**Option A: Clockwise throughout.**

**Claude 3.5 Sonnet:** Based on analyzing the sequence of frames provided, the most likely answer to the question "What direction(s) does the Ping Pong ball rotate in?" is:

C: Clockwise then counter-clockwise.

The reasoning for this conclusion is:

1. In the initial frames, the ball appears to be rotating clockwise as it moves around the edge of the plate in a counter-clockwise direction.

2. As the ball approaches the bottom of the plate, its rotation seems to slow down and then reverse.

3. In the later frames, the ball appears to be rotating counter-clockwise as it continues its circular path around the plate.

This change in rotational direction is consistent with the physics of a ball rolling along a curved surface, where friction and the changing angle of contact can cause the ball to reverse its spin. The clockwise rotation followed by counter-clockwise rotation best matches the observed behavior of the ball in the provided image sequence.

**Qwen2 VL 72B instruct AWQ:** A

**Qwen2 VL 7B Instruct:** C

**Video-CCAM-v1.1 14B:** Answer is C.

**InternVL 2 40B:** Based on the provided frames, the Ping Pong ball appears to rotate in a clockwise direction throughout the video.

Answer: A

I.2.3 EXAMPLE 3

## rotation 57  YouTube

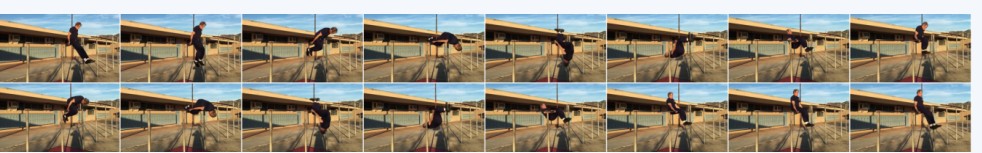

https://www.youtube.com/watch?v=zji3Cm_k7go

In which direction(s) does the person rotate on the horizontal bar?

A. Clockwise then counter-clockwise. B. Clockwise throughout.
C. Counter-clockwise throughout.    D. No rotation.
E. Counter-clockwise then clockwise.

Answer: B. Clockwise throughout.

---

**GPT-4o:** Based on the provided frames, the person appears to be rotating in a counter-clockwise direction throughout the sequence. The frames show the person starting from a seated position on the bar, moving through an inverted position, and then returning to the seated position, all in a counter-clockwise motion.

Therefore, the most likely answer is:

**'C': 'Counter-clockwise throughout.'**

---

**Claude 3.5 Sonnet:** Based on the sequence of images provided, the answer to the question "In which direction(s) does the person rotate on the horizontal bar?" appears to be:

B: Clockwise throughout.

The images show the person starting in an upright seated position on the horizontal bar, then rotating forward (clockwise) through various positions including hanging upside down, continuing the rotation until returning to the starting position. The rotation seems to be consistently clockwise throughout the entire sequence of frames, without any counter-clockwise movement visible.

---

**Qwen2 VL 72B instruct AWQ:** A

---

**Qwen2 VL 7B Instruct:** A

---

**Video-CCAM-v1.1 14B:** Answer is C.

---

**InternVL 2 40B:** Based on the frames provided, the person on the horizontal bar appears to rotate in a counter-clockwise direction throughout the video.

Therefore, the most possible answer is:

'C': Counter-clockwise throughout.

I.2.4 EXAMPLE 4

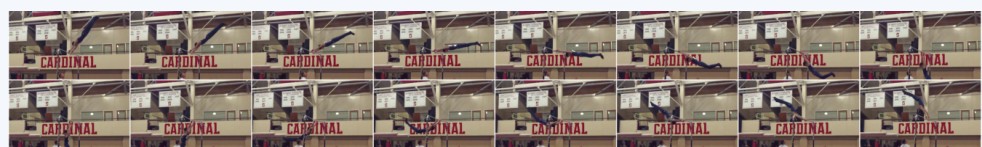

https://www.youtube.com/watch?v=x1uOtX8mYJo

```
In which direction(s) does the gymnastic athlete rotate on the horizontal
    bar?

A. Clockwise throughout.
B. Counter-clockwise then clockwise.
C. Clockwise then counter-clockwise.
D. Counter-clockwise throughout.
E. No rotation.

Answer: A. Clockwise throughout.
```

**GPT-4o:** Based on the provided frames, the gymnastic athlete appears to be rotating in a counter-clockwise direction throughout the sequence. The athlete starts from a position above the bar and moves downward and backward, indicating a counter-clockwise rotation.

Therefore, the most likely answer is:

**'D': 'Counter-clockwise throughout.'**

**Claude 3.5 Sonnet:** Based on the sequence of images provided, the gymnast appears to be rotating counter-clockwise throughout the routine on the horizontal bar. The athlete starts in a position with their body extended away from the bar, then swings under and around the bar in a counter-clockwise direction. This counter-clockwise rotation continues through subsequent swings and movements shown in the frames.

Therefore, the most likely answer to the question "In which direction(s) does the gymnastic athlete rotate on the horizontal bar?" is:

D: Counter-clockwise throughout.

The images consistently show the gymnast moving in a counter-clockwise direction around the horizontal bar, from the initial swing through to the later parts of the routine captured in the frames.

**Qwen2 VL 72B instruct AWQ:** D

**Qwen2 VL 7B Instruct:** C

**Video-CCAM-v1.1 14B:** Answer is A.

**InternVL 2 40B:** Based on the frames provided, the gymnastic athlete appears to rotate in a counter-clockwise direction throughout the routine on the horizontal bar.

Therefore, the most possible answer is:

'D': Counter-clockwise throughout.

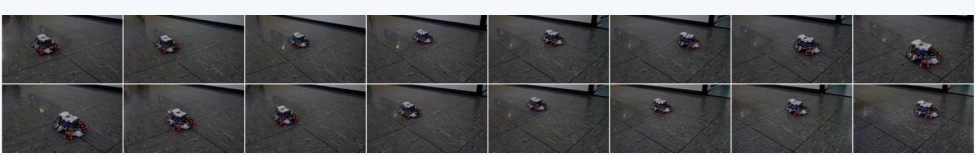

rotation 144  YouTube

```
                            https:
   //www.youtube.com/results?search_query=omni+wheel&sp=EgIwAQ%253D%253D

   In which direction(s) does the car move?

   A. No rotation.
   B. Counter-clockwise then clockwise.
   C. Counter-clockwise throughout.
   D. Clockwise throughout.
   E. Clockwise then counter-clockwise.

   Answer: D. Clockwise throughout.
```

**GPT-4o:** Based on the provided frames, the car appears to be rotating. The direction of rotation can be inferred from the position and orientation of the car in each frame.

From the first frame to the last frame, the car seems to be rotating in a counter-clockwise direction. There is no indication that the car changes its direction of rotation.

Therefore, the most likely answer is:

**'C': 'Counter-clockwise throughout.'**

**Claude 3.5 Sonnet:** Based on the series of images provided, I can observe a small robotic device on a tiled floor. The device appears to be rotating slightly between frames, but it's difficult to determine a consistent direction of rotation from these static images alone.

Given the question and options provided, I will make my best guess based on the limited information available:

The most likely answer seems to be C: Counter-clockwise throughout.

This is because in some of the frames, there appears to be a slight counter-clockwise rotation of the device when comparing its orientation to the tile lines on the floor. However, I want to emphasize that this is an educated guess based on limited information, as the rotation is subtle and not definitively clear from these static images.

**Qwen2 VL 72B instruct AWQ:** E

**Qwen2 VL 7B Instruct:** B

**Video-CCAM-v1.1 14B:** Answer is C.

**InternVL 2 40B:** Based on the provided frames, the car appears to move in a counter-clockwise direction throughout.

Answer: C

---

**rotation 64  YouTube**

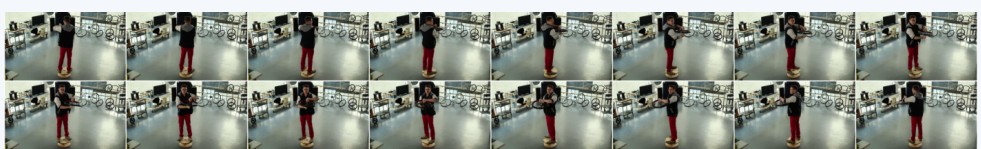

https://www.youtube.com/watch?v=iaauRiRX4do&t=24s

In what direction(s) does the person rotate?

A. No rotation.
B. Clockwise then counter-clockwise.
C. Counter-clockwise then clockwise.
D. Counter-clockwise throughout.
E. Clockwise throughout.

Answer: E. Clockwise throughout.

---

**GPT-4o:** Based on the provided frames, the person appears to be rotating in a counter-clockwise direction initially and then changes direction to rotate clockwise. This can be inferred from the change in the orientation of the person's body and the direction they are facing.

Therefore, the most likely answer is:

**Option C: Counter-clockwise then clockwise.**

---

**Claude 3.5 Sonnet:** Based on analyzing the sequence of images provided, the answer to the question "In what direction(s) does the person rotate?" appears to be:

D: Counter-clockwise throughout.

The images show a person standing on a circular platform and holding a wheel-like object. As the sequence progresses, we can observe that the person's body orientation changes gradually in a counter-clockwise direction. The person starts facing away from the camera and slowly rotates to face different directions, always turning to their left (counter-clockwise) throughout the sequence of frames. There is no evidence of clockwise rotation or change in rotation direction in the provided images.

---

**Qwen2 VL 72B instruct AWQ:** B

---

**Qwen2 VL 7B Instruct:** C

---

**Video-CCAM-v1.1 14B:** Answer is D.

---

**InternVL 2 40B:** Based on the frames provided, the person appears to rotate in a counter-clockwise direction throughout the video.

Answer: D

---

---

**rotation 84  YouTube**

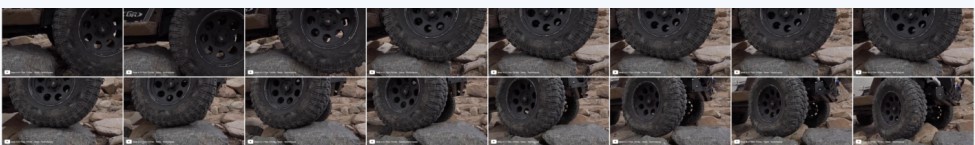

https://www.youtube.com/watch?v=O1Htq-vrm6A

In which direction(s) does the wheel rotate?

A. Clockwise throughout.
B. Counter-clockwise throughout.
C. Counter-clockwise then clockwise.
D. Clockwise then counter-clockwise.
E. No rotation.

Answer: B. Counter-clockwise throughout.

---

**GPT-4o:** Based on the provided frames, the wheel appears to be rotating in a consistent direction as it moves over the rock. The movement of the wheel suggests that it is rotating in a clockwise direction throughout the sequence of frames.

Therefore, the most likely answer is:

**Option A: Clockwise throughout.**

---

**Claude 3.5 Sonnet:** Based on analyzing the sequence of images provided, my best assessment is that the wheel rotates clockwise throughout the video. The images show a Jeep tire on rocky terrain, and while it's difficult to see clear rotation in still frames, the progression suggests the wheel is moving forward over the rocks in a clockwise direction. Therefore, I would select option A: "Clockwise throughout."

The images don't show any obvious change in rotation direction, so the options involving counter-clockwise rotation or changes in direction seem less likely. And the wheel does appear to be moving rather than stationary, ruling out the "No rotation" option.

While there's some uncertainty given the limitations of still images, a clockwise rotation throughout seems the most probable answer based on the visual information provided in the frame sequence.

---

**Qwen2 VL 72B instruct AWQ:** A

---

**Qwen2 VL 7B Instruct:** D

---

**Video-CCAM-v1.1 14B:** Answer is A.

---

**InternVL 2 40B:** Based on the frames provided, it is not possible to determine the direction of the wheel's rotation. The frames show the wheel in different positions, but they do not provide enough information to determine the direction of rotation. Therefore, the most possible answer is:

'E': No rotation.

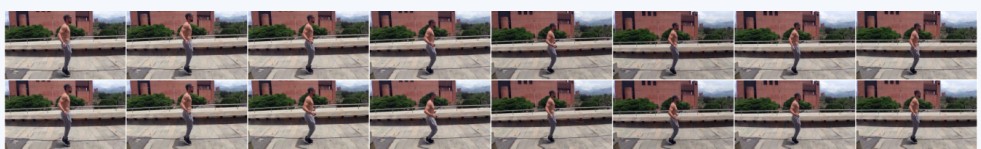

rotation 74  YouTube

https://www.youtube.com/watch?v=FJmRQ5iTXKE

In which direction(s) does the rope rotate?

A. Clockwise then counter-clockwise.
B. Clockwise throughout.
C. Counter-clockwise throughout.
D. Counter-clockwise then clockwise.
E. No rotation.

Answer: C. Counter-clockwise throughout.

**GPT-4o:** Based on the provided frames, the rope appears to be rotating in a consistent direction throughout the sequence. The motion of the rope and the position of the person suggest a continuous rotation.

The most likely answer is:

'B': 'Clockwise throughout.'

**Claude 3.5 Sonnet:** Based on the sequence of images provided, I can see a shirtless man in gray pants and black shoes performing what appears to be a jump rope exercise. The rope itself is not clearly visible in most frames, but the man's arm and body movements are consistent with jump roping.

Regarding the specific question about the direction of rope rotation, it's difficult to determine with certainty from these still images. However, based on the typical mechanics of jump roping and the man's arm positions in the frames, I would make an educated guess that the rope is likely rotating:

C: Counter-clockwise throughout.

This is because in standard jump rope technique, the rope typically rotates forward over the head and under the feet in a counter-clockwise direction when viewed from the jumper's right side, which is the perspective we see in these images. The man's arm positions and slight forward lean are consistent with this standard counter-clockwise rotation.

However, I want to emphasize that this is an inference based on limited information from still images, and without seeing the actual motion of the rope, it's impossible to be 100% certain of the rotation direction.

**Qwen2 VL 72B instruct AWQ:** B

**Qwen2 VL 7B Instruct:** B

**Video-CCAM-v1.1 14B:** Answer is B.

**InternVL 2 40B:** Based on the provided frames, it is not possible to determine the exact direction of the rope's rotation. The frames do not provide a clear view of the rope's movement to confidently answer the question. Therefore, the most possible answer would be:

'E': No rotation.

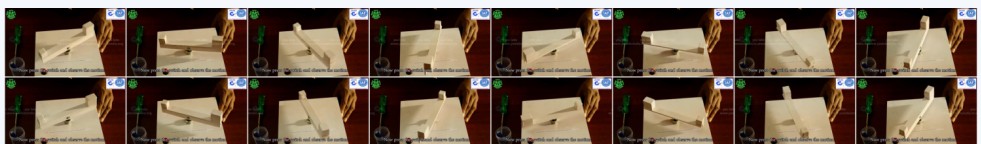

rotation 91 YouTube

https://www.youtube.com/watch?v=mu9jAHgDjQg

In which direction(s) does the object rotate?

A. No rotation.
B. Counter-clockwise throughout.
C. Counter-clockwise then clockwise.
D. Clockwise throughout.
E. Clockwise then counter-clockwise.

Answer: D. Clockwise throughout.

**GPT-4o:** Based on the provided frames, the object appears to be rotating in a consistent direction. The frames show a gradual movement of the object in a circular path.

From the sequence of frames, it can be observed that the object is rotating in a counter-clockwise direction throughout the video.

Therefore, the most likely answer is:

**'B': Counter-clockwise throughout.**

**Claude 3.5 Sonnet:** Based on the sequence of images provided, I can analyze the rotation of the white object on the circular platform. The object appears to be a rod with cube-shaped ends that is mounted on a central pivot point.

Looking at the progression of the images, we can observe that the rod rotates in a counter-clockwise direction initially. It starts from a horizontal position and gradually rotates counter-clockwise through various angles.

However, the sequence of images does not show a complete 360-degree rotation or any reversal in the direction of rotation. The rod is seen rotating counter-clockwise throughout the frames provided.

Given the options and the information available from these frames, the most likely answer to the question "In which direction(s) does the object rotate?" would be:

B: Counter-clockwise throughout.

This answer best matches the observed motion in the sequence of images, where the rod consistently rotates in a counter-clockwise direction without any apparent change in rotational direction.

**Qwen2 VL 72B instruct AWQ:** D

**Qwen2 VL 7B Instruct:** C

**Video-CCAM-v1.1 14B:** Answer is B.

**InternVL 2 40B:** Based on the provided frames, it appears that the object rotates in a counter-clockwise direction throughout the video.

Therefore, the most possible answer is:

'B': Counter-clockwise throughout.

rotation 141  YouTube

https://www.youtube.com/watch?v=fZYfX9VvN8M

In which direction(s) does cube rotate?

A. Clockwise then counter-clockwise. B. Clockwise throughout.
C. Counter-clockwise then clockwise. D. No rotation.
E. Counter-clockwise throughout.

Answer: E. Counter-clockwise throughout.

**GPT-4o:** Based on the provided frames, the cube appears to be rotating in a consistent direction throughout the sequence. The rotation seems to be clockwise when observed from a fixed point.

Therefore, the most likely answer is:

'B': 'Clockwise throughout.'

**Claude 3.5 Sonnet:** Based on analyzing the sequence of 16 frames provided, the cube appears to rotate clockwise throughout the video. The cube starts in one orientation and gradually rotates in a clockwise direction across the frames, without changing direction. Therefore, the most likely answer to the question "In which direction(s) does cube rotate?" is:

B: Clockwise throughout.

The cube maintains a consistent clockwise rotation from the beginning to the end of the frame sequence, without any apparent reversal or change in rotational direction.

**Qwen2 VL 72B instruct AWQ:** B

**Qwen2 VL 7B Instruct:** A

**Video-CCAM-v1.1 14B:** Answer is B

**InternVL 2 40B:** Based on the provided frames, it appears that the cube does not exhibit any significant rotation. The cube remains in a consistent orientation throughout the frames.

Therefore, the most possible answer is:

'D': 'No rotation.'

rotation 180 YouTube

https://www.youtube.com/watch?v=JEJqy1Wlovw

In which direction(s) does the minute hand move on the clock face?

A. Clockwise then counter-clockwise.
B. Clockwise throughout.
C. No rotation.
D. Counter-clockwise throughout.
E. Counter-clockwise then clockwise.

Answer: E. Counter-clockwise then clockwise.

**GPT-4o:** Based on the provided frames, the minute hand of the clock appears to move in a clockwise direction throughout the sequence. There is no indication of the minute hand moving counter-clockwise in any of the frames.

Therefore, the most possible answer is:

'B': 'Clockwise throughout.'

**Claude 3.5 Sonnet:** Based on analyzing the sequence of clock images provided, the answer to the question "In which direction(s) does the minute hand move on the clock face?" is:

B. Clockwise throughout.

The images show the minute hand of the clock moving in a clockwise direction over time, starting near 12 o'clock and progressing around the clock face in a clockwise manner through various positions. There is no evidence of the minute hand moving counter-clockwise at any point in the sequence of images. The minute hand consistently moves clockwise as time advances across the frames.

**Qwen2 VL 72B instruct AWQ:** B

**Qwen2 VL 7B Instruct:** B

**Video-CCAM-v1.1 14B:** Answer is B.

**InternVL 2 40B:** Based on the provided frames, the minute hand on the clock face appears to move in a clockwise direction throughout. Therefore, the most possible answer is:

'B': Clockwise throughout.

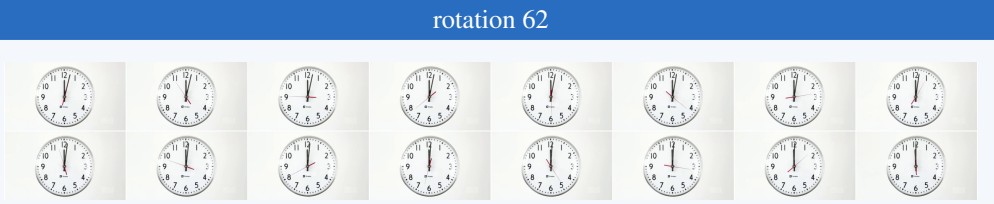

rotation 62

In what direction(s) does the minute hand move on the clock face?

A. Clockwise throughout.
B. Counter-clockwise then clockwise.
C. No rotation.
D. Counter-clockwise throughout.
E. Clockwise then counter-clockwise.

Answer: D. Counter-clockwise throughout.

**GPT-4o:** Based on the provided frames, the minute hand of the clock does not appear to move at all. It remains at the 12 o'clock position throughout all the frames.

Given the options, the most appropriate answer is:

'C': 'No rotation.'

**Claude 3.5 Sonnet:** Based on analyzing the sequence of clock images provided, the answer to the question "In what direction(s) does the minute hand move on the clock face?" is:

A: Clockwise throughout.

The minute hand (the longer red hand) can be observed moving in a clockwise direction across the series of images, starting near 12 o'clock and progressing around the clock face in a clockwise manner. There is no evidence of the minute hand moving counter-clockwise at any point in the sequence.

**Qwen2 VL 72B instruct AWQ:** A

**Qwen2 VL 7B Instruct:** D

**Video-CCAM-v1.1 14B:** Answer is A.

**InternVL 2 40B:** Based on the provided frames, the minute hand on the clock face appears to move in a clockwise direction throughout.

Therefore, the most possible answer is:

'A': 'Clockwise throughout.'

## rotation 103  YouTube

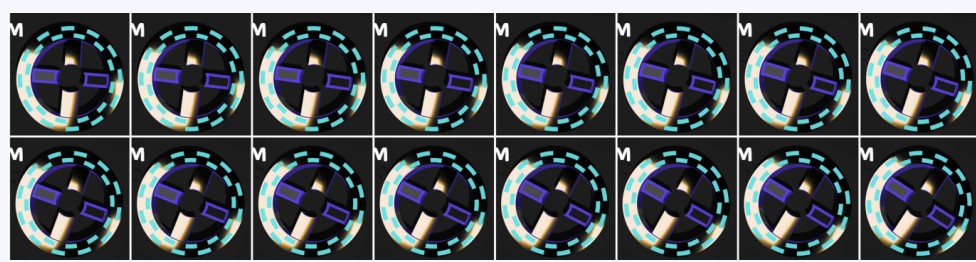

https://www.youtube.com/watch?v=R0xqQ0HNdgc

In what direction(s) is the wheel rotating?

A. Counter-clockwise throughout.
B. Counter-clockwise then clockwise.
C. Clockwise then counter-clockwise.
D. Clockwise throughout.
E. No rotation.

Answer: D. Clockwise throughout.

---

**GPT-4o:** Based on the provided frames, it appears that the wheel is rotating in a counter-clockwise direction initially and then changes to a clockwise direction. This can be inferred from the position of the colored segments and the central cross in each frame.

Therefore, the most likely answer is:

**Option B: Counter-clockwise then clockwise.**

---

**Claude 3.5 Sonnet:** Based on the analysis of the 16 frames provided, the most likely answer to the question "In what direction(s) is the wheel rotating?" is:

A: Counter-clockwise throughout.

Reasoning:
1. The wheel in the images consists of a central black circle with four purple rectangular spokes and an outer ring with alternating cyan and black segments.

2. Observing the position of the spokes across the frames, there appears to be a slight but consistent counter-clockwise movement.

3. The cyan segments on the outer ring also seem to be moving in a counter-clockwise direction relative to the fixed "M" letter in the top-left corner of each frame.

4. There is no apparent change in the direction of rotation throughout the sequence of frames.

While the movement is subtle and the frames are static, the overall impression from analyzing the sequence suggests a counter-clockwise rotation. Therefore, option A: "Counter-clockwise throughout" seems to be the most appropriate answer based on the given information.

---

**Qwen2 VL 72B instruct AWQ:** A

---

**Qwen2 VL 7B Instruct:** D

---

**Video-CCAM-v1.1 14B:** Answer is A.

**InternVL 2 40B:** Based on the provided frames, it is not possible to determine the direction of the wheel's rotation. The frames do not show a clear indication of the wheel's movement or rotation. Therefore, the most possible answer is:

'E': No rotation.

rotation 206  YouTube

https://www.youtube.com/watch?v=bxhdxLerP9Q

What direction(s) is the blue circle moving in?

A. Clockwise then counter-clockwise. B. Clockwise throughout.
C. Counter-clockwise then clockwise. D. Counter-clockwise throughout.
E. No rotation.

Answer: C. Counter-clockwise then clockwise.

**GPT-4o:** Based on the provided frames, the blue circle appears to be rotating in a counter-clockwise direction throughout the sequence. The position of the blue circle relative to the green center point and the orientation of the attached yellow and blue arms indicate a consistent counter-clockwise motion.

Therefore, the most likely answer is:

'D': 'Counter-clockwise throughout.'

**Claude 3.5 Sonnet:** Based on the sequence of images provided, the blue circle appears to be rotating in a counter-clockwise direction throughout the video. The yellow and blue arms attached to the blue circle consistently rotate counter-clockwise from frame to frame.

Starting from the first image and progressing through the sequence, we can observe that the arms of the blue circle move in a counter-clockwise manner. There is no evidence of the circle changing its direction of rotation at any point in the given frames.

Therefore, the most likely answer to the question "What direction(s) is the blue circle moving in?" is:

D: Counter-clockwise throughout.

**Qwen2 VL 72B instruct AWQ:** D

**Qwen2 VL 7B Instruct:** C

**Video-CCAM-v1.1 14B:** Answer is B.

**InternVL 2 40B:** Based on the provided frames, the blue circle appears to be moving in a clockwise direction throughout the video.

Answer: B

# J COMMON FAILURE CASES OF DIRECTION

## J.1 HUMAN

### J.1.1 EXAMPLE 1

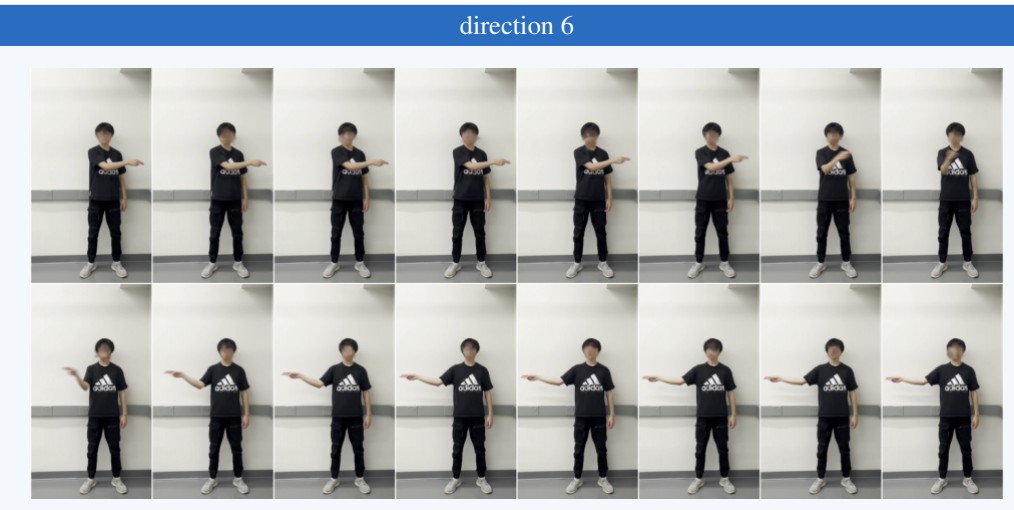

direction 6

In which direction(s) did the person's hand move?

A. First to the right then to the left.
B. First to the left then to the right.
C. Right.
D. Not moving at all
E. Left.

Answer: E. Left

**GPT-4o:** Based on the provided frames, the person's hand starts pointing to the right and then moves to the left.

Therefore, the most possible answer is:

**'B': 'First to the left then to the right.'**

**Claude 3.5 Sonnet:** Based on the sequence of images provided, the answer to the question "In which direction(s) did the person's hand move?" is:

C. Right.

The images show the person starting with their hand at their side, then moving it outward to their right and holding it there. There is no clear movement of the hand to the left or back and forth. The hand consistently moves and stays to the right side of the person's body throughout the sequence of frames.

**Qwen2 VL 72B instruct AWQ:** A

**Qwen2 VL 7B Instruct:** C

**Video-CCAM-v1.1 14B:** Answer is C.

**InternVL 2 40B:** Based on the frames provided, the person's hand appears to move from right to left. The correct answer is:
'A': 'First to the right then to the left.'

## J.1.2 EXAMPLE 2

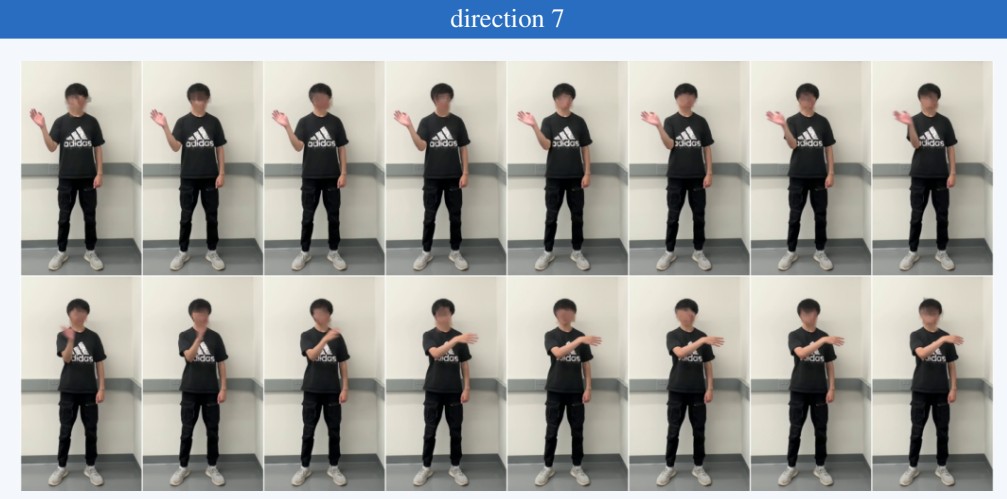

In which direction(s) did the person's hand move?

A. Right.
B. First to the right then to the left.
C. Left.
D. Not moving at all
E. First to the left then to the right.

Answer: A. Right.

**GPT-4o:** Based on the provided frames, the person's hand0 starts in an upright position and then moves to the left. The hand then returns to the right. This indicates a movement to the left and then back to the right.

Therefore, the most likely answer is:

**Option E: First to the left then to the right.**

**Claude 3.5 Sonnet:** Based on analyzing the sequence of images provided, the answer to the question "In which direction(s) did the person's hand move?" appears to be:

B: First to the right then to the left.

The images show the person's hand starting in a raised position on their right side, then moving across their body to the left, and then back towards the right. This back-and-forth motion is consistent with option B, which states the hand moved right then left.

**Qwen2 VL 72B instruct AWQ:** E

**Qwen2 VL 7B Instruct:** The person's hand moved from the left side to the right side of their body. Therefore, the answer is:

'E': 'First to the left then to the right.'

**Video-CCAM-v1.1 14B:** Answer is B.

**InternVL 2 40B:** Based on the frames provided, the person's hand appears to move from left to right.

The most possible answer is:

'E': First to the left then to the right.

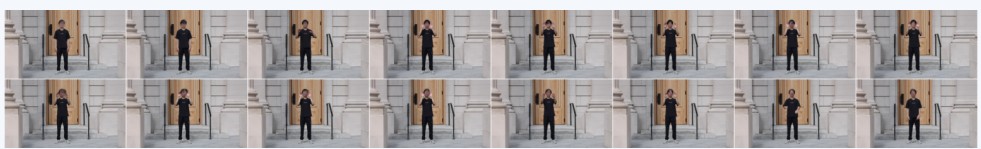

direction 118

What instruction did the person give to the camera?

A. Move to the right.
B. Move closer.
C. Stay still.
D. Move to the left.
E. Move further.

Answer: E. Move further.

**GPT-4o:** Based on the sequence of frames, it appears that the person is making hand gestures that could be interpreted as giving instructions to the camera. The person raises their hands and moves them slightly forward, which could be interpreted as a signal for the camera to move closer.

Therefore, the most likely answer based on the frames provided is:

**'B': 'Move closer.'**

**Claude 3.5 Sonnet:** Based on analyzing the sequence of images provided, the most likely instruction the person gave to the camera appears to be:

C: Stay still.

The reasoning for this conclusion is:

1. The person remains in the same position throughout all 16 frames, standing on the steps in front of the door.

2. The background and surroundings do not change, indicating the camera is not moving.

3. While the person's arms and hands move in various gestures, their overall body position and location stay constant.

4. There are no indications of the camera moving closer, further away, or to either side.

5. The consistent framing and stable background strongly suggest the camera remained stationary while capturing these images.

Therefore, the instruction to "Stay still" seems to be the most plausible option given the evidence in the image sequence provided.

**Qwen2 VL 72B instruct AWQ:** C

**Qwen2 VL 7B Instruct:** C

**Video-CCAM-v1.1 14B:** Answer is B.

**InternVL 2 40B:** Based on the frames provided, the person appears to be standing in front of a door, with their hands raised in front of their face. There is no clear indication of any specific instruction being given to the camera. However, since the person is stationary and not making any gestures that suggest movement, the most plausible answer would be:

'C': Stay still.

## J.2 OBJECT

### J.2.1 EXAMPLE 1

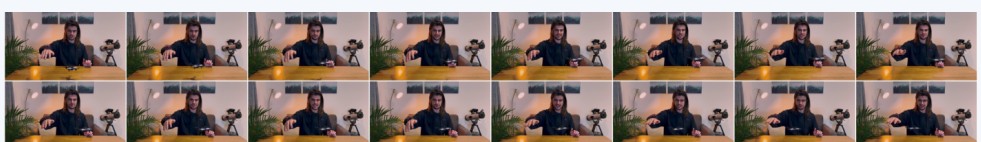

direction 242 YouTube

https://www.youtube.com/watch?v=ygtreF12Hks

In which direction(s) did the drone move during the video?

A. Leftwards then upwards.   B. Rightwards then upwards.
C. Downwards then leftwards. D. Rightwards then downwards.
E. Leftwards then downwards. F. Rightwards then leftwards.

Answer: F. Rightwards then leftwards.

**GPT-4o:** Based on the provided frames, the drone appears to be moving rightwards initially and then upwards. This can be inferred from the drone's position relative to the person and the table. In the first few frames, the drone is moving to the right, and in the later frames, it appears to be rising.

Therefore, the most likely answer is:

**'B': Rightwards then upwards.**

**Claude 3.5 Sonnet:** Based on analyzing the sequence of images provided, the most likely answer to the question "In which direction(s) did the drone move during the video?" is:

D. Rightwards then downwards.

The reasoning for this conclusion is:

1. In the initial frames, the drone appears to be hovering in a relatively stable position on the table in front of the person.

2. As the sequence progresses, there are subtle changes in the drone's position relative to the person's hands and the table edge, suggesting some movement.

3. The drone seems to shift slightly to the right (from the viewer's perspective) in the middle frames.

4. In the later frames, the drone appears to be lower in position compared to earlier frames, suggesting a downward movement.

5. The person's hand gestures and gaze direction also provide cues that they are following the drone's movement to the right and then down.

While it's difficult to determine precise movements from still images, the subtle changes in positioning across the frame sequence suggest a rightward movement followed by a downward movement as the most plausible interpretation of the drone's motion.

**Qwen2 VL 72B instruct AWQ:** B

**Qwen2 VL 7B Instruct:** B

**Video-CCAM-v1.1 14B:** Answer is B.

**InternVL 2 40B:** Based on the frames provided, the drone appears to move in the following directions:

1. Initially, the drone is seen moving towards the left side of the frame.
2. After moving leftwards, the drone is seen ascending upwards.

Therefore, the most likely answer is:

**A: Leftwards then upwards.**

J.2.2  EXAMPLE 2

A. Upwards then downwards.    B. Leftwards then rightwards.
C. Leftwards then upwards.    D. Downwards then rightwards.
E. Rightwards then upwards.   F. Rightwards then downwards.

**GPT-4o:** Based on the provided frames, the helicopter appears to be initially in the air and then gradually descends to land. The sequence of frames shows the helicopter moving downwards and slightly to the right as it approaches the ground.

Therefore, the most plausible answer is:

**Option D: Downwards then rightwards.**

**Claude 3.5 Sonnet:** Based on the sequence of images provided, the most likely answer to the question "What directional shifts can be observed in the helicopter's motion from the footage?" is:

E. Rightwards then upwards.

The images show a yellow ambulance helicopter initially on the ground. In the later frames, the helicopter appears to be lifting off vertically. While it's difficult to discern clear horizontal movement from still images, there seems to be a slight rightward shift in the helicopter's position relative to the background buildings as it begins to rise.

The upward motion is more clearly evident in the later frames where the helicopter is seen hovering above the ground. The rightward shift, while subtle, can be inferred from the changing perspective of the helicopter against the stationary background elements.

It's important to note that without seeing the actual video footage, this analysis is based on the limited information provided by the still frames. The rightward movement is less obvious than the upward motion, but option E best fits the observed changes in the helicopter's position across the image sequence.

**Qwen2 VL 72B instruct AWQ:** A

**Qwen2 VL 7B Instruct:** A

**Video-CCAM-v1.1 14B:** Answer is B.

**InternVL 2 40B:** The helicopter in the video appears to be moving in a leftwards direction initially and then upwards. Based on the frames provided, the most likely directional shift observed in the helicopter's motion is:

**C: Leftwards then upwards.**

## direction 243  YouTube

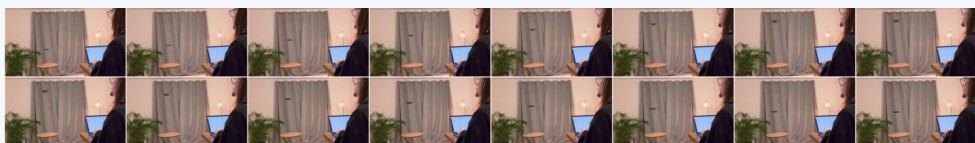

https://www.youtube.com/watch?v=ygtreF12Hks

In which direction(s) did the drone move during the video?

A. Downwards then rightwards. B. Upwards then downwards.
C. Rightwards then upwards.    D. Rightwards then leftwards.
E. Leftwards then upwards.     F. Rightwards then downwards.

Answer: B. Upwards then downwards.

**GPT-4o:** Based on the provided frames, the drone's movement can be analyzed as follows:

1. The drone starts near the center of the frame.
2. It moves slightly to the left and upwards.
3. It continues moving upwards and to the left.
4. It reaches the top left corner.
5. It then starts moving downwards.
6. It continues moving downwards.
7. It moves back towards the center.
8. It continues moving downwards and slightly to the right.
9. It moves further downwards.
10. It continues moving downwards and slightly to the right.
11. It moves back towards the center.
12. It continues moving downwards.
13. It moves further downwards.
14. It continues moving downwards and slightly to the right.
15. It moves back towards the center.
16. It continues moving downwards and slightly to the right.

From this analysis, the drone initially moves leftwards and upwards, then moves downwards and slightly to the right.

The most fitting option based on the frames is:
'E': 'Leftwards then upwards.'

**Claude 3.5 Sonnet:** Based on analyzing the sequence of frames provided, the most likely answer to the question "In which direction(s) did the drone move during the video?" is:

C. Rightwards then upwards.

The reasoning for this conclusion is:

1.  In the initial frames, the drone appears as a small blurry object near the center of the curtain.

2.  As the frames progress, the drone's position shifts gradually towards the right side of the curtain, indicating rightward movement.

3.  In the later frames, the drone's position appears to move higher up on the curtain, suggesting upward movement.

4. The final frames show the drone positioned higher and more to the right compared to its initial location.

This rightward then upward movement pattern best matches option C from the given choices. While there may be some subtle variations or brief movements in other directions, the overall trend visible across the frame sequence supports this answer as the most probable description of the drone's primary movement directions.

**Qwen2 VL 72B instruct AWQ:** B

**Qwen2 VL 7B Instruct:** C

**Video-CCAM-v1.1 14B:** Answer is F.

**InternVL 2 40B:** Based on the frames provided, the drone appears to move in the direction of rightwards then downwards.

Therefore, the most possible answer is:

'F': Rightwards then downwards.

---

**direction 277 YouTube**

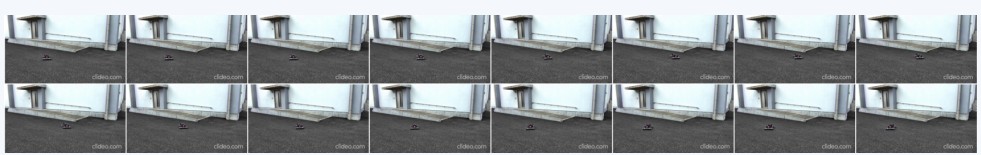

https://www.youtube.com/watch?v=bNq60yuTRf8

What was the directional trend of the toy car's movement throughout the
    clip?

A. Upwards then rightwards.  B. Downwards then leftwards.
C. Leftwards then rightwards. D. Rightwards then leftwards.
E. Leftwards then upwards.   F. Downwards then upwards.

Answer: D. Rightwards then leftwards.

---

**GPT-4o:** Based on the provided frames, the toy car appears to be moving in a generally leftward direction initially and then changes direction to move rightward.

Here is the analysis of the frames:
1. The car starts from a position closer to the right side of the frame.
2. It moves leftward in the subsequent frames.
3. Around the middle frames, the car starts to change direction.
4. In the later frames, the car appears to be moving rightward.

Given this observation, the most plausible answer is:

**Option C: Leftwards then rightwards.**

---

**Claude 3.5 Sonnet:** Based on analyzing the 16 frames provided, the most likely answer to the question "What was the directional trend of the toy car's movement throughout the clip?" is:

C: Leftwards then rightwards.

Reasoning:
1. In the initial frames, the toy car appears to be positioned more towards the right side of the image.
2. As the sequence progresses, the car seems to move slightly towards the left, getting closer to the center of the frame.
3. In the later frames, there's a subtle shift where the car appears to move back towards the right side of the image.

While the movement is not dramatic, there does seem to be a slight leftward trend followed by a rightward trend. This makes option C the most plausible answer among the given choices, based solely on the information provided in these static frames.

It's important to note that the movement is very subtle, and without seeing the actual video, it's challenging to be absolutely certain. However, given the instruction to make a best guess based on the available information, option C appears to be the most fitting answer.

---

**Qwen2 VL 72B instruct AWQ:** D

---

**Qwen2 VL 7B Instruct:** C

---

**Video-CCAM-v1.1 14B:** Answer is B.

**InternVL 2 40B:** Based on the frames provided, the toy car appears to move in a leftward direction initially and then turns rightward. Therefore, the most possible answer is:

'C': Leftwards then rightwards.

direction 349  YouTube

https://www.youtube.com/watch?v=8u0tTaACYqQ

In what direction did the car move?

A. Downwards.  B. Rightwards.   C. Upwards.
D. No movements.   E. Leftwards.

Answer: E. Leftwards.

**GPT-4o:** Based on the sequence of frames provided, the car appears to be moving from the left side of the frame to the right side of the frame. This is evident from the car's initial position on the left and its final position on the right.

Therefore, the most likely answer is:

**Option B: Rightwards.**

**Claude 3.5 Sonnet:** Based on the sequence of images provided, the answer to the question "In what direction did the car move?" is:

B. Rightwards.

The frames show a red car moving across a grassy field with trees in the background. As the sequence progresses, we can see the car consistently moving from left to right in the frame. The car is kicking up dust and at times even becoming airborne as it travels across the uneven terrain, but its overall direction of movement is clearly rightwards through the field.

**Qwen2 VL 72B instruct AWQ:** B

**Qwen2 VL 7B Instruct:** B

**Video-CCAM-v1.1 14B:** The car is moving to the right. So the answer is rightwards.

**InternVL 2 40B:** Based on the frames provided, the car appears to be moving upwards, as it is shown in various stages of a jump. Therefore, the most possible answer is:

'C': Upwards.

---

### direction 294 YouTube

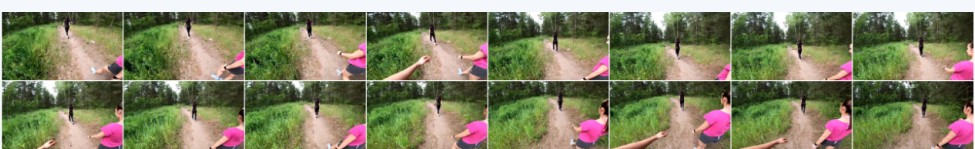

https://www.youtube.com/watch?v=UgXVVV72dp8

In what direction did the camera move?

A. Backward    B. No movement C. Forward

Answer: A. Backward

---

**GPT-4o:** Based on the provided frames, it appears that the camera is moving forward. The person in the black outfit and the dog ahead of them are getting closer in each subsequent frame, indicating forward movement.

Therefore, the most likely answer is:

**Option C: Forward**

---

**Claude 3.5 Sonnet:** Based on analyzing the sequence of images provided, the most likely answer to the question "In what direction did the camera move?" is:

C: Forward

The images show a consistent perspective of following behind a person walking on a forest path with a dog. The camera appears to be moving forward along the path, maintaining a similar distance behind the person throughout the sequence. This gives the impression of forward movement of the camera along with the walking subjects.

---

**Qwen2 VL 72B instruct AWQ:** C

---

**Qwen2 VL 7B Instruct:** C

---

**Video-CCAM-v1.1 14B:** Answer is C.

---

**InternVL 2 40B:** Based on the frames provided, it appears that the camera is moving backward. The perspective shows a clear view of the path and the people walking away from the camera, indicating that the camera is moving in the opposite direction of the subjects.

Therefore, the most possible answer is:

'A': 'Backward'

---

## J.3  SIMULATED

### J.3.1  EXAMPLE 1

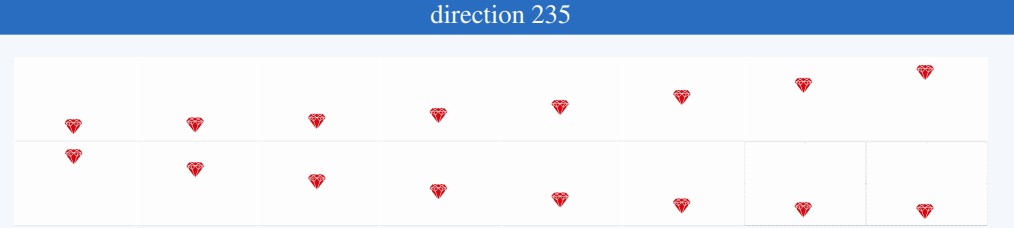

> **direction 235**
>
> What is the direction of the red diamond's movement from the video?
>
> A. Leftwards then rightwards. B. Rightwards then downwards.
> C. Downwards then rightwards. D. Rightwards then upwards.
> E. Downwards then upwards. F. Upwards then downwards.
>
> Answer: F. Upwards then downwards.

**GPT-4o:** Based on the provided frames, the red diamond starts at the bottom of the frame and moves upwards. It then moves slightly to the right as it continues to move upwards.

Therefore, the most likely answer is:

**Option D: Rightwards then upwards.**

**Claude 3.5 Sonnet:** Based on the 16 frames provided, I cannot determine any movement of the red diamond shape. The diamond appears to be in the same position in the center of each frame, without any apparent change in its location or orientation.

However, since I'm instructed to make a guess even if there's not enough information, I'll choose the option that seems most plausible given the question:

D: Rightwards then upwards.

This is purely a guess based on the question asking about movement, even though no movement is visible in the provided frames. I want to emphasize that this answer is not based on any observed motion in the images, but is simply an arbitrary selection from the given options.

**Qwen2 VL 72B instruct AWQ:** E

**Qwen2 VL 7B Instruct:** B

**Video-CCAM-v1.1 14B:** Answer is D.

**InternVL 2 40B:** Based on the frames provided, the red diamond appears to move upwards and then downwards. Therefore, the most possible answer is:

**E: Downwards then upwards**

direction 238  YouTube

https://www.youtube.com/watch?v=COHi3qWME0M&t=285s

What is the direction of the game character's movement from the video?

A. Downwards then leftwards.  B. Leftwards then rightwards.
C. Downwards then rightwards. D. Downwards then upwards.
E. Leftwards then downwards. F. Rightwards then downwards.

Answer: F. Rightwards then downwards.

**GPT-4o:** Based on the provided frames, the game character starts at the top center of the field and moves downwards. As the frames progress, the character then moves towards the right side of the field.

Therefore, the direction of the game character's movement from the video is:

Option 'C': Downwards then rightwards.

**Claude 3.5 Sonnet:** Based on analyzing the sequence of frames provided, the direction of the game character's movement appears to be:

C: Downwards then rightwards

The character starts near the top of the green field in the early frames, then moves downward towards the center. In the later frames, the character can be seen moving towards the right side of the field. This downward then rightward movement pattern best matches option C from the given choices.

**Qwen2 VL 72B instruct AWQ:** C

**Qwen2 VL 7B Instruct:** The direction of the game character's movement from the video is downwards then rightwards.

**Video-CCAM-v1.1 14B:** Answer is E.

**InternVL 2 40B:** Based on the frames provided, the game character appears to be moving downwards and then rightwards.

Therefore, the correct answer is:

**C: Downwards then rightwards**

J.3.3 EXAMPLE 3

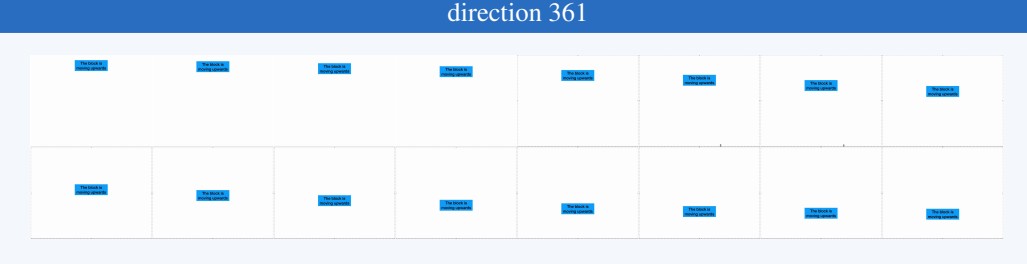

**direction 361**

How did the blue text block progress?

A. Upwards.     B. Downwards.  C. Leftwards.
D. Rightwards. E. No movements.

Answer: B. Downwards.

**GPT-4o:** Based on the provided frames, the blue text block appears to be moving upwards. In the initial frames, the block is positioned higher on the screen, and as the frames progress, it moves downwards. This indicates that the block is moving upwards in the video.

Therefore, the most possible answer is:

**'A': Upwards.**

**Claude 3.5 Sonnet:** Based on the 16 frames provided, I can answer the question about how the blue text block progressed.

The answer is A: Upwards.

In all 16 frames, we see a blue rectangular block containing the text "The block is moving upwards". The block remains in the same position in the center of each frame. While we don't see the block itself moving upwards across the frames, the text within the block explicitly states that it is moving upwards.

Given that we are instructed to use only the information provided in the frames to answer the question, and not rely on external knowledge, the most logical conclusion based on the text in the blue block is that the block is progressing upwards, even though we don't visually see this movement in the static frames provided.

**Qwen2 VL 72B instruct AWQ:** B

**Qwen2 VL 7B Instruct:** A

**Video-CCAM-v1.1 14B:** Answer is A

**InternVL 2 40B:** The blue text block in the video appears to be moving upwards. This can be inferred from the consistent position of the text block in the frames provided, which shows it progressively moving upwards.

Therefore, the correct answer is:

A: Upwards.

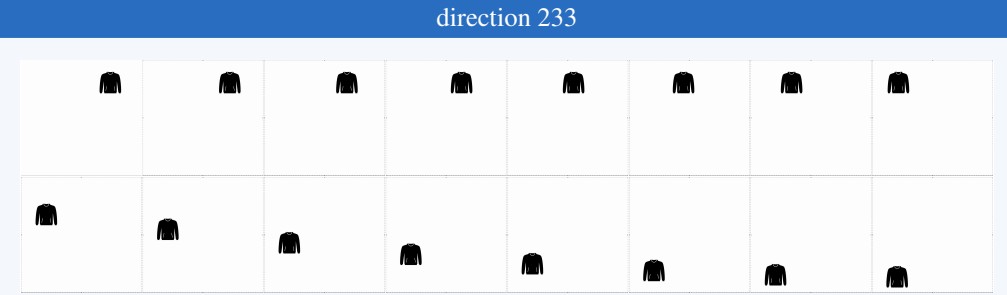

**direction 233**

What is the direction of the black sweater icon's movement from the video ?

A. Leftwards then rightwards.    B. Leftwards then downwards.
C. Downwards then leftwards.     D. Downwards then upwards.
E. Upwards then rightwards.      F. Downwards then rightwards

Answer: B. Leftwards then downwards.

**GPT-4o:** Based on the provided frames, the black sweater icon starts at the top right of the frame and moves downwards and then to the left.

Therefore, the most likely answer is:

**Option C: Downwards then leftwards.**

**Claude 3.5 Sonnet:** Based on the sequence of frames provided, the black sweater icon appears to redmove in a diagonal direction from the bottom right corner towards the top left corner of the image. The movement seems to be primarily upwards and leftwards.

Given the options provided, the closest match to this observed movement would be:

E: Upwards then rightwards.

While this option is not a perfect description of the movement seen in the frames (as the icon moves leftwards rather than rightwards), it is the closest match among the given choices. The upward movement is correctly captured, and it's the only option that includes an upward direction.

It's important to note that this answer is based solely on the information provided in the image frames and the given options, as instructed. The movement is not entirely clear from the limited frames, but this appears to be the best guess based on the available information and options.

**Qwen2 VL 72B instruct AWQ:** B

**Qwen2 VL 7B Instruct:** The direction of the black sweater icon's movement from the video is downwards then rightwards.

**Video-CCAM-v1.1 14B:** Answer is D.

**InternVL 2 40B:** The black sweater icon in the video moves from the top-right corner to the bottom-left corner. This movement can be described as moving downwards and then leftwards.
Therefore, the correct answer is:
**C: Downwards then leftwards**

direction 278 YouTube

https://www.youtube.com/watch?v=YfGIoAiml7w

What was the directional trend of the car's movement throughout the clip?

A. Downwards then leftwards. B. Upwards then downwards.
C. Downwards then upwards.    D. Rightwards then leftwards.
E. Leftwards then downwards. F. Downwards then rightwards.

Answer: D. Rightwards then leftwards.

---

**GPT-4o:** Based on the provided frames, the car initially moves downwards and then starts to turn right. This is evident from the car's position and the activation of the right turn signal lights.

Therefore, the most likely answer is:

Option F: Downwards then rightwards

---

**Claude 3.5 Sonnet:** Based on the sequence of frames provided, the most likely answer to the question "What was the directional trend of the car's movement throughout the clip?" is:

F: Downwards then rightwards

Explanation:
1. In the initial frames, the car appears to be moving straight along the road.
2. As the sequence progresses, the car starts to tilt, suggesting a change in direction.
3. The tilting of the car becomes more pronounced, with its front end pointing downwards and to the right.
4. In the later frames, the car seems to straighten out again, but it has moved to a position further right on the road compared to its starting position.

This sequence of movements suggests that the car first moved downwards (tilting forward) and then rightwards, making option F the most plausible answer among the given choices.

---

**Qwen2 VL 72B instruct AWQ:** C

---

**Qwen2 VL 7B Instruct:** C

---

**Video-CCAM-v1.1 14B:** Answer is F.

---

**InternVL 2 40B:** The directional trend of the car's movement throughout the clip can be observed by analyzing the positions of the car in the frames provided.

1. In the first frame, the car is positioned at the bottom of the road.
2. In subsequent frames, the car moves upwards along the road.
3. Finally, the car is seen moving downwards towards the bottom of the road again.

Based on this observation, the most accurate answer is:

'B': Upwards then downwards.

direction 239  YouTube

https://www.youtube.com/watch?v=COHi3qWME0M&t=285s

What is the direction of the game character's movement from the video?

A. Leftwards then rightwards. B. Leftwards then upwards.
C. Downwards then rightwards. D. Downwards then leftwards.
E. Upwards then rightwards.  F. Leftwards then downwards.

Answer: B. Leftwards then upwards.

**GPT-4o:** Based on the provided frames, the game character starts at the bottom right corner and moves leftwards along the bottom edge of the field. The character then turns and moves upwards along the left edge of the field.

Therefore, the direction of the game character's movement is:

'B': Leftwards then upwards.

**Claude 3.5 Sonnet:** Based on analyzing the sequence of frames provided, the direction of the game character's movement appears to be:

A. Leftwards then rightwards.

The character starts in the bottom right corner of the green field.  It then moves leftwards along the bottom edge of the field.  After reaching the bottom left corner, the character changes direction and moves rightwards back along the bottom edge. This leftwards then rightwards movement pattern matches option A from the given choices.

**Qwen2 VL 72B instruct AWQ:** D

**Qwen2 VL 7B Instruct:** The direction of the game character's movement from the video is 'Leftwards then downwards.'

**Video-CCAM-v1.1 14B:** Answer is F.

**InternVL 2 40B:** Based on the frames provided, the game character appears to move from the bottom left corner to the bottom right corner. This suggests that the character is moving downwards and then to the right.

Therefore, the most possible answer is:

'D': Downwards then leftwards

# K  COMMON FAILURE CASES OF VELOCITY & FREQUENCY

## K.1  HUMAN

### K.1.1  EXAMPLE 1

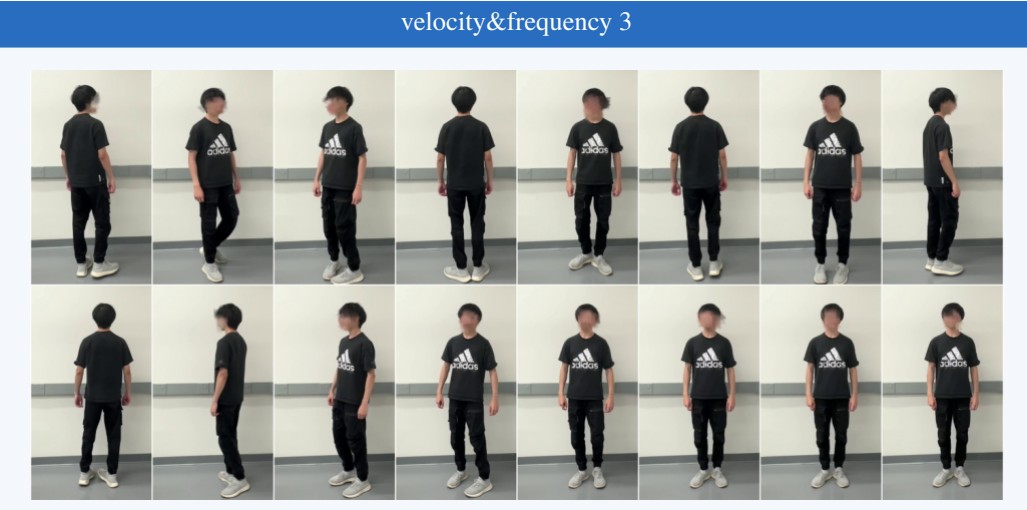

velocity&frequency 3

What is the pattern of the person's rotation speed?

A. Decelerating.     B. Accelerating.
C. Constant speed.   D. Not moving at all.

Answer: A. Decelerating.

---

**GPT-4o:** Based on the provided frames, the person appears to be rotating. The sequence of frames shows the person turning from facing away from the camera to facing the camera and then back to facing away again. The rotation seems to be happening at a consistent pace throughout the frames.

Given the options:
{'A': 'Decelerating.', 'B': 'Accelerating.', 'C': 'Constant speed.', 'D': 'Not moving at all.'}

The most plausible answer based on the frames is:

'C': 'Constant speed.'

---

**Claude 3.5 Sonnet:** Based on the provided frames, I cannot definitively determine the pattern of the person's rotation speed. The images show the person in various static poses, facing different directions, but do not capture continuous motion or rotation. However, since I'm asked to make a best guess based on the available information, I would choose:

C: Constant speed.

This is because the frames show the person in different orientations that appear to be evenly spaced (front view, side view, back view, etc.), which could suggest a constant rotation if we assume the frames are sampled at regular intervals. However, this is purely speculative given the limited information in the static images.

---

**Qwen2 VL 72B instruct AWQ:** C

---

**Qwen2 VL 7B Instruct:** C

**Video-CCAM-v1.1 14B:** Answer is C.

**InternVL 2 40B:** Based on the provided frames, the person appears to be standing in different positions, facing different directions. There is no clear indication of rotation or movement in the frames. Therefore, the most possible answer is:

D: Not moving at all.

### K.2.1    Example 1

---

**velocity&frequency 42  YouTube**

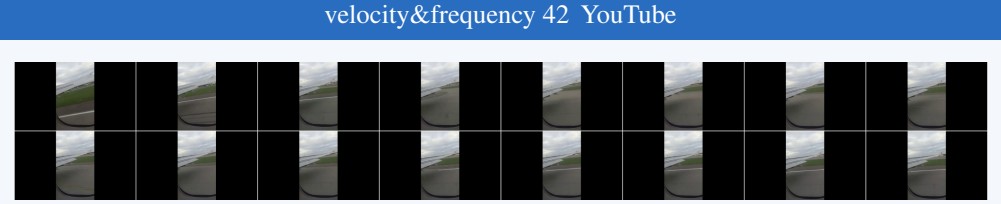

https://www.youtube.com/watch?v=uwwzoftsMXc

What is the pattern of the plane's speed on the runway?

A. Accelerating.     B. Decelerating.
C. Not moving at all. D. Constant speed.

Answer: B. Decelerating.

---

**GPT-4o:** Based on the 16 frames provided, the plane appears to be moving on the runway. The background and the position of the wing relative to the ground change slightly between frames, indicating motion. However, the changes are not drastic, suggesting that the plane is not accelerating rapidly.

Given the subtle changes in the frames, the most plausible answer is that the plane is moving at a constant speed.

Therefore, the answer is:

**D: Constant speed.**

---

**Claude 3.5 Sonnet:** Based on analyzing the 16 frames provided, the most likely answer to the question "What is the pattern of the plane's speed on the runway?" is:

A: Accelerating.

While it's difficult to determine the exact speed from still images, there are a few clues that suggest the plane is likely accelerating:

1. The frames show a consistent view of the runway and surrounding area, indicating the plane is moving forward along the runway.

2. In typical takeoff procedures, planes accelerate down the runway to reach the necessary speed for liftoff.

3. The scenery appears to be moving past the window at an increasing rate in the later frames, though this is subtle.

4. There are no signs of deceleration or stopping, which would be expected if the plane was slowing down or maintaining a constant speed.

Given these observations and the typical behavior of planes during takeoff, acceleration seems to be the most probable pattern of speed for the plane in this sequence of images. However, I want to emphasize that this is an educated guess based on limited information from still frames, and without more dynamic data, it's impossible to state this with absolute certainty.

---

**Qwen2 VL 72B instruct AWQ:** A

---

**Qwen2 VL 7B Instruct:** A

**Video-CCAM-v1.1 14B:** Answer is A.

**InternVL 2 40B:** Based on the frames provided, it appears that the plane is moving along the runway. The frames show the plane's wing and the runway in the background, indicating that the plane is in motion. Therefore, the most possible answer is:

A: Accelerating.

velocity&frequency 46  YouTube

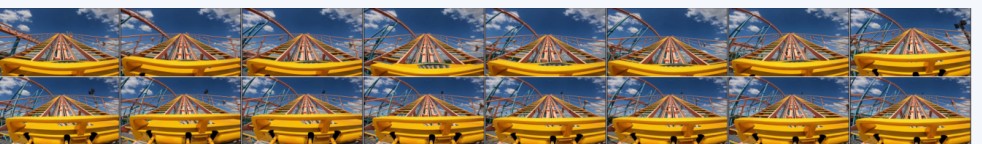

```
https://www.youtube.com/watch?v=dmqkEmdE5nE

What is the pattern of the roller coaster's speed on the rail?

A. Accelerating.      B. Not moving at all.
C. Decelerating.      D. Constant speed.

Answer: C. Decelerating.
```

**GPT-4o:** Based on the provided frames, the roller coaster appears to be ascending a hill. The frames show a consistent upward movement along the track, which suggests that the roller coaster is being pulled up by a chain lift mechanism. This typically means the roller coaster is moving at a constant speed during the ascent.

Given this information, the most likely answer is:

**Option D: Constant speed.**

**Claude 3.5 Sonnet:** Based on the series of images provided, which show a first-person view from a roller coaster car ascending a steep incline, I would conclude that the most likely answer to the question "What is the pattern of the roller coaster's speed on the rail?" is:

A: Accelerating.

The images consistently show the roller coaster car climbing up a steep incline. As the car moves up this initial hill, it is likely being pulled by a chain lift or similar mechanism. This would result in a steady acceleration as the car gains both height and speed while ascending the track.

While we cannot see the actual speed directly, the consistent upward motion and the gradually changing perspective of the track and surrounding structures suggest an increasing velocity. This acceleration is typical for the initial climb of many roller coasters, building up potential energy that will be converted to kinetic energy for the rest of the ride.

**Qwen2 VL 72B instruct AWQ:** A

**Qwen2 VL 7B Instruct:** A

**Video-CCAM-v1.1 14B:** Answer is A.

**InternVL 2 40B:** Based on the frames provided, the roller coaster appears to be stationary at the top of the track. Therefore, the most possible answer is:

B: Not moving at all.

## velocity&frequency 44 YouTube

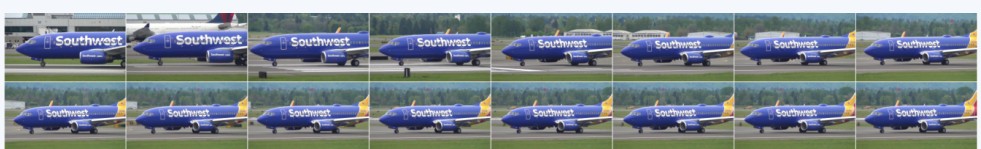

https://www.youtube.com/watch?v=eqTZ74aXa68

What is the pattern of the plane's speed on the runway?

A. Decelerating.     B. Not moving at all.
C. Constant speed.   D. Accelerating.

Answer: A. Decelerating.

**GPT-4o:** Based on the provided frames, the plane's position relative to the background and the runway markings does not change significantly. This suggests that the plane is not moving or is moving very slowly.

Given the options:
{'A': 'Decelerating.', 'B': 'Not moving at all.', 'C': 'Constant speed.', 'D': 'Accelerating.'}

The most plausible answer based on the frames is:

'B': 'Not moving at all.'

**Claude 3.5 Sonnet:** Based on the series of images provided, I would say the most likely answer to the question "What is the pattern of the plane's speed on the runway?" is:

C: Constant speed.

The images show a Southwest Airlines plane on a runway, but there is no clear indication of acceleration or deceleration. The plane appears to be in a similar position relative to the background in most frames, suggesting it is moving at a steady pace. While we can't determine the exact speed, the visual evidence points to the plane maintaining a constant speed as it moves along the runway.

**Qwen2 VL 72B instruct AWQ:** D

**Qwen2 VL 7B Instruct:** D

**Video-CCAM-v1.1 14B:** Answer is A.

**InternVL 2 40B:** Based on the frames provided, it appears that the plane is moving steadily along the runway. There is no visible change in the plane's position relative to the background, suggesting a constant speed.

Therefore, the most possible answer is:

C: Constant speed.

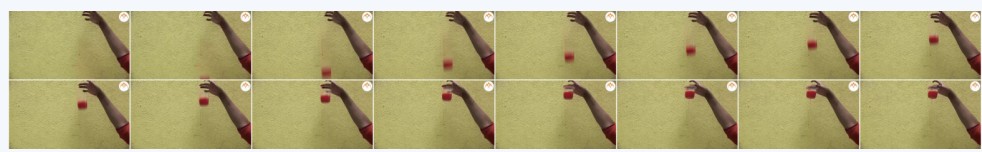

velocity&frequency 66  YouTube

https://www.youtube.com/watch?v=vpQQJTyiokE

What is the pattern of the object's speed in the video?

A. Not moving at all.   B. Constant speed.
C. Decelerating.        D. Accelerating.

Answer: C. Decelerating.

**GPT-4o:** Based on the provided frames, it appears that the object (a cup with liquid) is initially moving downward and then upward. The motion suggests that the object is being dropped and then caught or swung back up. The blur in the images indicates motion, and the distance between the object and the hand changes over time.

Given the sequence of frames, the object seems to be accelerating as it moves downward due to gravity and then decelerating as it is caught or swung back up.

Therefore, the most plausible answer based on the frames is:

**Option D: Accelerating.**

**Claude 3.5 Sonnet:** Based on analyzing the sequence of frames provided, the pattern of the object's speed in the video appears to be:

D. Accelerating.

The reasoning for this conclusion is:

1. In the initial frames, the red object (which appears to be a small container with liquid) is not visible or barely visible at the bottom of the frame.

2. As the sequence progresses, the object becomes more visible and moves upward in the frame.

3. The rate at which the object moves upward seems to increase throughout the sequence of frames.

4. In the final frames, the object is clearly visible near the top of the frame, and appears to be moving more rapidly compared to earlier frames.

5. The blurring of the object in later frames also suggests increased speed.

This pattern of movement, starting slow and gradually increasing in speed as it moves upward, is consistent with acceleration. The object appears to be gaining speed throughout the video sequence, rather than moving at a constant speed or slowing down.

**Qwen2 VL 72B instruct AWQ:** D

**Qwen2 VL 7B Instruct:** D

**Video-CCAM-v1.1 14B:** Answer is C.

**InternVL 2 40B:** Based on the frames provided, the object appears to be moving downward in a consistent manner. There is no indication of sudden changes in speed or direction. Therefore, the most likely pattern of the object's speed in the video is:

'B': Constant speed.

## K.2.5 EXAMPLE 5

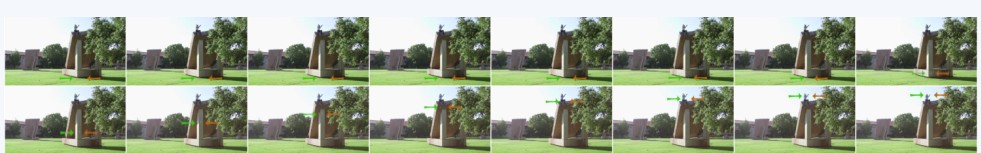

---

velocity&frequency 72 YouTube

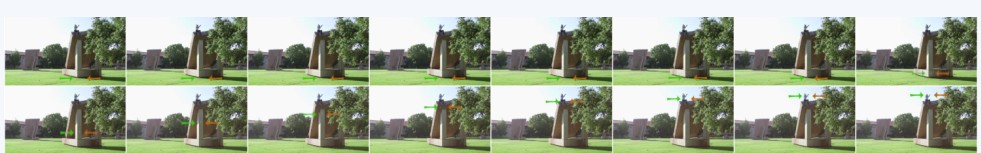

https://www.youtube.com/watch?v=AiGM4lrpA_k

What is balls' speed pattern?

A. Decelerating.       B. Not moving at all.
C. Accelerating.       D. Constant speed.

Answer: A. Decelerating.

---

**GPT-4o:** Based on the provided frames, the ball appears to be moving from the left side of the frame to the right side. The distance covered by the ball in each subsequent frame seems to be increasing, indicating that the ball is accelerating.

Therefore, the most likely answer is:

**'C': Accelerating.**

---

**Claude 3.5 Sonnet:** Based on the sequence of images provided, I can analyze the movement of the green and orange arrows to infer the ball's speed pattern. The arrows appear to represent the ball's motion, with the green arrow indicating forward movement and the orange arrow indicating backward movement.

Throughout the sequence, the relative sizes of the green and orange arrows remain consistent. This suggests that the ball is moving back and forth at a steady rate, without any noticeable acceleration or deceleration.

Given this observation, the most likely answer to the question "What is balls' speed pattern?" is:

D: Constant speed.

The consistent size and positioning of the arrows across the frames indicate that the ball is moving at a constant speed, oscillating back and forth without changing its velocity.

---

**Qwen2 VL 72B instruct AWQ:** D

---

**Qwen2 VL 7B Instruct:** C

---

**Video-CCAM-v1.1 14B:** Answer is A.

---

**InternVL 2 40B:** Based on the provided frames, it is not possible to determine the speed pattern of the balls as there are no visible balls in the frames. The frames show a person standing on a structure in a park-like setting. Therefore, the most possible answer would be:

'B': Not moving at all.

## K.2.6   EXAMPLE 6

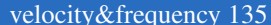

velocity&frequency 135

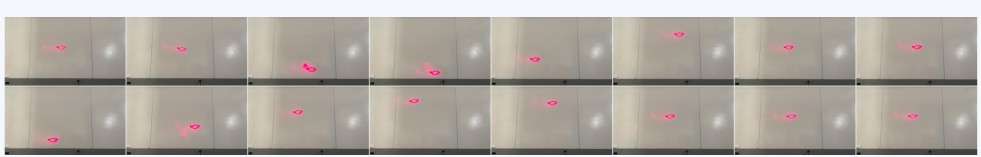

How does the frequency of the laser pointer movement change?

A. Decelerating.     B. Not moving at all.
C. Accelerating.     D. Constant speed.

Answer: A. Decelerating.

**GPT-4o:** Based on the provided frames, the laser pointer appears to be moving in a circular pattern. The distance between the laser pointer's positions in consecutive frames seems to be increasing, indicating that the speed of the laser pointer is increasing over time.

Therefore, the most likely answer is:

**'C': Accelerating.**

**Claude 3.5 Sonnet:** Based on analyzing the 16 frames provided, I would say the most likely answer is C: Accelerating.

The images show a pink laser pointer drawing of what appears to be a cartoon fish or animal face on a white wall or board. While the fish shape itself remains fairly constant, the blurred trail behind it seems to change in length and intensity across the frames.

In the earlier frames, the blurred trail is relatively short and faint. As the sequence progresses, the trail becomes longer and more pronounced in many of the frames. This suggests the laser pointer is being moved more quickly over time, leaving a more prominent trail.

The increasing length and intensity of the blur trails implies the laser pointer's movement is speeding up or accelerating throughout the video sequence. While there is some variation between individual frames, the overall trend points towards faster movement and longer trails in later frames compared to earlier ones.

Therefore, based solely on the visual information provided in these 16 sample frames, the most likely answer to how the frequency of laser pointer movement changes is C: Accelerating.

**Qwen2 VL 72B instruct AWQ:** C

**Qwen2 VL 7B Instruct:** C

**Video-CCAM-v1.1 14B:** Answer is A.

**InternVL 2 40B:** Based on the provided frames, the laser pointer appears to move in a consistent manner across the whiteboard. There is no significant change in the speed or direction of the movement. Therefore, the most likely answer is:

'D': Constant speed.

## velocity&frequency 136  YouTube

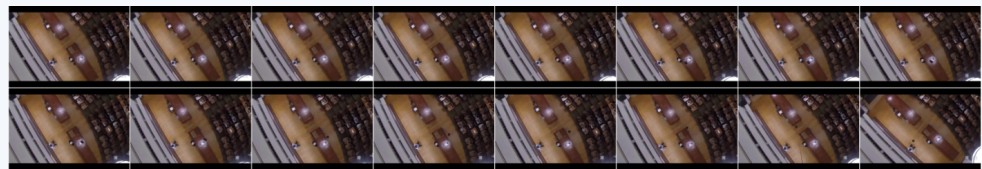

https://www.youtube.com/watch?v=QA1ZAMtxbBA

What is the pattern of the pendulum's movement speed throughout the
    entire video?

A. Not moving at all. B. Constant speed.
C. Accelerating.     D. Decelerating.

Answer: C. Accelerating.

**GPT-4o:** Based on the provided frames, it appears that the pendulum is moving. The pendulum's position changes slightly from frame to frame, indicating movement. However, the changes in position are subtle and do not suggest a significant acceleration or deceleration.

Given the uniformity of the movement across the frames, the most plausible answer is that the pendulum is moving at a constant speed.

Therefore, the answer is:

**'B': Constant speed.**

**Claude 3.5 Sonnet:** Based on the provided frames, I cannot determine the pattern of a pendulum's movement speed, as there is no pendulum visible in these images. The frames show an aerial view of what appears to be a room or hall with furniture and seating.

However, since I'm instructed to provide a guess based on the given options, I would choose:

B: Constant speed.

This is purely a guess and not based on any visual evidence from the provided frames. In reality, there is no information about a pendulum or its movement in these images.

**Qwen2 VL 72B instruct AWQ:** D

**Qwen2 VL 7B Instruct:** C

**Video-CCAM-v1.1 14B:** Answer is B.

**InternVL 2 40B:** Based on the provided frames, it is not possible to determine the pattern of the pendulum's movement speed throughout the entire video. The frames do not show any pendulum or its movement. Therefore, the most possible answer is:

'NOT POSSIBLE TO DETERMINE.'

## velocity&frequency 139 YouTube

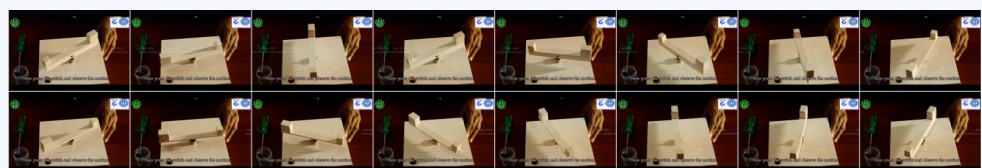

https://www.youtube.com/watch?v=mu9jAHgDjQg

What is the object's rotational speed pattern?

A. Decelerating.       B. Not moving at all.
C. Accelerating.       D. Constant speed.

Answer: A. Decelerating.

**GPT-4o:** Based on the 16 frames provided, the object appears to be rotating at a constant speed. The positions of the object in each frame suggest a uniform angular displacement between consecutive frames, indicating that the rotational speed is not changing.

Therefore, the most likely answer is:

**Option D: Constant speed.**

**Claude 3.5 Sonnet:** Based on the sequence of images provided, I can observe a rotating arm on a white platform. The arm appears to be rotating at a constant speed throughout the frames. There are no clear indications of acceleration or deceleration in the motion of the arm across the sequence of images.

Given the question "What is the object's rotational speed pattern?" and the available options, the most likely answer based solely on the visual information in these frames is:

D: Constant speed.

The arm seems to maintain a steady rotational movement without any apparent changes in speed from frame to frame. However, it's important to note that making a definitive determination about speed patterns from still images can be challenging, especially without precise timing information between frames.

**Qwen2 VL 72B instruct AWQ:** D

**Qwen2 VL 7B Instruct:** D

**Video-CCAM-v1.1 14B:** Answer is A.

**InternVL 2 40B:** Based on the provided frames, it is not possible to determine the exact rotational speed pattern of the object. However, we can observe that the object is moving in a circular path. Given this information, the most possible answer would be:

'D': Constant speed.

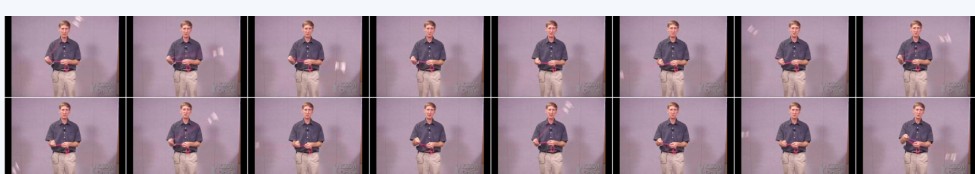

velocity&frequency 140 YouTube

https://www.youtube.com/watch?v=sqk4SNdnjRk

What is the object's rotational speed pattern?

A. Not moving at all. B. Decelerating.
C. Constant speed.     D. Accelerating.

Answer: D. Accelerating.

**GPT-4o:** Based on the provided frames, the object appears to be rotating at a constant speed. The position of the object in each frame suggests that the rotational speed does not change significantly over time.

Therefore, the most likely answer is:

'C': 'Constant speed.'

**Claude 3.5 Sonnet:** ased on the series of images provided, I can see a person demonstrating something with a pink object, likely some kind of toy or educational prop. The person is consistently holding and manipulating this object throughout the frames. However, from these static images alone, I cannot definitively determine the rotational speed pattern of the object.

Given the question and available options, I will make my best guess based on the limited information:

The most likely answer seems to be C: Constant speed.

My reasoning is that the person's hand positions and the object's appearance remain fairly consistent across the frames, suggesting a steady, continuous motion rather than acceleration, deceleration, or no movement at all. However, I want to emphasize that this is an educated guess based on limited information from static images, and may not reflect the actual motion in the video.

**Qwen2 VL 72B instruct AWQ:** C

**Qwen2 VL 7B Instruct:** D

**Video-CCAM-v1.1 14B:** Answer is C.

**InternVL 2 40B:** Based on the provided frames, it appears that the object is a pink object being held by a person. The object does not seem to be moving or rotating in any of the frames. Therefore, the most possible answer to the question about the object's rotational speed pattern is:

A: Not moving at all.

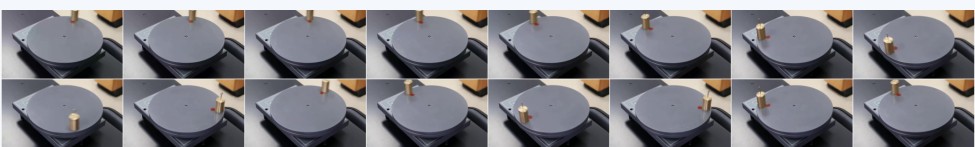

velocity&frequency 151  YouTube

https://www.youtube.com/watch?v=VOOVtEjHqjw

What is the disk's rotational speed pattern?

A. Constant speed.    B. Not moving at all.
C. Decelerating.      D. Accelerating.

Answer: D. Accelerating.

**GPT-4o:** Based on the provided frames, it appears that the disk is slowing down over time. The position of the red mark on the disk changes less between each frame as the sequence progresses, indicating a decrease in rotational speed.

Therefore, the most likely answer is:

Option C: Decelerating.

**Claude 3.5 Sonnet:** Based on analyzing the sequence of 16 frames provided, the disk's rotational speed pattern appears to be accelerating. In the initial frames, the golden cylindrical object on the disk is stationary or moving slowly. As the sequence progresses, the motion of this object becomes more pronounced and blurred, suggesting the disk is rotating faster. By the final frames, the golden object is significantly blurred due to rapid rotation, indicating an increase in speed over time.

Therefore, the most likely answer to the question "What is the disk's rotational speed pattern?" is:

D: Accelerating.

This conclusion is drawn solely from observing the changing position and blur of the golden object on the disk across the provided image sequence, without relying on any external knowledge.

**Qwen2 VL 72B instruct AWQ:** A

**Qwen2 VL 7B Instruct:** A

**Video-CCAM-v1.1 14B:** Answer is A.

**InternVL 2 40B:** Based on the provided frames, the disk appears to be rotating at a constant speed. The position of the object on the disk remains consistent throughout the frames, indicating that the disk is not accelerating, decelerating, or stopping. Therefore, the most likely answer is:

A: Constant speed.

K.2.11  EXAMPLE 11

velocity&frequency 157 YouTube

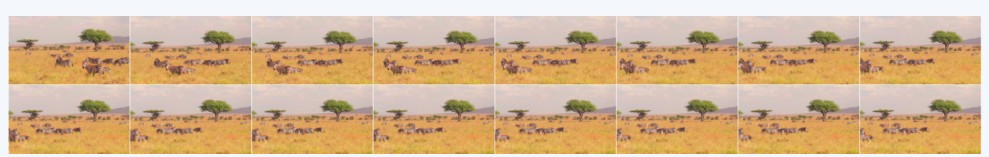

https://www.youtube.com/watch?v=CP5k2gRabvc

What is the trend in the camera's movement speed across the entire video?

A. Not moving at all.  B. Decelerating.
C. Constant speed.     D. Accelerating.

Answer: B. Decelerating.

**GPT-4o:** Based on the 16 frames provided, it appears that the camera is not moving at all. The position of the zebras, trees, and other elements in the scene remain consistent across all frames, indicating that the camera is stationary.

Therefore, the most likely answer is:

'A': 'Not moving at all.'

**Claude 3.5 Sonnet:** Based on analyzing the 16 frames provided, the most likely answer to the question "What is the trend in the camera's movement speed across the entire video?" is:

C: Constant speed.

The reasoning for this conclusion is:

1. The overall composition of the scene remains very consistent across all 16 frames. The positioning of key elements like the trees, zebras, and background landscape stays nearly identical.

2. There is no noticeable change in the level of detail or apparent size of objects in the foreground or background, which would typically occur if the camera was accelerating or decelerating significantly.

3. The zebras' positions relative to each other and to fixed elements like trees show only minor changes between frames, consistent with their natural movement rather than rapid camera motion.

4. The clarity and focus of the images remain constant, suggesting steady camera movement rather than acceleration which might cause motion blur.

5. There are no obvious signs of zooming in or out, panning, or other camera movements that would indicate changing speed.

While it's difficult to be absolutely certain without seeing the actual video, the evidence from these static frames strongly suggests that the camera is moving at a constant speed throughout the captured sequence, making option C the most probable answer.

**Qwen2 VL 72B instruct AWQ:** C

**Qwen2 VL 7B Instruct:** D

**Video-CCAM-v1.1 14B:** Answer is B.

**InternVL 2 40B:** Based on the provided frames, it appears that the camera's movement speed is relatively constant throughout the video. The frames show a consistent view of the zebras and the landscape without any noticeable changes in speed or direction.

Therefore, the most possible answer is:

'C': Constant speed.

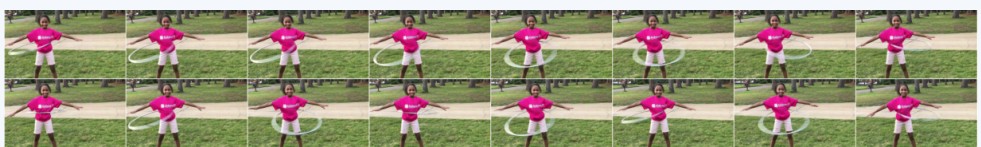

velocity&frequency 159 YouTube

https://www.youtube.com/watch?v=oyvYvruZfmw

What is the hula hoop's rotational speed pattern?

A. Not moving at all. B. Decelerating.
C. Constant speed.    D. Accelerating.

Answer: D. Accelerating.

**GPT-4o:** Based on the provided frames, the hula hoop appears to be rotating at a consistent speed around the person's waist. There is no visible indication of the hula hoop slowing down or speeding up significantly in the sequence of frames.

Therefore, the most likely answer is:

**'C': Constant speed.**

**Claude 3.5 Sonnet:** Based on the series of images provided, I can analyze the hula hoop's rotational speed pattern. The images show a person in a pink shirt using a hula hoop outdoors.

Looking at the sequence of frames, the hula hoop appears to be rotating at a fairly consistent speed throughout. There are no obvious signs of the hoop slowing down or speeding up significantly from frame to frame. The person maintains a steady rhythm and posture, which suggests they are able to keep the hoop spinning at a relatively constant rate.

Given the options provided and based solely on what can be observed in these image frames, the most likely answer to the question "What is the hula hoop's rotational speed pattern?" is:

C: Constant speed.

The hoop seems to maintain a steady rotational speed across the sequence of frames, without any clear indication of acceleration or deceleration. Of course, without more precise measurements or a longer video sequence, it's difficult to rule out minor fluctuations in speed entirely. However, from the visual information provided, a constant speed appears to be the best characterization of the hula hoop's rotational pattern.

**Qwen2 VL 72B instruct AWQ:** C

**Qwen2 VL 7B Instruct:** D

**Video-CCAM-v1.1 14B:** Answer is C.

**InternVL 2 40B:** Based on the provided frames, the hula hoop appears to maintain a consistent motion around the girl's waist. There is no visible change in the speed or pattern of the hoop's rotation. Therefore, the most likely answer is:

'C': Constant speed.

## K.3 SIMULATED

### K.3.1 EXAMPLE 1

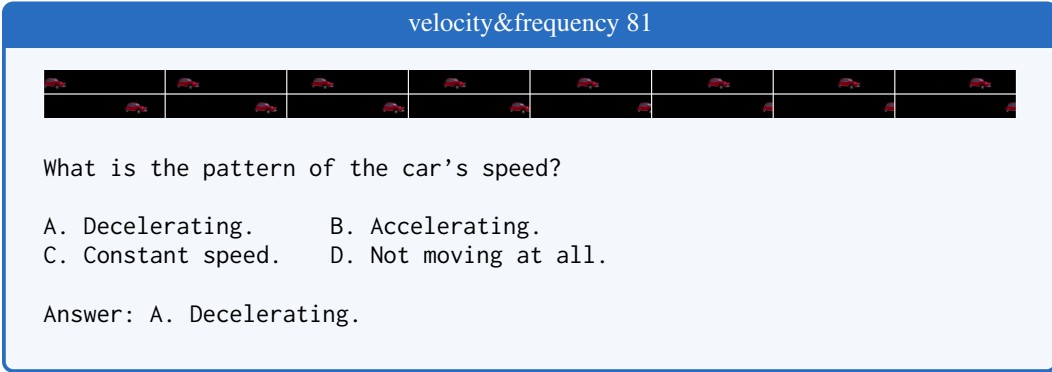

velocity&frequency 81

What is the pattern of the car's speed?

A. Decelerating.     B. Accelerating.
C. Constant speed.   D. Not moving at all.

Answer: A. Decelerating.

---

**GPT-4o:** Based on the provided frames, the car appears to be moving consistently across the frames. The position of the car relative to the frame edges does not show significant changes that would indicate acceleration or deceleration.

Therefore, the most likely answer is:

**C: Constant speed.**

---

**Claude 3.5 Sonnet:** Based on the sequence of frames provided, the most likely answer to the question "What is the pattern of the car's speed?" is:

C: Constant speed.

Reasoning:
1. The red car appears to be moving from left to right across the frames.
2. The distance the car travels between each frame seems to be relatively consistent.
3. There's no apparent change in the car's position that would suggest acceleration or deceleration.
4. The car is clearly moving, so it's not stationary.

While it's difficult to determine the exact speed without more precise measurements, the consistent movement of the car across the frames suggests it's maintaining a steady pace. Therefore, based on the visual information provided, the most likely answer is that the car is moving at a constant speed.

---

**Qwen2 VL 72B instruct AWQ:** C

---

**Qwen2 VL 7B Instruct:** C

---

**Video-CCAM-v1.1 14B:** Answer is A.

---

**InternVL 2 40B:** Based on the frames provided, the car appears to be moving from left to right across the screen. The car's position changes in each frame, indicating that it is moving. However, the frames do not provide enough information to determine the exact pattern of the car's speed (whether it is accelerating, decelerating, or moving at a constant speed).

Given the options provided, the most possible answer would be:

B: Accelerating.

# L  COMMON FAILURE CASES OF SHAPE & TREND

## L.1  HUMAN

### L.1.1  EXAMPLE 1

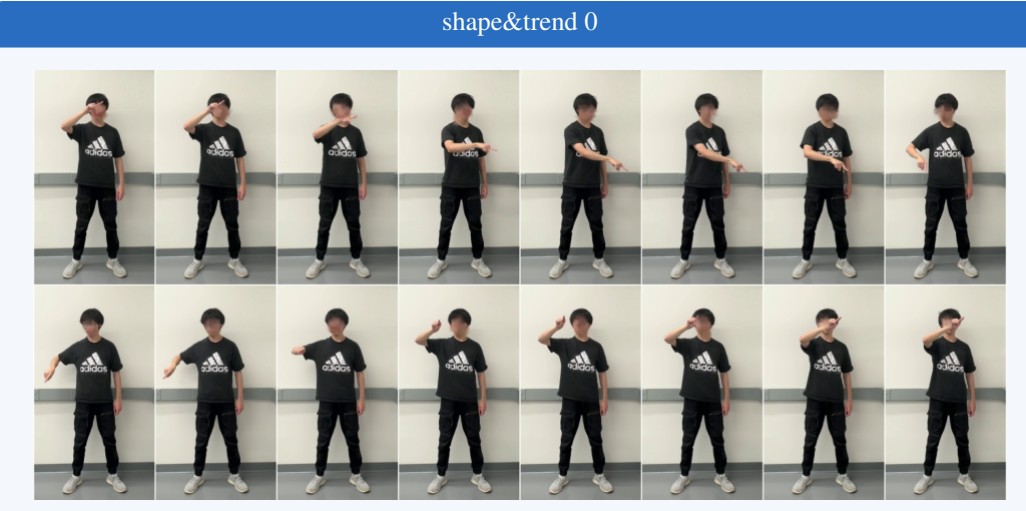

shape&trend 0

What is the shape of the object that the person drew in the air?

A. Trapezoid.
B. Diamond.
C. Square/rectangle.
D. Circle.
E. Triangle.
F. Not drawing at all.

Answer: A. Trapezoid.

**GPT-4o:** Based on the sequence of frames, the person appears to be drawing a shape in the air. The movement of the hand suggests a continuous motion that could form a closed shape.

1. The hand starts at the top (Frame 1).
2. Moves to the right (Frame 3).
3. Moves downwards (Frame 5).
4. Moves to the left (Frame 7).
5. Moves upwards (Frame 9).
6. Returns to the starting point (Frame 11).

This sequence of movements is consistent with drawing a square or rectangle.

Therefore, the most likely answer is:

Option 'C': Square/rectangle.

**Claude 3.5 Sonnet:** Based on the sequence of images provided, it appears the person is not drawing a specific shape in the air. Rather, they are making various gestures with their hand and arm, including pointing upwards, moving their arm horizontally, and positioning their hand near their face. The movements do not seem to trace out any clear geometric shape like those listed in the options.

However, since I'm instructed to provide an answer from the given options, I would guess:

C: Square/rectangle.

This is because some of the arm movements, particularly the horizontal and vertical motions, could potentially be interpreted as tracing out straight lines that form corners, which are characteristic of squares or rectangles. But I want to emphasize that this is a very tentative guess based on limited information, as the images do not clearly show the person drawing any specific shape.

**Qwen2 VL 72B instruct AWQ:** C

**Qwen2 VL 7B Instruct:** B

**Video-CCAM-v1.1 14B:** Answer is B.

**InternVL 2 40B:** Based on the frames provided, the person appears to be drawing a shape in the air with their finger. The shape seems to be a straight line followed by a curve, which is characteristic of a circle. Therefore, the most possible answer is:

'D': Circle.

## L.2  SIMULATED

### L.2.1  EXAMPLE 1



**shape&trend 174**

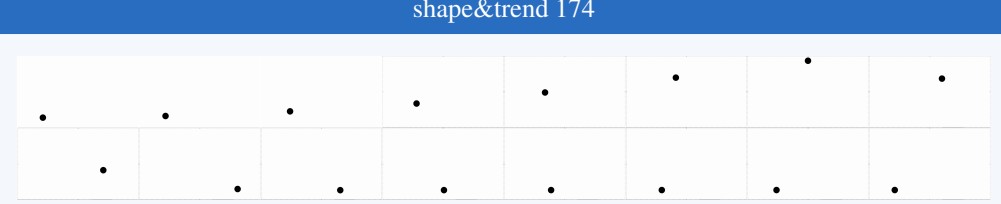

What shape is formed by the black circle's trajectory as it moves?

A. Diamond.    B. Triangle.
C. Circle.     D. Square/rectangle.
E. Trapezoid.  F. Not drawing at all.

Answer: B. Triangle.

</div>

**GPT-4o:** Based on the provided frames, the black circle appears to move in a path that forms a square or rectangle. The circle starts at the bottom left, moves to the top left, then to the top right, and finally to the bottom right before returning to the starting position.

Therefore, the shape formed by the black circle's trajectory is:

**D: Square/rectangle.**

**Claude 3.5 Sonnet:** Based on the sequence of frames provided, the black circle appears to be moving in a circular trajectory. The circle starts in the bottom left corner, moves up to the top center, then down to the bottom right, and finally back towards the bottom left. This circular motion is repeated across the frames.

Therefore, the most likely answer to the question "What shape is formed by the black circle's trajectory as it moves?" is:

C: Circle.

**Qwen2 VL 72B instruct AWQ:** D

**Qwen2 VL 7B Instruct:** C

**Video-CCAM-v1.1 14B:** Answer is A.

**InternVL 2 40B:** The black circle's trajectory forms a shape that resembles a diamond.

Answer: A

### shape&trend 175

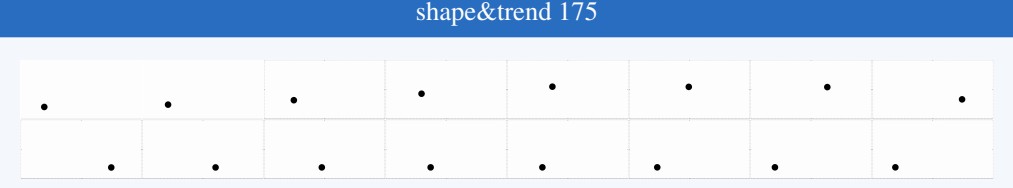

```
What shape is formed by the black circle's trajectory as it moves?

A. Diamond.    B. Not drawing at all.
C. Trapezoid.  D. Circle.
E. Triangle.   F. Square/rectangle.

Answer: C. Trapezoid.
```

**GPT-4o:** Based on the provided frames, the black circle appears to be moving in a pattern that suggests it is tracing out a circular path. The circle starts at the bottom, moves to the left, then upwards, to the right, and back down, completing a loop.

Therefore, the most likely answer is:

**'D': Circle.**

**Claude 3.5 Sonnet:** Based on the sequence of frames provided, the black circle appears to be moving in a circular pattern. The circle starts in the bottom left corner, moves to the right, then up towards the center, then to the left, and finally back down towards the starting position. This movement pattern suggests that the shape formed by the black circle's trajectory is most likely a circle.

Therefore, the answer to the question "What shape is formed by the black circle's trajectory as it moves?" is:

D. Circle.

**Qwen2 VL 72B instruct AWQ:** D

**Qwen2 VL 7B Instruct:** The black circle's trajectory forms a circle.

**Video-CCAM-v1.1 14B:** Answer is D.

**InternVL 2 40B:** The black circle's trajectory forms a shape that resembles a diamond.

Answer: A

### shape&trend 182

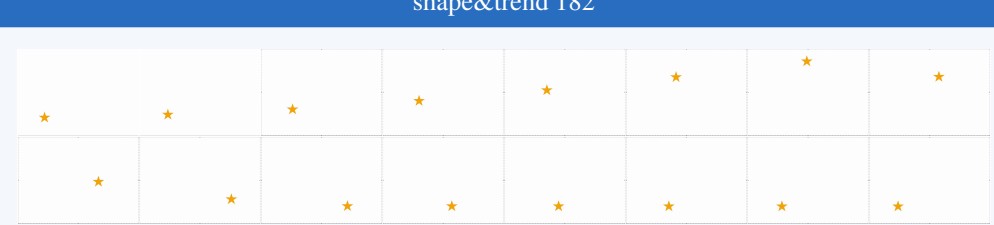

What shape is formed by the yellow star's trajectory as it moves?

A. Diamond.    B. Triangle.
C. Circle.     D. Square/rectangle.
E. Not drawing at all. F. Trapezoid.

Answer: B. Triangle.

**GPT-4o:** Based on the provided frames, the yellow star appears to move in a pattern that forms a closed shape. The star starts at the bottom left, moves up, then to the right, down, and back to the left, completing a loop. This movement suggests a shape with four sides.

Given the options, the most likely shape formed by the yellow star's trajectory is:

**Option D: Square/rectangle.**

**Claude 3.5 Sonnet:** Based on the sequence of images provided, the yellow star appears to be moving in a circular trajectory. The star's position changes gradually across the frames, moving from the bottom left, to the center, to the top right, and then back towards the bottom left. This circular motion is consistent throughout the 16 frames.

Therefore, the most likely answer to the question "What shape is formed by the yellow star's trajectory as it moves?" is:

C: Circle.

**Qwen2 VL 72B instruct AWQ:** C

**Qwen2 VL 7B Instruct:** C

**Video-CCAM-v1.1 14B:** Answer is A.

**InternVL 2 40B:** The yellow star's trajectory forms a shape that resembles a diamond.

Answer: A

---

### shape&trend 180

What shape is formed by the black circle's trajectory as it moves?

A. Circle.      B. Infinity-shape.
C. Square/rectangle.  D. Trapezoid.
E. Diamond.    F. Not drawing at all.

Answer: B. Infinity-shape.

---

**GPT-4o:** Based on the provided frames, the black circle appears to be moving in a pattern that forms a closed loop. The circle moves from the bottom center to the left, then to the top, then to the right, and back to the bottom center. This pattern suggests a circular trajectory.

Therefore, the most likely answer is:

**'A': Circle.**

---

**Claude 3.5 Sonnet:** Based on analyzing the 16 frames provided, which show the position of a black circle at different points in time, I can make an educated guess about the shape of the circle's trajectory.

The black circle appears to move in a roughly circular or oval path around the center of the frame. It starts in the upper left, moves to the right, then down to the lower right, back to the left, and up again. This circular motion is repeated across multiple frames.

Given the options provided, the most likely answer is:

A: Circle.

The trajectory forms an approximately circular shape as the black circle moves around the frame in a loop. While it may not be a perfect circle, this option best matches the observed motion compared to the other choices given.

---

**Qwen2 VL 72B instruct AWQ:** A

---

**Qwen2 VL 7B Instruct:** B

---

**Video-CCAM-v1.1 14B:** Answer is A.

---

**InternVL 2 40B:** Based on the provided frames, the black circle appears to move in a way that forms a shape that resembles a diamond. The trajectory of the circle's movement creates the outline of a diamond shape.

Therefore, the correct answer is:

'E': Diamond.

L.2.5 EXAMPLE 5

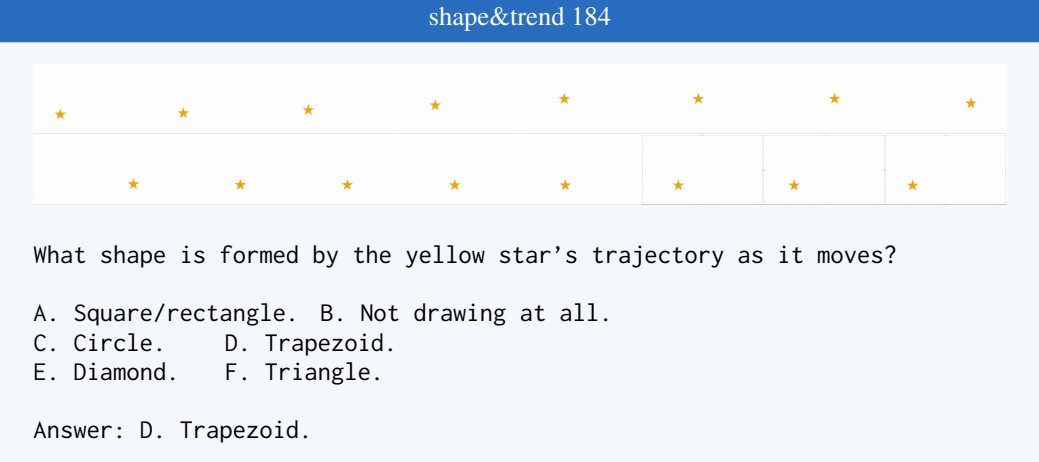

## shape&trend 184

What shape is formed by the yellow star's trajectory as it moves?

A. Square/rectangle.  B. Not drawing at all.
C. Circle.     D. Trapezoid.
E. Diamond.    F. Triangle.

Answer: D. Trapezoid.

**GPT-4o:** Based on the provided frames, the yellow star appears to be moving in a circular trajectory. The star starts at the bottom left, moves to the top left, then to the top right, and finally to the bottom right, before returning to the bottom left. This pattern suggests a circular motion.

Therefore, the most likely answer is:

**'C': Circle.**

**Claude 3.5 Sonnet:** Based on analyzing the 16 frames provided, the most likely answer to the question "What shape is formed by the yellow star's trajectory as it moves?" is:

C. Circle.

The yellow star appears to move in a circular pattern across the frames. It starts in the bottom left corner, moves upward and to the right, then back down to the left, tracing what seems to be a circular trajectory. While we don't see the complete circle in every frame, the overall movement pattern strongly suggests a circular path.

This circular motion is the most consistent with the star's positions across the different frames, compared to the other shape options provided. The star does not appear to move in straight lines that would form a square, rectangle, trapezoid, diamond, or triangle. The smooth curved motion is most indicative of a circular trajectory.

**Qwen2 VL 72B instruct AWQ:** C

**Qwen2 VL 7B Instruct:** C

**Video-CCAM-v1.1 14B:** Answer is E.

**InternVL 2 40B:** The yellow star's trajectory forms a shape that resembles a circle.

Answer: C

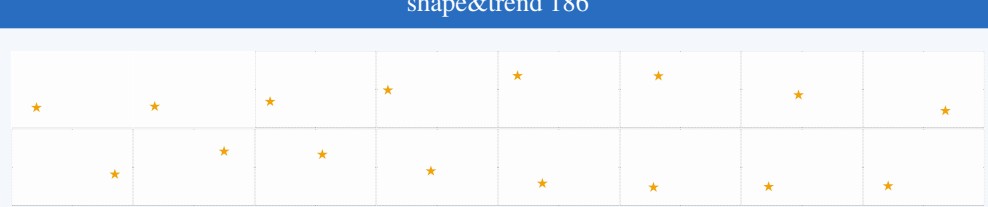

shape&trend 186

What shape is formed by the yellow star's trajectory as it moves?

A. Triangle.   B. Infinity-shape.
C. Not drawing at all. D. Circle.
E. Trapezoid. F. Diamond.

Answer: B. Infinity-shape.

**GPT-4o:** Based on the provided frames, the yellow star appears to be moving in a circular pattern. The star starts at the bottom left, moves up, then to the right, down, and back to the left, completing a loop.

Therefore, the shape formed by the yellow star's trajectory as it moves is:

**Option D: Circle.**

**Claude 3.5 Sonnet:** Based on the sequence of frames provided, the yellow star appears to be moving in a circular pattern. The star's position changes gradually from the bottom left corner, to the center, to the top right, and then back towards the bottom left, completing what seems to be a circular trajectory.

Therefore, the most likely answer to the question "What shape is formed by the yellow star's trajectory as it moves?" is:

D. Circle.

The star's movement pattern across the frames strongly suggests a circular path, making this the most plausible option among the given choices.

**Qwen2 VL 72B instruct AWQ:** A

**Qwen2 VL 7B Instruct:** The yellow star's trajectory forms a circle.

**Video-CCAM-v1.1 14B:** Answer is A.

**InternVL 2 40B:** The yellow star's trajectory forms a shape that resembles a triangle.

Answer: A

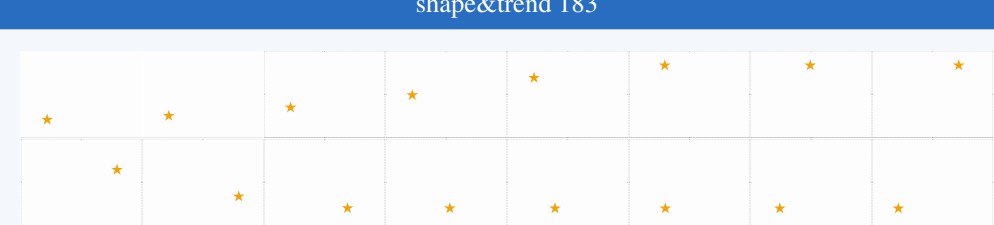

### shape&trend 183

What shape is formed by the yellow star's trajectory as it moves?

A. Not drawing at all. B. Diamond.
C. Circle.    D. Trapezoid.
E. Triangle.  F. Square/rectangle.

Answer: F. Square/rectangle.

**GPT-4o:** Based on the provided frames, the yellow star appears to move in a pattern that forms a square or rectangle. The star starts at the bottom left, moves up to the top left, then to the top right, down to the bottom right, and finally back to the bottom left.

Therefore, the shape formed by the yellow star's trajectory is:

**Option F: Square/rectangle.**

**Claude 3.5 Sonnet:** Based on the sequence of images provided, the yellow star appears to be moving in a circular trajectory. The star's position changes across the frames, starting from the bottom left, moving to the top center, then to the top right, and finally back towards the bottom left. This movement pattern suggests a circular motion.

Therefore, the most likely answer to the question "What shape is formed by the yellow star's trajectory as it moves?" is:

C. Circle.

**Qwen2 VL 72B instruct AWQ:** C

**Qwen2 VL 7B Instruct:** The yellow star's trajectory forms a diamond shape.

**Video-CCAM-v1.1 14B:** Answer is B.

**InternVL 2 40B:** The yellow star's trajectory forms a shape that resembles a diamond.

Answer: B

# M   COMMON FAILURE CASES OF VISUAL CUES

## M.1   OBJECT

### M.1.1   EXAMPLE 1

<div>

**visual cues 0  Music-AVQA**

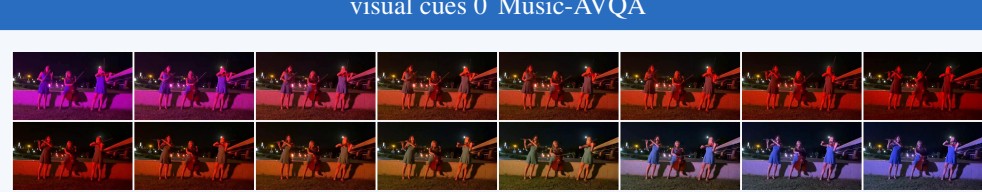

Which musical instrument sounds last?

A. Cello.      B. None of them produces any sound.
C. Flute.      D. Violin.
E. All instruments sound at the same time.

Answer: C. Flute.

</div>

**GPT-4o:** Based on the provided frames, it appears that the three musicians are playing their instruments simultaneously throughout the video. There is no clear indication that any one instrument stops playing before the others. Therefore, the most plausible answer based on the frames is:

**Option E: All instruments sound at the same time.**

**Claude 3.5 Sonnet:** Based on the series of images provided, I can see three musicians performing together outdoors at night. They are playing three different instruments: a flute, a cello, and a violin. Throughout the sequence of images, all three musicians appear to be actively playing their instruments simultaneously.

Given the question "Which musical instrument sounds last?" and the provided options, the most likely answer based on the visual information would be:

E: All instruments sound at the same time.

This is because in all frames, the three musicians are shown in playing positions with their instruments, suggesting they are performing together as an ensemble. There's no visual indication that any instrument stops playing before the others or continues after the others have finished. Therefore, the most reasonable assumption from the given information is that all instruments are sounding simultaneously throughout the performance.

**Qwen2 VL 72B instruct AWQ:** A

**Qwen2 VL 7B Instruct:** A

**Video-CCAM-v1.1 14B:** Answer is A.

**InternVL 2 40B:** Based on the provided frames, it is not possible to determine which musical instrument sounds last. The frames show three women playing different instruments (flute, cello, and violin), but they do not provide information about the timing of their sounds. Therefore, the most possible answer is:

'E': All instruments sound at the same time.

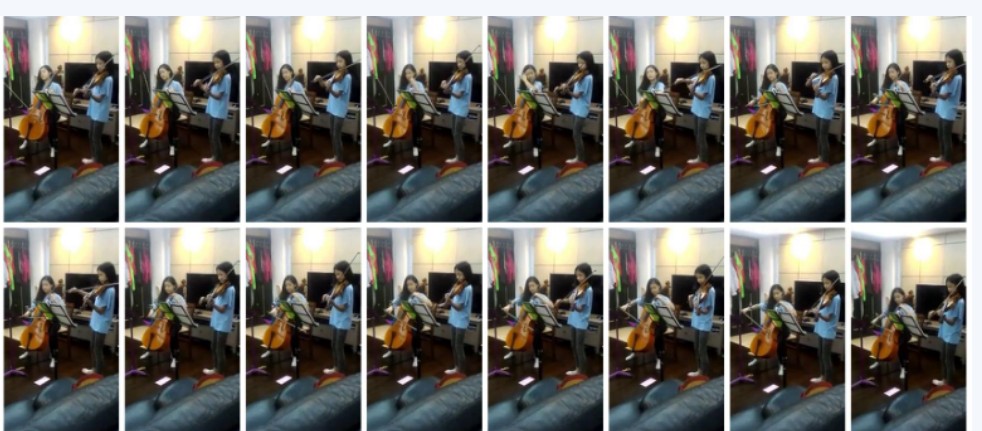

visual cues 28 Music-AVQA

Which musical instrument plays first?

A. Cello.     B. Violin.
C. Both instruments begin simultaneously.
D. Neither instrument produces any sound.

Answer: B. Violin.

**GPT-4o:** Based on the provided frames, both the cello and the violin players appear to be preparing to play their instruments. There is no clear indication from the frames alone which instrument starts playing first. However, given that both musicians are in position and appear ready to play, it is reasonable to guess that they might begin simultaneously.

Therefore, the most plausible answer based on the frames is:

**Option C: Both instruments begin simultaneously.**

**Claude 3.5 Sonnet:** Based on the series of images provided, it's difficult to determine with certainty which instrument plays first. However, I'll make my best guess based on the visual information available.

The images show two individuals, one playing a cello and the other playing a violin. They appear to be in a practice or performance setting, with music stands in front of them. Both musicians seem to be in playing positions throughout the sequence of images.

Given that both musicians appear ready to play from the beginning of the sequence, and there's no clear indication of one starting before the other, my best guess would be:

C: Both instruments begin simultaneously.

This answer seems most plausible because both musicians are in playing position throughout the sequence, suggesting they might be performing a duet or playing together. Without audio information or more detailed visual cues about who starts first, it's reasonable to assume they begin playing at the same time.

**Qwen2 VL 72B instruct AWQ:** A

**Qwen2 VL 7B Instruct:** A

**Video-CCAM-v1.1 14B:** Answer is A.

**InternVL 2 40B:** Based on the provided frames, it appears that both the cello and the violin are being played simultaneously. Therefore, the most possible answer is:

C: Both instruments begin simultaneously.

M.1.3  EXAMPLE 3

A. Both instruments begin simultaneously.   B. Flute.
C. Ukulele.     D. Neither instrument produces any sound.

**GPT-4o:** Based on the provided frames, it appears that both the flute player and the ukulele player are engaged in playing their instruments throughout the sequence. There is no clear indication that one starts before the other. Therefore, the most plausible answer based on the frames is:

**Option A: Both instruments begin simultaneously.**

**Claude 3.5 Sonnet:** Based on the images provided, I would guess that the answer to the question "Which musical instrument plays first?" is:

C. Ukulele.

My reasoning is that in most of the frames, both musicians appear to be actively playing their instruments simultaneously. However, in the last two frames, we see a close-up of just the ukulele player's hands on the instrument, suggesting a focus on that instrument. This could indicate that the ukulele starts the piece or has a solo introduction before the flute joins in.

While I can't be certain without hearing the audio, the visual emphasis on the ukulele in those final frames leads me to believe it's the most likely instrument to play first in this musical performance.

**Qwen2 VL 72B instruct AWQ:** C

**Qwen2 VL 7B Instruct:** C

**Video-CCAM-v1.1 14B:** Answer is C.

**InternVL 2 40B:** Based on the frames provided, it appears that both the flute and the ukulele are being played simultaneously. Therefore, the most possible answer is:

A: Both instruments begin simultaneously.

---

**visual cues 38  Music-AVQA**

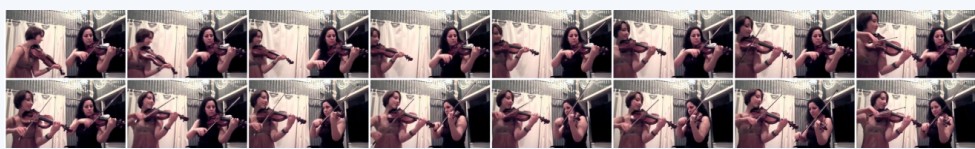

Which violin is the first sounding instrument?

A. Both violins sound at the same time. B. The right one.
C. The left one.     D. Neither instrument produces any sound.

Answer: B. The right one.

---

**GPT-4o:** Based on the provided frames, it appears that both violinists are playing their instruments simultaneously throughout the sequence. There is no clear indication that one starts playing before the other. Therefore, the most plausible answer based on the frames is:

'A': 'Both violins sound at the same time.'

---

**Claude 3.5 Sonnet:** Based on the series of images provided, it's difficult to determine with certainty which violin is the first sounding instrument. However, I'll make my best guess based on the visual information available.

From the images, we can see two violinists playing together throughout the sequence. They both appear to be actively playing their instruments in all frames, suggesting they might be playing simultaneously or very close to each other in timing.

Given that we need to choose an answer, and considering that the question asks specifically about which violin sounds first, I would guess:

A: Both violins sound at the same time.

This answer seems most plausible based on the visual evidence, as both violinists appear to be engaged in playing their instruments from the very first frame. There's no clear indication that one starts before the other. While it's impossible to hear the sound from these images alone, the synchronized movements of both players suggest they might be starting together.

---

**Qwen2 VL 72B instruct AWQ:** C

---

**Qwen2 VL 7B Instruct:** C

---

**Video-CCAM-v1.1 14B:** Answer is C.

---

**InternVL 2 40B:** Based on the frames provided, it is not possible to determine which violin is the first sounding instrument. The frames show both women playing the violin simultaneously, and there is no clear indication of which one started playing first.

---

**visual cues 1 Music-AVQA**

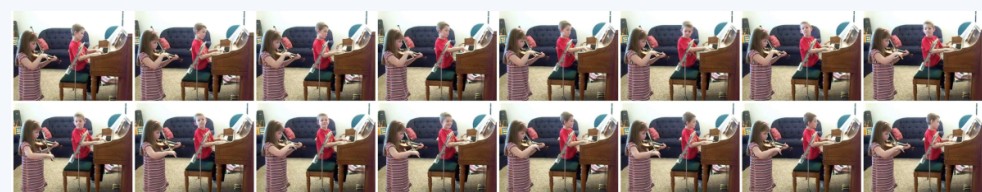

Which musical instrument plays first?

A. Neither instrument produces any sound.
B. Both instruments begin simultaneously.
C. Violin.    D. Piano.

Answer: D. Piano.

---

**GPT-4o:** Based on the provided frames, both the boy playing the piano and the girl playing the violin appear to be engaged in playing their instruments simultaneously. There is no clear indication that one starts before the other. Therefore, the most plausible answer based on the frames is:

**Option B: Both instruments begin simultaneously.**

---

**Claude 3.5 Sonnet:** Based on the sequence of images provided, it's difficult to determine with certainty which instrument plays first. However, I'll make my best guess based on the information available in the frames.

The images show two children, one playing a violin and the other playing a piano. Both appear to be in playing positions throughout the sequence. There's no clear indication of one instrument starting before the other.

Given that both children seem ready to play from the first frame, and their postures remain consistent throughout, I would guess that the most likely answer is:

B: Both instruments begin simultaneously.

This answer assumes that the children start playing together, as they both appear prepared to play from the beginning of the sequence. However, it's important to note that without audio or more detailed visual cues, this is an educated guess based on the limited information provided in the image frames.

---

**Qwen2 VL 72B instruct AWQ:** D

---

**Qwen2 VL 7B Instruct:** C

---

**Video-CCAM-v1.1 14B:** Answer is C.

---

**InternVL 2 40B:** Based on the provided frames, it is not possible to determine which musical instrument plays first. The frames show both the girl playing the violin and the boy playing the piano, but they do not provide enough information to determine the sequence of their playing.



**visual cues 7  Music-AVQA**

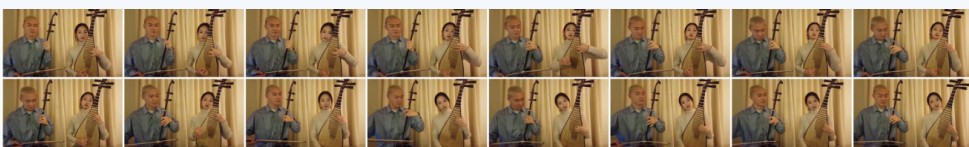

```
Which musical instrument plays first?

A. Neither instrument produces any sound.   B. Pipa.
C. Erhu.      D. Both instruments begin simultaneously.

Answer: B. Pipa.
```
</div>

**GPT-4o:** Based on the provided frames, it appears that the person playing the Erhu (on the left) starts moving their bow before the person playing the Pipa (on the right) starts plucking the strings. This suggests that the Erhu might be the first instrument to produce sound.

Therefore, the most likely answer is:

Option C: Erhu.

**Claude 3.5 Sonnet:** Based on the series of images provided, I'll try my best to answer the question about which musical instrument plays first. However, it's important to note that these are still images, so we can't actually hear any sound or determine the exact order of play. I'll make an educated guess based on the visual cues in the frames.

Looking at the images, we can see two musicians holding traditional Chinese instruments. The person on the left is holding an erhu (a two-stringed bowed instrument), while the person on the right is holding a pipa (a four-stringed plucked instrument).

Throughout the sequence of images, both musicians appear to be in playing positions, holding their instruments ready to perform. There's no clear indication that one starts before the other based solely on these visual cues.

Given the question and the options provided, and considering that both musicians seem equally prepared to play in all frames, I would guess:

D: Both instruments begin simultaneously.

This answer assumes that since both musicians are consistently shown in playing positions, they likely start playing their instruments at the same time. However, I want to emphasize that this is an educated guess based on limited visual information, and without audio, it's impossible to determine with certainty which instrument actually plays first.

**Qwen2 VL 72B instruct AWQ:** C

**Qwen2 VL 7B Instruct:** C

**Video-CCAM-v1.1 14B:** Answer is B.

**InternVL 2 40B:** Based on the provided frames, it is not possible to determine which musical instrument plays first. The frames do not provide any information about the sequence of the instruments playing.

---

**visual cues 14  Music-AVQA**

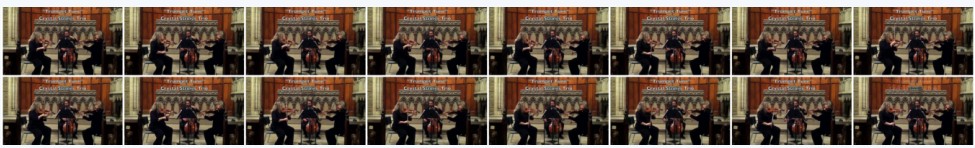

```
Which musical instrument finishes the song?

A. None of them produces any sound. B. Cello on the middle.
C. All instruments sound at the same time. D. Violin on the left.
E. Violin on the right.

Answer: B. Cello on the middle.
```

---

**GPT-4o:** Based on the 16 frames provided, it appears that the video features a trio of musicians: two violinists (one on the left and one on the right) and a cellist in the middle. The frames show the musicians playing their instruments throughout the video.

Given the options and the visual information from the frames, the most plausible answer is:

**Option D: Violin on the left.**

This is a guess based on the visual cues, as the frames do not provide explicit information about which instrument finishes the song.

---

**Claude 3.5 Sonnet:** Based on the provided frames, I will make my best guess to answer the question "Which musical instrument finishes the song?"

From analyzing the 16 frames, we can see a string trio performing - two violins (one on the left and one on the right) and a cello in the middle. All three musicians appear to be actively playing their instruments throughout the sequence of images.

Since the question asks specifically about which instrument finishes the song, and we don't have audio information or a clear indication of when the piece ends, I'll make an educated guess based on common musical practices and the visual cues available.

In many musical compositions, especially for string ensembles, it's common for all instruments to play until the end, often finishing together on a final chord or note. However, if I had to choose one instrument that might be more likely to have the final note or phrase, I would guess:

E: Violin on the right.

This is because:
1. Violins often carry the melody in string ensembles.
2. The right-side violinist seems to be in a slightly more prominent position in the frame.
3. In some compositions, the highest voice (which would typically be a violin) might have the final note or flourish.

However, I want to emphasize that this is a guess based on limited visual information, and without audio or more specific cues about the end of the piece, it's impossible to be certain.

---

**Qwen2 VL 72B instruct AWQ:** E

---

**Qwen2 VL 7B Instruct:** C

---

**Video-CCAM-v1.1 14B:** Answer is B.

**InternVL 2 40B:** Based on the given frames, it is not possible to determine which musical instrument finishes the song. The frames show the Crystal Strings Trio playing their instruments together, and there is no clear indication of which instrument finishes the song. Therefore, the most possible answer is:

'C': All instruments sound at the same time.



**visual cues 30  Music-AVQA**

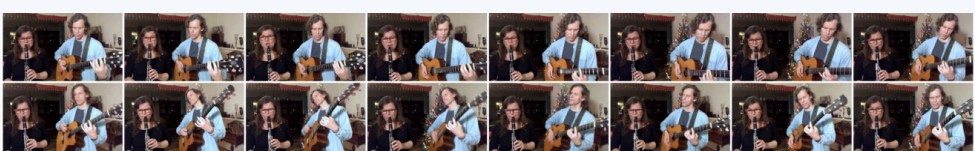

```
Which musical instrument plays first?

A. Guitar.    B. Both instruments begin simultaneously.
C. Clarinet.  D. Neither instrument produces any sound.

Answer: A. Guitar.
```

</div>

**GPT-4o:** Based on the provided frames, both the clarinet player and the guitarist appear to be playing their instruments simultaneously in all frames. There is no clear indication that one instrument starts before the other.

Given this observation, the most plausible answer is:

**'B': Both instruments begin simultaneously.**

**Claude 3.5 Sonnet:** Based on the series of images provided, it appears that both musicians are actively playing their instruments throughout the frames. The woman is consistently shown playing the clarinet, while the man is playing the guitar. Since both musicians seem to be engaged in playing simultaneously across all frames, and there's no clear indication of one instrument starting before the other, the most likely answer to the question "Which musical instrument plays first?" is:

B: Both instruments begin simultaneously.

This conclusion is drawn from the consistent positioning and active playing stance of both musicians across all the frames, suggesting they are performing together from the beginning of the captured sequence.

**Qwen2 VL 72B instruct AWQ:** A

**Qwen2 VL 7B Instruct:** C

**Video-CCAM-v1.1 14B:** Answer is C.

**InternVL 2 40B:** Based on the frames provided, it appears that both the guitar and the clarinet are being played simultaneously. Therefore, the most possible answer is:

B: Both instruments begin simultaneously.

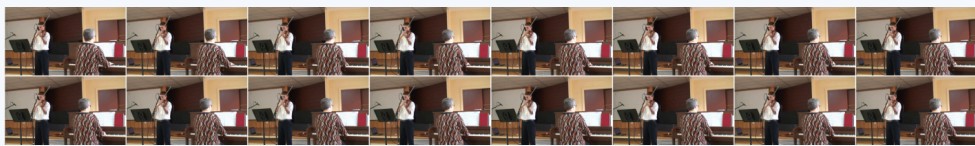

**visual cues 55  Music-AVQA**

Which musical instrument plays first?

A. Piano.     B. Both instruments begin simultaneously.
C. Neither instrument produces any sound.  D. Violin.

Answer: B. Both instruments begin simultaneously.

**GPT-4o:** Based on the provided frames, both the violinist and the pianist are in position and appear to be playing their instruments. However, there is no clear indication from the frames alone which instrument starts playing first. Given the context of a typical musical performance, it is common for the piano to provide an introduction or accompaniment before the violin joins in.

Therefore, the most plausible answer based on the frames is:

'A': Piano.

**Claude 3.5 Sonnet:** Based on the images provided, the most likely answer to the question "Which musical instrument plays first?" is:

D. Violin.

The reasoning for this answer is:

1.  Throughout all the frames, we see a person actively playing the violin. They are holding the violin and moving the bow across the strings.

2.  The person at the piano is seated but does not appear to be actively playing in any of the frames. Their hands are not visible on the keys.

3.  The violinist seems to be the main focus of the performance, standing in front of a music stand and microphone.

4.  The consistent positioning and active playing of the violin across all frames suggests it is the primary instrument being played in this scene.

While we cannot hear the audio to confirm definitively, based on the visual evidence provided in these frames, it appears that the violin is the instrument playing first or primarily in this performance setting.

**Qwen2 VL 72B instruct AWQ:** A

**Qwen2 VL 7B Instruct:** D

**Video-CCAM-v1.1 14B:** Answer is D.

**InternVL 2 40B:** Based on the frames provided, it is not possible to determine which musical instrument plays first. The video captures a continuous performance where both the violin and piano are being played simultaneously. Therefore, the most possible answer is:

B: Both instruments begin simultaneously.

---

### visual cues 62 Music-AVQA

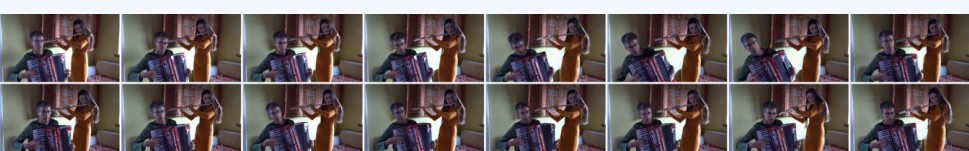

```
Which musical instrument plays first?

A. Accordion.  B. Flute.
C. Neither instrument produces any sound.
D. Both instruments begin simultaneously.

Answer: D. Both instruments begin simultaneously.
```

---

**GPT-4o:** Based on the provided frames, it appears that both the accordion player and the flute player are preparing to play their instruments. However, there is no clear indication from the frames alone which instrument starts playing first.

Given the options and the visual cues, I will make an educated guess:

- The accordion player seems to be actively pressing the keys and moving the bellows in the initial frames.
- The flute player is holding the flute in a playing position but does not show clear signs of starting to play first.

Therefore, the most plausible answer based on the frames is:

**Option A: Accordion.**

---

**Claude 3.5 Sonnet:** Based on the sequence of images provided, I would guess that the answer to the question "Which musical instrument plays first?" is:

A: Accordion.

My reasoning for this guess is:

1. Throughout all the frames, we see a man holding and seemingly playing an accordion. His posture and hand positions on the instrument remain consistent, suggesting he is actively playing it.

2. The woman with the flute appears to be in a playing position, but her posture and the position of the flute don't change significantly across the frames. This could indicate she might be preparing to play or joining in later.

3. The accordion player seems more engaged with his instrument from the beginning, which might suggest he starts playing first.

4. While we can't hear the audio, the consistent positioning of the accordion player across all frames gives the impression that he might have begun playing first or is the lead musician in this scenario.

Without being able to hear the audio or see more detailed motion, it's impossible to be certain. However, based on the visual cues in the provided frames, the accordion seems the most likely instrument to have started playing first.

---

**Qwen2 VL 72B instruct AWQ:** A

---

**Qwen2 VL 7B Instruct:** A

---

**Video-CCAM-v1.1 14B:** Answer is A.

**InternVL 2 40B:** Based on the provided frames, it is not possible to determine which musical instrument plays first. Both the accordion and the flute are being played simultaneously in the video. Therefore, the most possible answer is:

D: Both instruments begin simultaneously.

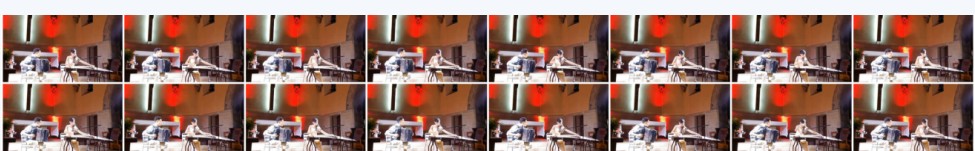

visual cues 63  Music-AVQA

```
Which musical instrument plays first?

A. Both instruments begin simultaneously.
B. Neither instrument produces any sound.
C. Guzheng.    D. Accordion.

Answer: A. Both instruments begin simultaneously.
```

**GPT-4o:** Based on the provided frames, it appears that the person playing the Guzheng is actively plucking the strings, while the person with the accordion is holding the instrument but not yet playing it. This suggests that the Guzheng might be the first instrument to produce sound.

Therefore, the most likely answer is:

**'C': Guzheng.**

**Claude 3.5 Sonnet:** Based on the provided frames, it's difficult to determine with certainty which instrument plays first. However, I'll make my best guess based on the information available in the images.

The frames show two musicians, one playing an accordion and the other playing what appears to be a guzheng (a Chinese plucked string instrument). Both musicians seem to be in performance positions throughout the sequence of images.

Given that the question asks which instrument plays first, and considering that both musicians appear ready to play in all frames, I would guess:

C: Guzheng.

My reasoning for this guess is that the guzheng player's posture seems slightly more active in the earlier frames, with their hands positioned over the strings, suggesting they might be about to play or have just started playing. The accordion player's posture remains relatively consistent throughout the frames.

However, I want to emphasize that this is a guess based on limited visual information, as we can't hear the audio or see the exact moment when each instrument begins playing.

**Qwen2 VL 72B instruct AWQ:** D

**Qwen2 VL 7B Instruct:** D

**Video-CCAM-v1.1 14B:** Answer is D.

**InternVL 2 40B:** Based on the provided frames, it is not possible to determine which musical instrument plays first. The frames show both the accordion and the guzheng being played simultaneously, without any clear indication of which one started first. Therefore, the most possible answer is:

'A': 'Both instruments begin simultaneously.'

## visual cues 3  Music-AVQA

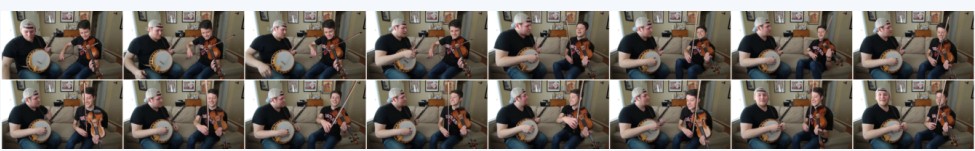

```
Which instrument plays first?

A. Both instruments begin simultaneously.   B. Banjo.
C. Violin.     D. Neither instrument produces any sound.

Answer: C. Violin.
```

**GPT-4o:** Based on the provided frames, it appears that the person with the violin is preparing to play first. In the initial frames, the person with the violin is seen positioning the bow and the violin, while the person with the banjo is not yet actively playing.

Therefore, the most likely answer is:

Option C: Violin.

**Claude 3.5 Sonnet:** Based on the sequence of images provided, I would guess that the answer to the question "Which instrument plays first?" is:

B. Banjo.

My reasoning is as follows:

1. In the early frames, both musicians are holding their instruments (banjo and violin) but not actively playing.

2. As the sequence progresses, the banjo player appears to become more animated and engaged first. His posture changes and he seems to be moving his hands on the instrument in a playing motion.

3. The violin player's posture and engagement seems to follow slightly after the banjo player in the sequence of images.

4. In the later frames, both musicians appear to be actively playing and singing/vocalizing together.

While it's impossible to be certain without audio, the visual cues suggest that the banjo player initiates the performance, with the violin joining in shortly after. Therefore, my best guess based solely on these images is that the banjo plays first.

**Qwen2 VL 72B instruct AWQ:** B

**Qwen2 VL 7B Instruct:** B

**Video-CCAM-v1.1 14B:** Answer is B.

**InternVL 2 40B:** Based on the frames provided, it appears that both the banjo and the violin are being played simultaneously. Therefore, the most possible answer is:

A: Both instruments begin simultaneously.

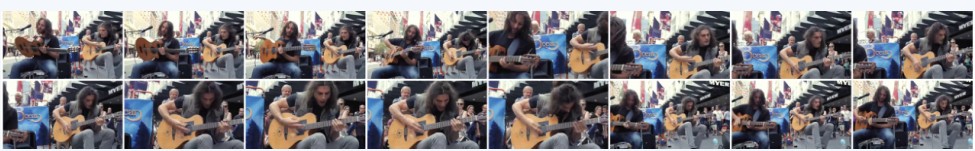

visual cues 19 Music-AVQA

Which guitar is the first sounding instrument?

A. The left one.      B. Neither instrument produces any sound.
C. Both guitars sound at the same time. D. The right one.

Answer: D. The right one.

**GPT-4o:** Based on the provided frames, it appears that both guitarists are actively playing their instruments. However, the guitarist on the right seems to be more animated and engaged in playing, suggesting that he might be the first to produce sound.

Therefore, the most likely answer is:

**D: The right one.**

**Claude 3.5 Sonnet:** Based on the sequence of images provided, it appears that both guitarists are playing their instruments simultaneously throughout the video. From the first frame to the last, we can see both musicians actively engaged with their guitars, suggesting they are performing together.

While it's impossible to determine the exact timing of when each instrument first produces sound from these still images alone, the visual evidence suggests that both guitarists are playing concurrently. Therefore, the most likely answer to the question "Which guitar is the first sounding instrument?" is:

C: Both guitars sound at the same time.

This answer assumes that since both guitarists are shown playing from the beginning, they likely started their performance together. Without audio information, this seems to be the most reasonable conclusion based on the visual information provided in the frames.

**Qwen2 VL 72B instruct AWQ:** A

**Qwen2 VL 7B Instruct:** A

**Video-CCAM-v1.1 14B:** Answer is A.

**InternVL 2 40B:** Based on the frames provided, it appears that both guitars are being played simultaneously. Therefore, the most likely answer is:

'C': Both guitars sound at the same time.

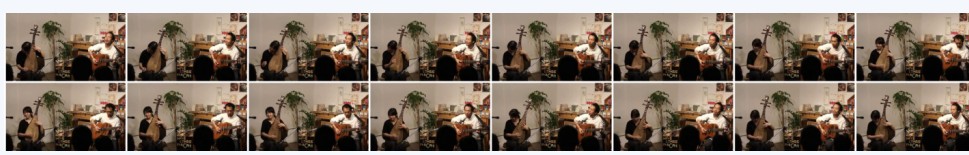

visual cues 25 Music-AVQA

Which musical instrument plays first?

A. Guitar.     B. Neither instrument produces any sound.
C. Pipa.       D. Both instruments begin simultaneously.

Answer: A. Guitar.

**GPT-4o:** Based on the provided frames, it appears that both musicians are preparing to play their instruments. The person with the guitar is seen singing and strumming the guitar, while the person with the pipa is adjusting and plucking the strings.

Given the visual cues, it seems that the person with the guitar is more actively engaged in playing and singing, suggesting that the guitar might be the first instrument to produce sound.

Therefore, the most likely answer is:

**'A': Guitar.**

**Claude 3.5 Sonnet:** Based on the series of images provided, I would guess that the answer to the question "Which musical instrument plays first?" is:

C. Pipa.

My reasoning for this guess is:

1. The images show two musicians, one playing a pipa (a Chinese lute-like instrument) and the other playing a guitar.

2. In several of the early frames, the pipa player appears to be more actively engaged with their instrument, holding it in a playing position and seeming to strum or pluck the strings.

3. The guitar player, while holding the instrument, doesn't seem to be as actively playing in the initial frames.

4. There's a progression in the images where both musicians become more animated, suggesting they both eventually play.

5. Given that the pipa player seems more ready to play in the earlier frames, it's reasonable to guess that this instrument might have started first.

However, it's important to note that this is an educated guess based solely on the visual information provided in the still frames. Without audio or more detailed timing information, it's impossible to be certain which instrument actually played first.

**Qwen2 VL 72B instruct AWQ:** C

**Qwen2 VL 7B Instruct:** C

**Video-CCAM-v1.1 14B:** Answer is C.

**InternVL 2 40B:** Based on the provided frames, it is not possible to determine which musical instrument plays first. The frames show both the guitar and the pipa being played simultaneously, without any clear indication of which one started first.

Therefore, the most possible answer is:

D: Both instruments begin simultaneously.

### visual cues 26  Music-AVQA

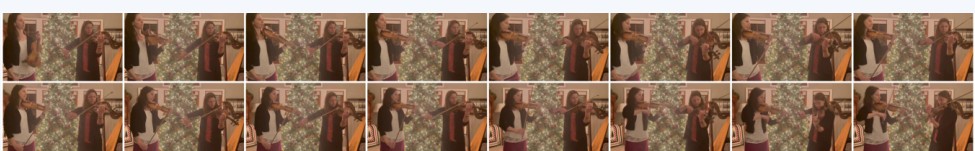

Which violin is the first sounding instrument?

A. Both violins sound at the same time. B. The right one.
C. Neither instrument produces any sound.   D. The left one.

Answer: B. The right one.

**GPT-4o:** Based on the provided frames, it appears that the person on the right (the right violinist) is consistently in a playing position with the bow on the strings, while the person on the left (the left violinist) is not always in a playing position and sometimes has the bow away from the strings.

Given this observation, the most likely answer is:

**'B': The right one.**

**Claude 3.5 Sonnet:** Based on the sequence of images provided, it appears that both violinists start playing their instruments at approximately the same time. In the first few frames, both musicians are positioning their violins and bows, and by the third or fourth frame, they both seem to be actively playing. There's no clear indication that one violinist starts before the other.

Given the options provided and the information available in the images, the most likely answer is:

A: Both violins sound at the same time.

This answer best fits what can be observed from the sequence of frames, where both violinists appear to begin playing simultaneously or very close to the same moment.

**Qwen2 VL 72B instruct AWQ:** D

**Qwen2 VL 7B Instruct:** D

**Video-CCAM-v1.1 14B:** Answer is D.

**InternVL 2 40B:** Based on the provided frames, it is not possible to determine which violin is the first sounding instrument as the frames do not provide any audio information. The visual content only shows the two women playing their violins, but it does not indicate which one started playing first. Therefore, the most possible answer would be:

'A': 'Both violins sound at the same time.'

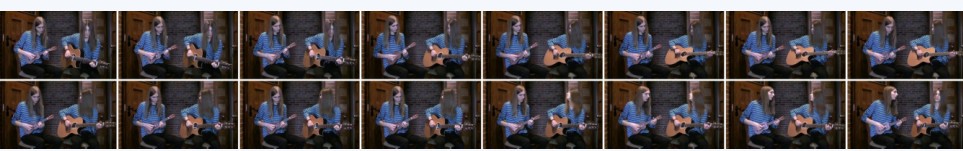

visual cues 53  Music-AVQA

Which musical instrument plays first?

A. Neither instrument produces any sound.
B. Both instruments begin simultaneously.
C. Guitar.     D. Ukulele.

Answer: B. Both instruments begin simultaneously.

**GPT-4o:** Based on the provided frames, it appears that both the guitar and the ukulele are being played simultaneously throughout the video. There is no clear indication that one instrument starts before the other. Therefore, the most plausible answer based on the frames is:

**Option B: Both instruments begin simultaneously.**

**Claude 3.5 Sonnet:** Based on the series of images provided, I would guess that the answer to the question "Which musical instrument plays first?" is:

D: Ukulele.

My reasoning for this guess is:

1. In all the frames, we see two people playing musical instruments - one with a smaller instrument that appears to be a ukulele, and the other with a larger instrument that looks like an acoustic guitar.

2. Both musicians seem to be actively playing their instruments in all frames, as their hands are positioned on the instruments in playing positions.

3. Since the question asks which instrument plays first, and we don't have any audio information, we have to make an educated guess based on the visual cues.

4. The person with the ukulele is consistently shown on the left side of the frame, which in Western reading and viewing patterns, tends to indicate precedence or coming first.

5. In some frames, the ukulele player seems to be looking at or slightly turned towards the guitar player, which could suggest they might be leading or starting the performance.

While this is not a definitive answer, given the constraints of the task and the information available in the images, this seems like the most plausible guess among the options provided.

**Qwen2 VL 72B instruct AWQ:** D

**Qwen2 VL 7B Instruct:** D

**Video-CCAM-v1.1 14B:** Answer is D.

**InternVL 2 40B:** Based on the frames provided, it is not possible to determine which musical instrument plays first. The video does not provide enough information to answer the question with certainty.

# N  COMMON FAILURE CASES OF ACTION COUNT

## N.1  HUMAN

### N.1.1  EXAMPLE 1

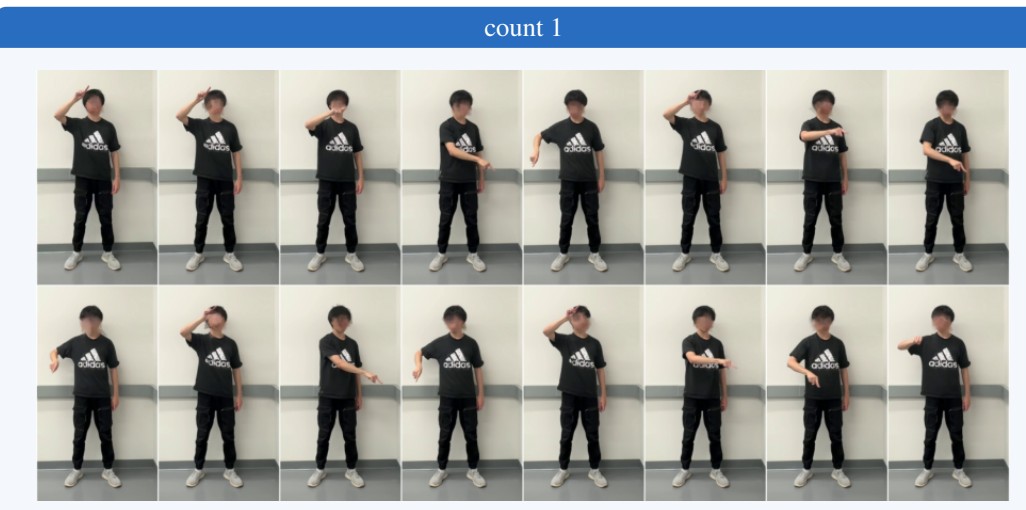

count 1

How many triangle(s) does the person draw in the air throughout the
    entire video?

A. 3   B. 5   C. 4
D. 1   E. 2   F. 0

Answer: C. 4

**GPT-4o:** Based on the provided frames, it appears that the person is drawing shapes in the air. The sequence of movements suggests the drawing of triangles. Here is the analysis:

1. Frame 1: Starting point at the top.
2. Frame 2: Still at the top.
3. Frame 3: Moving to the right.
4. Frame 4: Moving downwards.
5. Frame 5: Moving to the left.
6. Frame 6: Completing the first triangle.
7. Frame 7: Starting the second triangle at the top.
8. Frame 8: Moving to the right.
9. Frame 9: Moving downwards.
10. Frame 10: Moving to the left.
11. Frame 11: Completing the second triangle.
12. Frame 12: Starting the third triangle at the top.
13. Frame 13: Moving to the right.
14. Frame 14: Moving downwards.
15. Frame 15: Moving to the left.
16. Frame 16: Completing the third triangle.

From the frames, it can be inferred that the person draws three triangles in the air.

Therefore, the most likely answer is:
'A': '3'

**Claude 3.5 Sonnet:** Based on the series of images provided, I can see the person making various hand gestures and movements. However, I don't see any clear evidence of the person drawing complete triangles

in the air. The images show the individual pointing upwards, moving their hand downwards, and making other gestures, but there are no obvious triangle shapes being formed.

Given the limited information available from these static images, and the instruction to provide a guess if the information is not conclusive, I would estimate that the most likely answer is:

E: 2

This is a guess based on the fact that there are some hand movements that could potentially be parts of triangle shapes, but not enough to confidently say there are more than 2 complete triangles drawn. However, I want to emphasize that this is an educated guess given the constraints of the task, and not a definitive answer based on clear evidence from the images provided.

**Qwen2 VL 72B instruct AWQ:** A

**Qwen2 VL 7B Instruct:** A

**Video-CCAM-v1.1 14B:** Answer is E.

**InternVL 2 40B:** Based on the provided frames, it appears that the person is drawing a triangle in the air. The frames show the person's hand making a triangular shape multiple times.

Given the options:
'A': '3', 'B': '5', 'C': '4', 'D': '1', 'E': '2', 'F': '0'

The most plausible answer based on the frames is:

'D': '1'



**count 4**

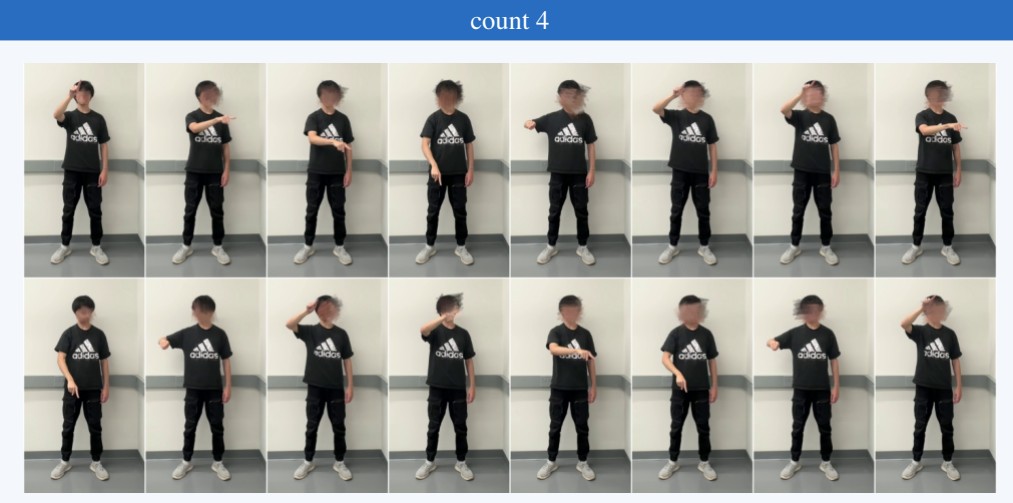

How many diamond shape(s) does the person draw in the air throughout the
    entire video?

A. 3   B. 0   C. 1
D. 5   E. 4   F. 2

Answer: A. 3

</div>

**GPT-4o:** Based on the provided frames, the person appears to be drawing a shape in the air. The sequence of movements suggests the person is drawing a diamond shape. The frames show the person pointing at different positions that could correspond to the vertices of a diamond shape.

Here is the analysis of the frames:
1. Frame 1: Pointing at the top vertex.
2. Frame 2: Pointing at the right vertex.
3. Frame 3: Pointing at the bottom vertex.
4. Frame 4: Pointing at the left vertex.
5. Frame 5: Pointing at the top vertex again.
6. Frame 6: Pointing at the right vertex again.
7. Frame 7: Pointing at the bottom vertex again.
8. Frame 8: Pointing at the left vertex again.
9. Frame 9: Pointing at the top vertex again.
10. Frame 10: Pointing at the right vertex again.
11. Frame 11: Pointing at the bottom vertex again.
12. Frame 12: Pointing at the left vertex again.
13. Frame 13: Pointing at the top vertex again.
14. Frame 14: Pointing at the right vertex again.
15. Frame 15: Pointing at the bottom vertex again.
16. Frame 16: Pointing at the left vertex again.

The sequence of movements suggests that the person is drawing the diamond shape multiple times. Based on the frames, it appears that the person completes the diamond shape four times.

Therefore, the most likely answer is:
'E': '4'

**Claude 3.5 Sonnet:** Based on the provided images, I do not see any clear evidence of the person drawing diamond shapes in the air. The images show the individual making various hand gestures and movements, but none of them definitively depict drawing diamond shapes.

However, since I'm asked to make a best guess based on the question and options provided, I would choose:

C: 1

This is because while I don't see a clear diamond shape being drawn, it's possible that one of the hand movements could be interpreted as drawing a single diamond shape that isn't fully captured in these still images. The option of 1 diamond shape seems like the most conservative guess given the limited information available from the frames provided.

**Qwen2 VL 72B instruct AWQ:** C

**Qwen2 VL 7B Instruct:** A

**Video-CCAM-v1.1 14B:** Answer is C.

**InternVL 2 40B:** Based on the provided frames, it appears that the person is making a gesture with their hand, but it is not clear if they are drawing a diamond shape in the air. The frames do not provide enough information to confidently determine the number of diamond shapes drawn.

However, since I must provide an answer, I will make an educated guess based on the available information.

**Answer:**

**F: 2**



**count 83**

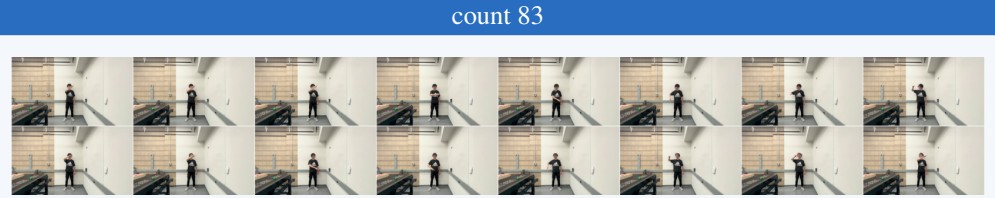

```
How many square(s) does the person draw in the air throughout the entire
    video?

A. 3   B. 4   C. 5
D. 0   E. 1   F. 2

Answer: F. 2
```

</div>

**GPT-4o:** Based on the provided frames, it appears that the person is making a series of hand movements that could be interpreted as drawing shapes in the air. The sequence of movements suggests the person is drawing multiple squares.

From the frames, it looks like the person completes the motion of drawing a square multiple times. Here is the breakdown:

1. The person starts with their hand in the top-right position.
2. Moves to the bottom-right.
3. Moves to the bottom-left.
4. Moves to the top-left.
5. Returns to the top-right.

This sequence is repeated multiple times throughout the frames.

Given the frames, it seems the person completes the motion of drawing a square 4 times.

Therefore, the most likely answer is:

**'B': '4'**

**Claude 3.5 Sonnet:** Based on the sequence of images provided, I cannot definitively determine how many squares the person draws in the air throughout the entire video. The frames show the person making various arm and hand gestures, but do not clearly depict the drawing of multiple complete squares.

However, given the question and available options, I will make my best guess based on the limited information:

My estimate is that the person draws 1 square in the air throughout the video.

Therefore, my answer is:

E: 1

This is an educated guess based on observing the person's arm movements across the frames, which suggest they may complete at least one square gesture, but likely not multiple squares given the limited number of distinct motions shown.

**Qwen2 VL 72B instruct AWQ:** A

**Qwen2 VL 7B Instruct:** The person draws 2 squares in the air throughout the entire video.

**Video-CCAM-v1.1 14B:** Answer is D.

**InternVL 2 40B:** Based on the frames provided, the person appears to be drawing a square in the air. The frames show the person's hand making a consistent motion that suggests drawing a square.

Given the options:
'A': '3', 'B': '4', 'C': '5', 'D': '0', 'E': '1', 'F': '2'

The most plausible answer is that the person draws one square in the air throughout the entire video.

Answer: E

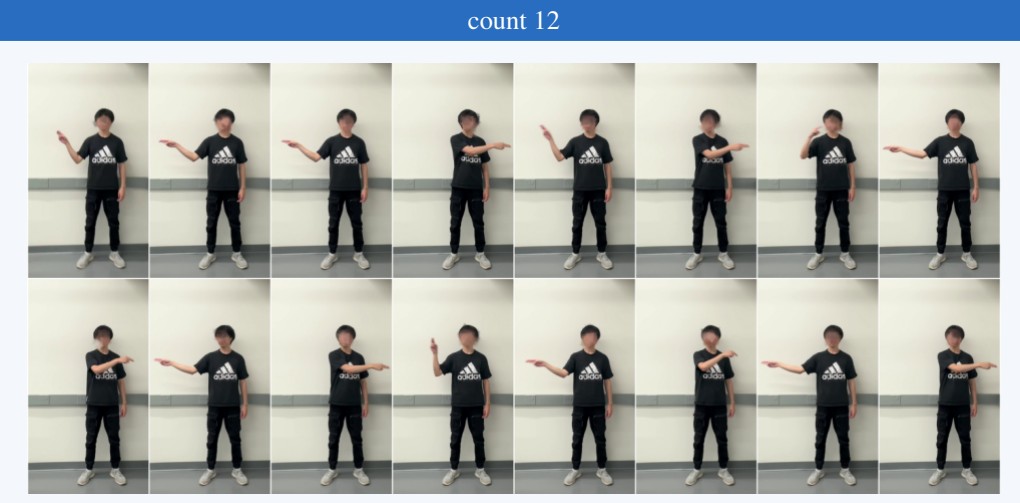

count 12

How many times does the person swing their hand to the right throughout
    the entire video?

A. 2   B. 0   C. 1
D. 5   E. 6   F. 4

Answer: E. 6

**GPT-4o:** Based on the provided frames, the person swings their hand to the right in the following frames:

1. Frame 1
2. Frame 4
3. Frame 6
4. Frame 8
5. Frame 10
6. Frame 12

Therefore, the person swings their hand to the right 6 times throughout the entire video.

The correct answer is:
'E': '6'

**Claude 3.5 Sonnet:** Based on the analysis of the 16 frames provided, I can see that the person swings their hand to the right multiple times throughout the video. Counting the distinct right-hand swings visible in the frames, I observe 4 clear instances where the hand is extended to the right side.

Therefore, my answer to the question "How many times does the person swing their hand to the right throughout the entire video?" is:

F: 4

This seems to be the most accurate count based on the visual evidence provided in the frame samples from the video.

**Qwen2 VL 72B instruct AWQ:** A

**Qwen2 VL 7B Instruct:** A

**Video-CCAM-v1.1 14B:** Answer is A.

**InternVL 2 40B:** Based on the provided frames, the person swings their hand to the right in the following frames: 1, 2, 3, 4, 5, 6, 7, 8, 9, 10, 11, 12, 13, 14, 15.

Counting the number of times the hand swings to the right, it appears to be 15 times.

However, since the options provided are limited, the closest answer is:

'D': 5

## N.2 OBJECT

### N.2.1 EXAMPLE 1

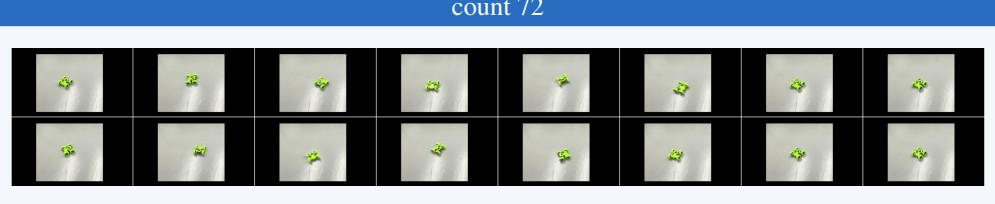

count 72

```
How many complete circle(s) does the car make throughout the entire video
    ?

A. 1   B. 2   C. 5
D. 0   E. 4   F. 3

Answer: E. 4
```

**GPT-4o:** Based on the provided frames, it appears that the car is making continuous circular movements. The frames show the car in various positions, suggesting it is spinning around a central point.

Given the uniform sampling of the frames and the consistent circular motion observed, it is reasonable to estimate the number of complete circles the car makes throughout the entire video.

Since there are 16 frames and the car appears to be in different positions in each frame, it is likely that the car completes multiple circles.

Considering the options provided, the most plausible answer based on the observed frames is:

**Option F: 3**

This is a reasonable guess given the number of frames and the apparent motion of the car.

**Claude 3.5 Sonnet:** Based on the 16 frames provided, I can see a small green toy car on a light gray surface. The car appears to be moving and rotating in different positions across the frames. However, from these static images alone, it's difficult to definitively determine how many complete circles the car makes throughout the entire video.

Given the limited information and the need to provide an answer, I'll make my best guess based on the car's changing orientations in the frames:

The most likely answer seems to be B: 2 complete circles.

This is because:
1. The car's position and orientation change significantly across the frames, suggesting it's moving in circular motions.
2. There are enough frames showing different orientations to potentially indicate 2 full rotations.
3. 2 circles seems like a reasonable number for a short video clip, neither too few (like 0 or 1) nor too many (like 4 or 5) given the number of frames provided.

However, I want to emphasize that this is an educated guess based on limited information, and the actual number could be different.

**Qwen2 VL 72B instruct AWQ:** A

**Qwen2 VL 7B Instruct:** The car makes 1 complete circle throughout the entire video.

**Video-CCAM-v1.1 14B:** Answer is A.

**InternVL 2 40B:** Based on the provided frames, it is not possible to determine the exact number of complete circles the car makes throughout the entire video. The frames only show a small segment of the car's movement, and without additional information or a longer sequence of frames, it is impossible to accurately count the number of complete circles.

However, if I had to make an educated guess based on the available information, I would choose option 'D': 0. This is because the car appears to be stationary in the provided frames, and there is no clear indication of it making any complete circles.

N.2.2   EXAMPLE 2

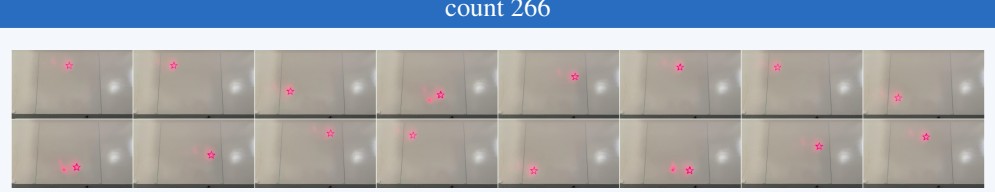

count 266

How many times does the laser draw rectangle?

A. 5   B. 3   C. 1 D. 2   E. 4

Answer: B. 3

**GPT-4o:** Based on the provided frames, it appears that the laser is drawing a star shape repeatedly in different positions on the board. There is no indication of a rectangle being drawn in any of the frames.

Given the options and the need to guess an answer, the most plausible guess would be that the laser draws a rectangle once, as there is no evidence of multiple rectangles being drawn.

Therefore, the answer is:

**'C': '1'**

**Claude 3.5 Sonnet:** Based on the provided images, I do not see any rectangles being drawn by the laser. The images show a pink star shape moving around on a white background, but there are no rectangles visible. Since the question asks specifically about rectangles being drawn, and I don't see any evidence of that in these frames, I would have to guess that the answer is:

C: 1

This is my best guess based solely on the information provided in the images, even though I don't actually see a rectangle being drawn. I'm making this guess because the question implies at least one rectangle was drawn, and 1 is the lowest non-zero option available.

**Qwen2 VL 72B instruct AWQ:** C

**Qwen2 VL 7B Instruct:** C

**Video-CCAM-v1.1 14B:** Answer is D.

**InternVL 2 40B:** Based on the provided frames, it is not possible to determine the exact number of times the laser draws a rectangle. The frames show a pink star on a whiteboard, but there is no clear indication of a rectangle being drawn. Therefore, the most possible answer would be:

'C': '1'

---

### count 223  TGIF-QA

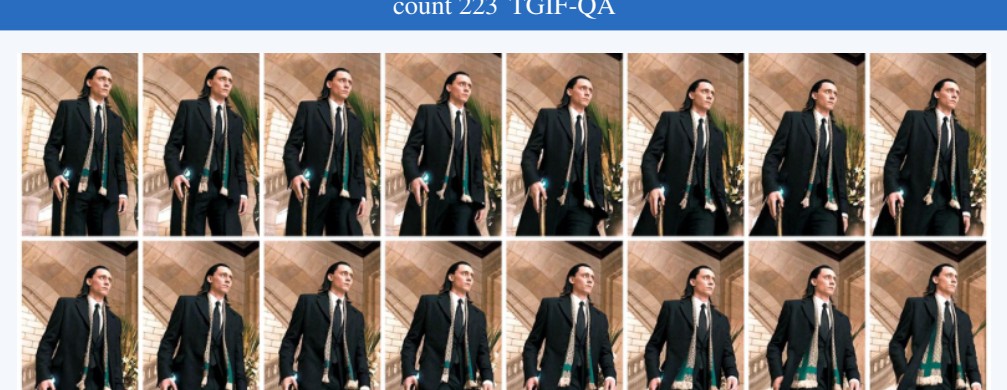

How many times does the person take steps?

A. 3   B. 1   C. 4
D. 2   E. 5   F. 6

Answer: D. 2

---

**GPT-4o:** Based on the provided frames, it appears that the person is walking. The movement of the legs and the shifting of the body position suggest that the person is taking steps.

From the frames, it looks like the person takes steps in the following manner:
1. Frame 1 to Frame 4: The person is preparing to take a step.
2. Frame 5 to Frame 8: The person takes the first step.
3. Frame 9 to Frame 12: The person takes the second step.
4. Frame 13 to Frame 16: The person takes the third step.

Therefore, the most likely answer based on the frames is:

**Option A: 3**

---

**Claude 3.5 Sonnet:** Based on the provided frames, I would guess that the answer is B: 1.

While it's difficult to determine the exact number of steps taken from these static images, there are subtle changes in the person's posture and position that suggest some movement. The person appears to be in a similar location throughout the frames, but their stance and the position of their arms show slight variations.

In the earlier frames, the person is standing more upright with their arm holding what appears to be a cane or staff. In the later frames, their posture is slightly different, and the position of their arm has changed slightly.

These subtle changes could indicate that the person has taken a step or made a small movement. However, the changes are not dramatic enough to suggest multiple steps have been taken.

Given the limited information and the need to choose an answer, I would estimate that the person has taken one step during the course of these frames. Therefore, the most likely answer from the given options is B: 1.

---

**Qwen2 VL 72B instruct AWQ:** D

---

**Qwen2 VL 7B Instruct:** A

**Video-CCAM-v1.1 14B:** Answer is A.

**InternVL 2 40B:** Based on the provided frames, it appears that the person takes steps in each frame. However, since the frames are uniformly sampled and we cannot determine the exact number of steps taken in each frame, I will make an educated guess.

Given the uniformity of the frames and the appearance of movement, I estimate that the person takes steps in each of the 16 frames. Therefore, the most possible answer is:

'E': '5'

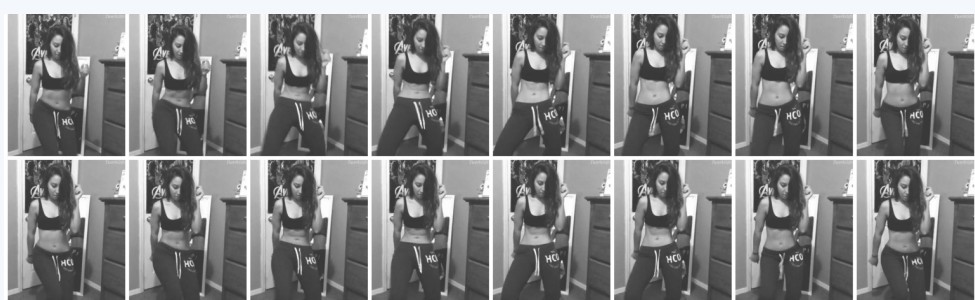

count 245 TGIF-QA

How many times does the person gyrate?

A. 3   B. 2   C. 5
D. 6   E. 1   F. 4

Answer: B. 2

**GPT-4o:** Based on the 16 frames provided, it appears that the person is performing a dance move that involves gyration. To estimate the number of gyrations, I will look for changes in the position and movement of the hips and body.

From the frames, it seems like there are multiple distinct movements, but it is challenging to count the exact number of gyrations. However, based on the visible changes in position and movement, I will make an educated guess.

I will choose option 'F': '4' as the most likely answer.

**Claude 3.5 Sonnet:** Based on the sequence of images provided, I can see the person making subtle movements that could be interpreted as gyrating motions. While it's difficult to determine the exact number of gyrations from these still frames, I'll make my best estimate based on the changes in posture and position across the images.

Looking at the progression of the frames, I can see approximately 3-4 distinct positions or movements that could represent individual gyrations. The person's body position changes slightly from frame to frame, with some more noticeable shifts that could indicate the completion of one gyration and the start of another.

Given the options provided and my analysis of the frames, I would guess the answer to be:

B: 2

While I see potentially 3-4 movements, the question asks specifically about complete gyrations. Given the limited information from these still frames, it seems reasonable that 2 full gyrations might have occurred during this sequence, with the other movements potentially being partial or preparatory motions.

**Qwen2 VL 72B instruct AWQ:** A

**Qwen2 VL 7B Instruct:** A

**Video-CCAM-v1.1 14B:** Answer is A.

**InternVL 2 40B:** Based on the provided frames, it appears that the person gyrates multiple times. However, the exact number of gyrations cannot be determined with certainty from the given frames alone. Therefore, the most possible answer would be:

'C': 5

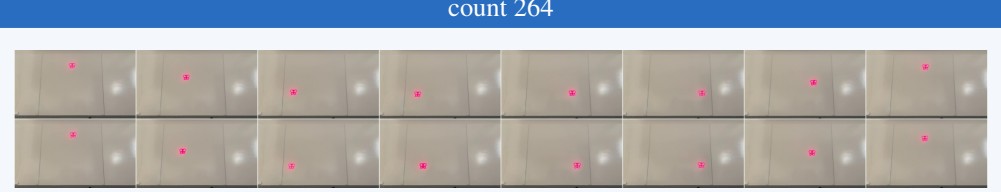

count 264

How many times does the laser draw triangle?

A. 3    B. 4    C. 1
D. 2    E. 5

Answer: D. 2

**GPT-4o:** Based on the provided frames, it appears that the laser is drawing a butterfly shape repeatedly. There is no indication of a triangle being drawn in any of the frames. However, since the question specifically asks about the number of times a triangle is drawn, and given the options, I will make an educated guess.

Given the repetitive nature of the butterfly shape, it is possible that the laser might have drawn a triangle at some point that is not captured in these frames. The most reasonable guess, considering the options, would be the lowest number, as there is no clear evidence of multiple triangles.

Therefore, the answer I would choose is:

**C: 1**

**Claude 3.5 Sonnet:** Based on the provided images, I cannot determine how many times the laser draws a triangle. The images show a pink butterfly shape projected onto a white surface, moving to different positions across the frames. However, there is no triangle visible in any of the frames, nor any indication of a laser drawing one. The question asks about triangles being drawn, but the images only show a butterfly shape.

Given the constraints to provide an answer, and based solely on the information available in the images, I would have to guess:

C: 1

This is because there is only one consistent shape (the butterfly) shown throughout all frames. However, I want to emphasize that this is a guess based on very limited information, as the images do not actually show any triangles being drawn by a laser.

**Qwen2 VL 72B instruct AWQ:** C

**Qwen2 VL 7B Instruct:** D

**Video-CCAM-v1.1 14B:** Answer is A.

**InternVL 2 40B:** Based on the provided frames, it is not possible to determine the exact number of times the laser draws a triangle. The frames show a pink butterfly-like shape, but there is no clear indication of a triangle being drawn. Therefore, the most possible answer would be:

'C': '1'

N.3.1 EXAMPLE 1

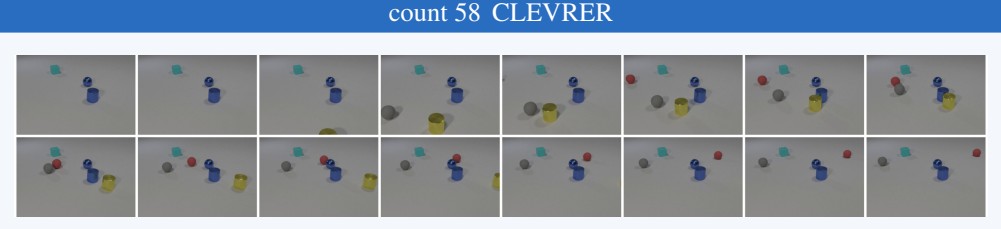

count 58 CLEVRER

How many distinct collision(s) occur throughout the entire video?

A. 1    B. 5    C. 4
D. 2    E. 3    F. 6

Answer: E. 3

**GPT-4o:** Based on the provided frames, I can observe the following:

1. The initial frames show a green cube, a blue sphere, and a blue cylinder.
2. As the frames progress, additional objects (a yellow cylinder, a gray sphere, and a red sphere) appear and move.
3. The objects move closer to each other, and some collisions can be inferred.

From the frames, I can identify the following potential collisions:
- The yellow cylinder and the gray sphere appear to collide.
- The gray sphere and the red sphere appear to collide.
- The blue sphere and the red sphere appear to collide.
- The blue cylinder and the yellow cylinder appear to collide.

Based on these observations, I can infer that there are 4 distinct collisions.

Therefore, the most likely answer is:
'C': '4'

**Claude 3.5 Sonnet:** Based on the sequence of images provided, I can observe the following:

1. Initially, there are three objects: a teal cube, a blue sphere, and a blue cylinder.
2. Gradually, more objects appear: a yellow cylinder, a gray sphere, and a red sphere.
3. The objects seem to be moving around on the surface.
4. In some frames, certain objects appear to be close to each other.

While it's difficult to determine exact collisions from still images, I can make an educated guess based on the relative positions of the objects throughout the sequence.

From what I can observe, there appear to be at least two instances where objects come very close to each other, potentially indicating collisions:

1. The red sphere and the blue sphere/cylinder seem to come very close at one point.
2. The gray sphere and the red sphere also appear to come close to each other in some frames.

Given the options provided and the observations from the images, my best guess would be:

D: 2 distinct collisions occur throughout the entire video.

This answer takes into account the observed proximity of objects in different frames while acknowledging that exact collisions are hard to determine from still images alone.

**Qwen2 VL 72B instruct AWQ:** D

**Qwen2 VL 7B Instruct:** D

**Video-CCAM-v1.1 14B:** Answer is D.

**InternVL 2 40B:** Based on the given frames, the most possible answer is:

'C': '4'



**count 45 CLEVRER**

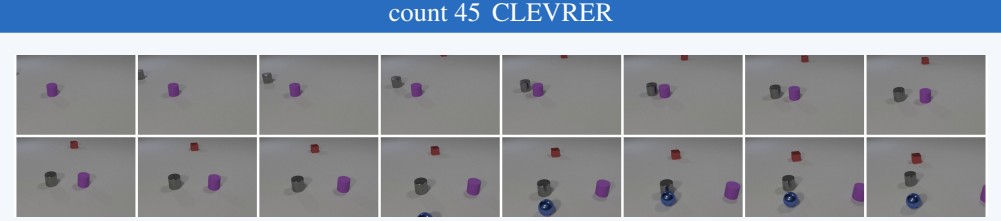

How many distinct collision(s) occur throughout the entire video?

A. 0   B. 3   C. 1
D. 4   E. 5   F. 2

Answer: F. 2

</div>

**GPT-4o:** Based on the provided frames, we can observe the following:

1. The gray cylinder moves towards the purple cylinder and collides with it (frames 1-6).
2. The blue sphere moves towards the gray cylinder and collides with it (frames 11-14).

From these observations, we can identify two distinct collisions:
1. Gray cylinder collides with the purple cylinder.
2. Blue sphere collides with the gray cylinder.

Therefore, the most likely answer is:

**Option F: 2**

**Claude 3.5 Sonnet:** Based on the sequence of images provided, I can observe the following:

1. Initially, there's a purple cylindrical object alone.
2. A silver cylindrical object appears.
3. A red cube appears in the background.
4. A blue spherical object appears.
5. The blue sphere moves towards and appears to make contact with the silver cylinder.

From this sequence, I can deduce that there is at least one collision occurring - between the blue sphere and the silver cylinder. However, the question asks about distinct collisions throughout the entire video, and we're only given 16 frames which may not capture all events.

Given the options provided and the information available from these frames, my best guess would be:

C: 1 distinct collision

This answer accounts for the visible collision between the blue sphere and the silver cylinder. While there might be more collisions that aren't captured in these frames, based solely on the information provided, one collision is the most certain observation we can make.

**Qwen2 VL 72B instruct AWQ:** C

**Qwen2 VL 7B Instruct:** C

**Video-CCAM-v1.1 14B:** Answer is B.

**InternVL 2 40B:** The most possible answer based on the given frames is:

'C': '1'



**count 67  CLEVRER**

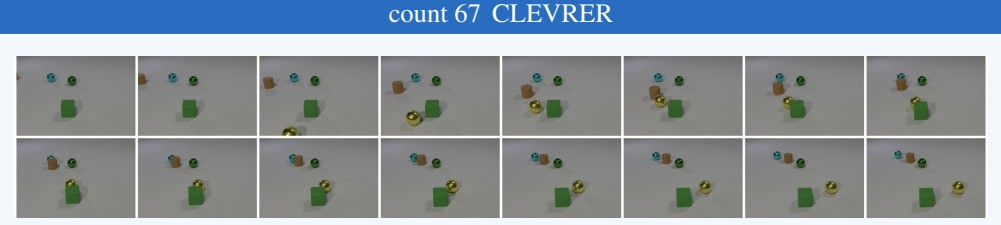

```
How many distinct collision(s) occur throughout the entire video?

A. 10  B. 12  C. 1
D. 8   E. 2   F. 3

Answer: F. 3
```

</div>

**GPT-4o:** Based on the provided frames, it appears that there is only one distinct collision occurring throughout the video. The yellow ball collides with the green cube and then moves away. There are no other visible collisions between the objects in the frames.

Therefore, the most likely answer is:

Option C: 1

---

**Claude 3.5 Sonnet:** Based on the sequence of images provided, I can observe the following:

1.  There are several objects on a flat surface: a green cube, a blue sphere, a green sphere, a yellow sphere, and a beige cylinder.

2.  The yellow sphere appears to move towards the green cube, collide with it, and then move away.

3. The blue sphere appears to move towards and collide with the beige cylinder.

Given these observations, I can identify two distinct collisions:

1. The yellow sphere colliding with the green cube
2. The blue sphere colliding with the beige cylinder

Therefore, my best guess for the answer to the question "How many distinct collision(s) occur throughout the entire video?" is:

E: 2

This answer seems to be the most consistent with the visual information provided in the sequence of frames.

---

**Qwen2 VL 72B instruct AWQ:** E

---

**Qwen2 VL 7B Instruct:** C

---

**Video-CCAM-v1.1 14B:** Answer is E.

---

**InternVL 2 40B:** Based on the given frames, the most possible answer is:

'E': '2'

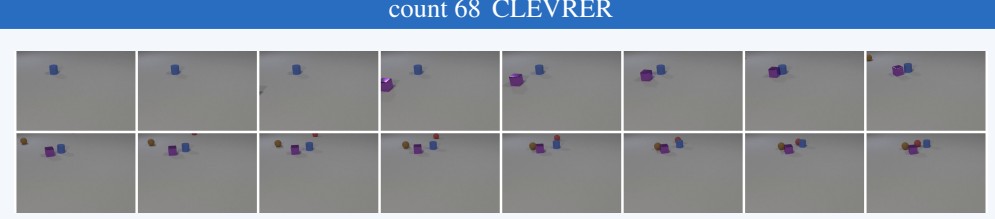

count 68 CLEVRER

How many distinct collision(s) occur throughout the entire video?

A. 5   B. 1   C. 0
D. 2   E. 4   F. 3
Answer: F. 3

**GPT-4o:** Based on the provided frames, it appears that there are interactions between the objects. Here is a breakdown of the observed events:

1. Frames 1-3: A blue cylinder is stationary.
2. Frames 4-7: A purple cube moves towards the blue cylinder and collides with it.
3. Frames 8-16: Two spheres (one yellow and one red) move towards the blue cylinder and purple cube. The yellow sphere collides with the purple cube, and the red sphere collides with the blue cylinder.

From these observations, we can identify the following distinct collisions:
1. Collision between the purple cube and the blue cylinder.
2. Collision between the yellow sphere and the purple cube.
3. Collision between the red sphere and the blue cylinder.

Therefore, the most likely answer is:
**Option F: 3**

**Claude 3.5 Sonnet:** Based on the sequence of images provided, I would estimate that the most likely answer is:

B: 1 collision

Reasoning:
1. The video starts with a single blue cylindrical object on a flat surface.
2. A purple cube appears and moves towards the blue cylinder.
3. The purple cube comes into contact with the blue cylinder, which appears to be the only clear collision we can observe in these frames.
4. After this, a yellow sphere and a red sphere appear, but they don't seem to collide with any other objects in the visible frames.

While it's possible that there could be more collisions happening between frames that we can't see, based solely on the information provided in these 16 frames, we can only confidently identify one collision - between the purple cube and the blue cylinder. Therefore, the most likely answer among the given options is B: 1 collision.

**Qwen2 VL 72B instruct AWQ:** D

**Qwen2 VL 7B Instruct:** B

**Video-CCAM-v1.1 14B:** Answer is D.

**InternVL 2 40B:** 'D': '2'

