# OpenReview forum: "TOMATO: Assessing Visual Temporal Reasoning Capabilities in Multimodal Foundation Models"
_ICLR.cc/2025/Conference — ICLR 2025 Poster_

### Official Review · Reviewer_xcjw · 2024-11-02

**Soundness:** 3
**Presentation:** 3
**Contribution:** 3
**Rating:** 6
**Confidence:** 4

**Summary:**

The paper introduces a novel video benchmark specifically designed to assess the temporal reasoning capabilities of multimodal foundation models. The benchmark is thoughtfully formulated, encompassing a wide range of dynamic and continuous reasoning scenarios over time. By evaluating state-of-the-art video foundation models, the paper highlights fundamental challenges these models face in Video Question Answering (VideoQA). This work provides valuable insights into the limitations of current multimodal models in handling temporal visual tasks, promising to advance research in both the video and large language model (LLM) communities. However, despite these contributions, I still have some concerns and will adjust my rating depending on whether they can be further addressed.

**Strengths:**

- Well-Established Benchmark: The benchmark is rigorously constructed with three quantitative principles that emphasize the complexity of temporal reasoning tasks, ensuring a challenging and relevant evaluation.

- Extensive and Comprehensive Coverage: The benchmark includes a wide range of tasks and scenarios, providing thorough coverage of diverse temporal reasoning challenges, which enhances the evaluation scope and relevance.

- In-Depth SOTA Model Comparison: The paper presents an extensive and detailed comparison of state-of-the-art video foundation models, providing valuable insights into their capabilities and limitations for video question answering.

- Clear and Cohesive Writing: The paper is well-written, presenting its ideas and findings in a clear and organized manner, making it accessible and informative for a broad audience.

**Weaknesses:**

- Sensitivity in Mathematical Formulation: The mathematical formulation of the three principles appears sensitive to certain parameters, particularly the choice of human-picked frames and the number of input frames. For instance, using 16 frames as the multiple-frame input seems arbitrary; different frame counts could significantly impact evaluation scores and model comparison across benchmarks. This choice requires further justification, along with ablation studies on multi-frame input numbers within the three principles, to confirm robustness and fairness.

- Data Imbalance Across Tasks: The sample sizes across the six tasks are imbalanced, with some categories, like Visual Cues, containing only 70 questions—significantly fewer than other categories. This imbalance could impact the benchmark's representativeness and reliability. Addressing this imbalance would improve the benchmark's overall consistency and rigor.

- Fairness in Benchmark Comparisons: Comparisons with other benchmarks may not be entirely fair due to the random sampling of 200 QAs for evaluations. This approach may inadvertently filter out complex temporal reasoning questions. To ensure fairness and maintain the benchmark's integrity, evaluating all QAs from other benchmarks would be preferable.

- Insufficient Text-Only Baseline in Table 5: The current text-only baseline presented in Table 5 may not fully capture the model's reasoning capabilities. Converting video content into textual descriptions for input into a model like GPT-4 would offer a more meaningful comparison with visual inputs. Including this enhanced baseline would provide deeper insights and enrich the discussion in Table 5.

**Questions:**

- Will the benchmark be publicly available?

- How do you generate the queries for each task in the benchmark? Please include this information in the main paper, which may be inspiring to the readers.

- Why not reporting the results of open-source models after fine-tuning them on the temporal reasoning questions. e.g, a training set of the proposed benchmark? It is possible that the model learns to reason from dynamics in the video after fine-tuning. Please justify this.

---

> ### Author Response · Authors · 2024-11-25
> **Response to Reviewer xcjw - Part 1**
>
> Thank you for taking the time to review our work and providing such insightful feedback. We would like to address your concerns and questions as follows:
>
> ----
> > Sensitivity in Mathematical Formulation: The mathematical formulation of the three principles appears sensitive to certain parameters, particularly the choice of human-picked frames and the number of input frames. For instance, using 16 frames as the multiple-frame input seems arbitrary; different frame counts could significantly impact evaluation scores and model comparison across benchmarks. This choice requires further justification, along with ablation studies on multi-frame input numbers within the three principles, to confirm robustness and fairness.
>
> In response to this common concern, we have now extended our analysis to study m uniformly-sampled frames (rather than just m=16) to better assess variations in video content and duration. Given the relatively short video duration and homogenous content within each second, it is very likely that a highly informative frame is sampled as the number of sampled frames increases. Our study across m = 1, 8, 16, 24, and 32 demonstrates that all performances stabilize as m approaches 16, as shown in Figure 4 in Appendix E.3 of the updated manuscript. Therefore, our choice of m = 16 is an outcome of experiments rather than an arbitrary one. The results show that m = 16 provides a sufficient window for models to perform temporal reasoning.
>
> Our handpicked single frame setting is to illustrate: if a model performs equally well (or nearly so) using handpicked single frames compared to how models typically process videos (uniformly-sampled multiple frames), it strongly suggests that it is not the additional frames that helped with answering and the question does not require temporal reasoning.
>
> ----
> > Data Imbalance Across Tasks: The sample sizes across the six tasks are imbalanced, with some categories, like Visual Cues, containing only 70 questions—significantly fewer than other categories. This imbalance could impact the benchmark's representativeness and reliability. Addressing this imbalance would improve the benchmark's overall consistency and rigor.
>
> The Visual Cues category indeed contains fewer questions (70) compared to other categories. These questions were carefully curated through an extensive manual inspection of over 8,000 questions from the Music-AVQA dataset. Expanding this category further poses significant challenges, as identifying and verifying appropriate videos for this specific reasoning type requires substantial human effort and domain expertise. Despite the smaller sample size, these questions effectively encapsulate the core characteristics of this reasoning type.
>
> ----
> > Fairness in Benchmark Comparisons: Comparisons with other benchmarks may not be entirely fair due to the random sampling of 200 QAs for evaluations. This approach may inadvertently filter out complex temporal reasoning questions. To ensure fairness and maintain the benchmark's integrity, evaluating all QAs from other benchmarks would be preferable.
>
> Thank you for your suggestion. To address fairness in benchmark comparisons, the selection of 200 QAs for each benchmark is done through proportional random sampling across all reasoning types (if applicable). This strategy is designed to provide a representative subset that reflects the overall characteristics of each benchmark.

---

> > ### Comment · Reviewer_xcjw · 2024-11-25
> >
> > Thanks for the response. I appreciate your effort to extend the study of frame numbers in the evaluation metrics. However, the sampling strategy for benchmark evaluation is still not a good solution for me. In overall, I would like to accept this paper with rating 6.

---

> ### Author Response · Authors · 2024-11-25
> **Response to Reviewer xcjw - Part 2**
>
> > Insufficient Text-Only Baseline in Table 5: The current text-only baseline presented in Table 5 may not fully capture the model's reasoning capabilities. Converting video content into textual descriptions for input into a model like GPT-4 would offer a more meaningful comparison with visual inputs. Including this enhanced baseline would provide deeper insights and enrich the discussion in Table 5.
>
> We hope to respond to this comment from three aspects.
>
> (1) In our pilot studies, we found that having models generate frame-by-frame descriptions when providing the questions and options does not yield satisfactory results, mainly because of the lack of granularity in the descriptions. An example description is like below:
> ```
> Frame 1: A person is standing next to a round table with a notebook, a tennis ball, an orange, and a bottle on it. The person is wearing a striped shirt and beige shorts.
> Frame 2: The person is now holding a notebook and placing it on the table.
> Frame 3: The person is holding a tennis ball and placing it on the table.
> Frame 4: The person is holding a notebook and placing it on the table.
> [...The above line repeated for frames 5-16…]
> ```
>
> (2) In our experiments, we instructed the models to output their explanations to their answers, serving the purpose of generating descriptions. We found that while GPT-4o often provides the correct descriptions for the transition between consecutive frames, it still fails to interpret across all frames as a continuous sequence. For example, GPT-4o outputs accurate descriptions of the movement of a moon around earth being upper-right, lower-right, lower-left, and then upper-left, but still concludes that such movement is counter-clockwise. Therefore, in response to your comment, we added an analysis paragraph in Section 6.3 titled **Models lack the basic ability to interpret frames as a continuous sequence**. Other example outputs can be found in Appendix I – N; the prompts used can be found in Appendix D1.
>
> (3) We also conducted experiments in converting video content into textual descriptions first using Qwen2-VL-7B (due to limited compute to run Qwen2-VL-72B), and the results from this "enhanced text-only baseline" indicate that model performance is significantly worse across all reasoning types compared to using video input:
>
> | Model  | Rotation | Direction | Velocity & Frequency | Shape & Trend| Visual Cues | Action Count | All  |
> |--------|----------|-----------|----------------------|--------------|-------------|--------------|------|
> Qwen2-VL-7B (enhanced text-only) | 10.1 | 24.8 | 11.9 | 12.1 | 28.6 | 16.1 | 16.7 |
> Qwen2-VL-7B (16 frames) | 23.8 | 29.5 | 41.9 | 29.6 | 37.1 | 34.2 | 31.5 |
>
> ----
> > Will the benchmark be publicly available?
>
> Yes, the benchmark will be made publicly available upon publication.
>
> ----
> > How do you generate the queries for each task in the benchmark? Please include this information in the main paper, which may be inspiring to the readers.
>
> Each task is composed of a video, a question, and 4-6 options. Section 3.3 (Section 4.3 in the updated manuscript) illustrates the question-answer annotation and quality check process. We have now added additional clarifications to this section.
>
> ----
> > Why not reporting the results of open-source models after fine-tuning them on the temporal reasoning questions. e.g, a training set of the proposed benchmark? It is possible that the model learns to reason from dynamics in the video after fine-tuning. Please justify this.
>
> We’d like to clarify that our benchmark is evaluation only, following the approach adopted by many contemporary benchmarks [1][2][3]. Traditional training-testing benchmark setups are less common, as models are increasingly evaluated in zero-shot or few-shot settings to better reflect real-world applications and the current state of foundation model development. Introducing a training set and then fine-tuning models on that set would shift the focus from evaluating general and out-of-the-box reasoning capabilities to testing model performance after domain-specific adaptation, which is outside the scope of our benchmark.
>
> [1] Wang, Zirui, et al. "Charxiv: Charting gaps in realistic chart understanding in multimodal llms." arXiv preprint arXiv:2406.18521 (2024).\
> [2]  Yue, Xiang, et al. "Mmmu: A massive multi-discipline multimodal understanding and reasoning benchmark for expert agi." Proceedings of the IEEE/CVF Conference on Computer Vision and Pattern Recognition. 2024.\
> [3] Lu, Pan, et al. "Mathvista: Evaluating mathematical reasoning of foundation models in visual contexts." arXiv preprint arXiv:2310.02255 (2023).

---

### Official Review · Reviewer_NXCv · 2024-11-02

**Soundness:** 4
**Presentation:** 4
**Contribution:** 3
**Rating:** 8
**Confidence:** 4

**Summary:**

This paper present a new benchmark for visual temporal reasoning. They start by defining three principles for video temporal understanding, namely SINGLE FRAME INSUFFICIENCY, FRAME ORDER SENSITIVITY and  FRAME CONTRIBUTION PARITY. Three principles are then quantified by a metric to reflect the headroom provided by indivisual benchmarks. Finally, a new new benchmark named TVBench with 1484 carefully curated examples are presented and a diverse set of general purpose MFMs are benchmarked against TVBench.

The paper is easy to follow and the key concepts are delivered sound and clear. I appreciate the details the authors provided in the appendix about the analysis and the dataset.

**Strengths:**

The paper is easy to follow and the key concepts are delivered clearly. The authors start by setting three principles and studied the existing video benchmarks around these three principles quantitatively, revealing the opportunities and curating datasets around them, I think the approach to the problem and processes are rigorous and effective. Their final results on evaluating MFMs on their datasets also show big opportunties for current system to improve in terms of temporal reasoning.

**Weaknesses:**

I have some concerns and questions for discussion.

* Clarification on Table 6. It is not very clear to me what "True" & "False" stand for in the table. More explanation and details on the experiment setups are needed. This leads to the question of how the conclusion "More Capable Models Are More Likely to Exploit Shortcuts Through Common Sense." is drawn. Does "True" and "False" mean different split of datasets? If so, those videos and their QAs are different, how could we compare their relative performance drop and reach the conclusion?

* L479 "Models Excel in First-Person Over Third-Person Perspective Temporal Reasoning Video Understanding." The conclusion of model excels FPV over TPV videos on comparing experiment results of 88 FPV QA v.s. 668 TPV QA. Since these are two set of different videos and QAs, it is hard to compare the final score side by side and draw the above conclusion in my opinion. More rigorous study or careful experiment design is needed, for instance, creating a matched set of FPV and TPV videos covering the same scenarios would make more sense.

* References to the visual temporal reasoning principles are needed. Some principles discussed in the manuscript for temporal reasoning in video understandings have been studied and revealed in previous literatures, e.g. "Single Frame Insufficiency" has been discussed in [1]; "Frame Order Sensitivity" has been explored/discussed in many literature, e.g. [2]; [3][4] studied "FRAME CONTRIBUTION PARITY" as well. Having a brief literature review section for each principle, and discuss the difference and contributions of current work could strengthen the manuscript.

[1] Lei, J., Berg, T.L. and Bansal, M., 2022. Revealing single frame bias for video-and-language learning. arXiv preprint arXiv:2206.03428.

[2] Qian, R., Li, Y., Yuan, L., Gong, B., Liu, T., Brown, M., Belongie, S., Yang, M.H., Adam, H. and Cui, Y., 2021. Exploring temporal granularity in self-supervised video representation learning. arXiv preprint arXiv:2112.04480.

[3] Huang, De-An, Vignesh Ramanathan, Dhruv Mahajan, Lorenzo Torresani, Manohar Paluri, Li Fei-Fei, and Juan Carlos Niebles. "What makes a video a video: Analyzing temporal information in video understanding models and datasets." In Proceedings of the IEEE Conference on Computer Vision and Pattern Recognition, pp. 7366-7375. 2018.

[4] Liu, X., Pintea, S.L., Nejadasl, F.K., Booij, O. and Van Gemert, J.C., 2021. No frame left behind: Full video action recognition. In Proceedings of the IEEE/CVF conference on computer vision and pattern recognition (pp. 14892-14901).

**Questions:**

* The author seem not mention it in the paper - will the benchmark be made publicly available at some point?
* The author selectively evaluate some proprietary MFMs in Table5. It would be interesting to report the results of different model sizes in the same model family  (e.g. Gemini 1.5 pro v.s. Gemini 1.5 flash; Claude 3.5 Sonnet v.s. Opus) to understand the scaling effect of current SOTA MFMs on the proposed benchmarks.

---

> ### Author Response · Authors · 2024-11-25
> **Response to Reviewer NXCv - Part 1**
>
> Thank you for taking the time to review our work and providing suggestions for our analysis. We would like to address your concerns and questions as follows:
>
> ----
> > Clarification on Table 6. It is not very clear to me what "True" & "False" stand for in the table. More explanation and details on the experiment setups are needed. This leads to the question of how the conclusion "More Capable Models Are More Likely to Exploit Shortcuts Through Common Sense." is drawn. Does "True" and "False" mean different split of datasets? If so, those videos and their QAs are different, how could we compare their relative performance drop and reach the conclusion?
>
> In Table 6 (Table 7 in Appendix B in updated manuscript), “True” and “False” indicate different splits of whether the questions meet the criteria specified in the merged cell above the respective column. For instance, in the “Counterfactual” column, “True” denotes performance on questions that are counterfactual, while “False” denotes performance on questions that are not counterfactual.
>
> As shown in Table 6, more capable models (GPT-4o and Qwen2-VL-72B) show much lower performance in the counterfactual scenarios (i.e., those in the  “True” sub-column). This observation suggests that these models are more likely to exploit shortcuts through common sense rather than performing reasoning over the video content.
>
> Our videos are labeled after collection and generation, meaning that the video scenarios are not purposefully designed to prove such conclusions but rather to suggest general trends displayed in our experiment results. We have now added clarifications and downgraded these analysis to Appendix B Scenario Analysis.
>
> ----
> > L479 "Models Excel in First-Person Over Third-Person Perspective Temporal Reasoning Video Understanding." The conclusion of model excels FPV over TPV videos on comparing experiment results of 88 FPV QA v.s. 668 TPV QA. Since these are two set of different videos and QAs, it is hard to compare the final score side by side and draw the above conclusion in my opinion. More rigorous study or careful experiment design is needed, for instance, creating a matched set of FPV and TPV videos covering the same scenarios would make more sense.
>
> Thank you for this comment. We acknowledge that a comprehensive comparison of FPV and TPV requires a more rigorous study. However, by including the comparison of FPV and TPV, our goal is to explore the general trends in model performances based on the presence or absence of a *main subject* in the video, rather than a comprehensive study. The results *suggest* that not having a main subject does not negatively impact the model performance. We acknowledge that it needs a better video collection process to rigorously *prove* the effect of perspectives where the scene variables are more strictly controlled. We have carefully rephrased our conclusion of these results to reflect this point.

---

> ### Author Response · Authors · 2024-11-25
> **Response to Reviewer NXCv - Part 2**
>
> > References to the visual temporal reasoning principles are needed. Some principles discussed in the manuscript for temporal reasoning in video understandings have been studied and revealed in previous literatures, e.g. "Single Frame Insufficiency" has been discussed in [1]; "Frame Order Sensitivity" has been explored/discussed in many literature, e.g. [2]; [3][4] studied "FRAME CONTRIBUTION PARITY" as well. Having a brief literature review section for each principle, and discuss the difference and contributions of current work could strengthen the manuscript.
>
> Thank you for your suggestions. We have added a brief literature review for each principle in the updated manuscript.
>
> ----
> > The author seem not mention it in the paper - will the benchmark be made publicly available at some point?
>
> Yes, the benchmark will be made publicly available upon publication.
>
> ----
> > The author selectively evaluate some proprietary MFMs in Table5. It would be interesting to report the results of different model sizes in the same model family (e.g. Gemini 1.5 pro v.s. Gemini 1.5 flash; Claude 3.5 Sonnet v.s. Opus) to understand the scaling effect of current SOTA MFMs on the proposed benchmarks.
>
> Table 5 presents evaluation results on Gemini 1.5 flash, Claude 3.5 Sonnet, and Claude 3 Opus. We have now added evaluation of Gemini 1.5 pro, and Claude 3 Haiku. We also attach results of different model sizes in the same model family below, which highlights the scaling effect of the current SOTA MFMs:
>
> | **Model Family**    | **Variant**         | **Accuracy (%)** | **Note** |
> |----------------------|--------------------|------------------| ------- |
> | **Gemini 1.5**       | Pro                | 36.1             ||
> |                      | Flash              | 27.8             ||
> | **Claude 3**         | 3 Opus             | 28.2             ||
> |                      | 3.5 Sonnet         | 27.8             ||
> |                      | 3 Haiku            | 26.2             |3.5 Haiku is text-only|
> | **GPT**              | GPT-4o             | 37.7             ||
> |                      | GPT-4o-mini        | 28.8             ||

---

> > ### Comment · Reviewer_NXCv · 2024-11-26
> > **Thank you for the rebuttal**
> >
> > I'd like to thank authors for their rebuttal. I think the dataset is a good contribution to the community and the evaluation results show current state-of-the-art MFM have large rooms for improvement on this dataset. I'd recommend for acceptance.

---

### Official Review · Reviewer_tp5Z · 2024-11-03

**Soundness:** 3
**Presentation:** 4
**Contribution:** 2
**Rating:** 8
**Confidence:** 4

**Summary:**

This paper investigates whether current multimodal foundation models can perform visual temporal reasoning, which is a complex task requiring models to incorporate visual and temporal information simultaneously. The authors show that current benchmarks on video understanding may be insufficient to truly test the model's capabilities in visual temporal reasoning and introduce TVBench which consists of challenging video and question pairs that expose a significant gap between multimodal foundation models and humans, hence exhibiting limitations in current multimodal models.

**Strengths:**

- The paper tackles an important aspect of video understanding, which with benchmarks that are not carefully designed, might be overestimated.
- The dataset seems to consist of well designed and curated data, as their design philosophy matches the results on the three metrics they propose. This is likely due to the rigorous quality check, and the diversity of videos (including simulations) that the authors include in the data.
- Intensive experiments and the analyses provide valuable insights into the performance of multimodal foundation models on video data.

**Weaknesses:**

- If the authors could include an experiment to test the effect of language prior - i.e. the model's accuracy when only given the text part of the dataset questions, and report the difference, that might also help reinforce the soundness of their benchmark design.
- If the authors could provide some information on the resolution and scale of the videos, that might be helpful especially because the authors discuss models not benefiting from more than 8 frames.

**Questions:**

1. What would happen if corresponding accuracy is zero in the denominator for the three metrics? Might need an epsilon term, or it might just be sufficient to subtract the two accuracies.
2. Why might there be a performance plateau beyond 8 frames? Is it in the design of common foundation models? Is it likely due to there being too much information to process (for example, the way the models process more than 8 images) ? Do various models show the plateau behavior exactly at 8 frames? What about at smaller frames?
3. How does the performance of zoomed-in views differ when only tested on simulation videos with less noise?

**Details Of Ethics Concerns:**

There are sourced YouTube videos in the dataset, which might have some implications in terms of privacy, security and safety.

---

> ### Author Response · Authors · 2024-11-25
> **Response to Reviewer tp5Z**
>
> Thank you for taking the time to review and recognizing the strengths of our work. We would like to address your concerns and questions as follows:
>
> ----
> > There are sourced YouTube videos in the dataset, which might have some implications in terms of privacy, security and safety.
>
> All videos sourced from Youtube are licensed under Creative Commons, which states that “This license enables reusers to distribute, remix, adapt, and build upon the material in any medium or format, so long as attribution is given to the creator”. We also included further clarification in Appendix G (Appendix F in the updated manuscript): “We do not hold the copyright to any of the YouTube videos; we ask that you respect the rights of the original video creators. If you believe that any content in TVBench infringes on your rights, please contact the authors immediately, and we will remove the video accordingly.”
>
> ----
> > If the authors could include an experiment to test the effect of language prior - i.e. the model's accuracy when only given the text part of the dataset questions, and report the difference, that might also help reinforce the soundness of their benchmark design.
>
> Our text-only baseline (Random (GPT-4o)) in Table 5 is intended to address this. In this setting, GPT-4o was prompted to answer the questions using only the textual input, without access to the videos. The overall performance in this text-only setting is 23.1%, resulting in a difference of 4.6% with the random baseline (Random Choice (42)), which shows that we introduce limited bias in the textual input.
>
> ----
> > If the authors could provide some information on the resolution and scale of the videos, that might be helpful especially because the authors discuss models not benefiting from more than 8 frames.
>
> Thank you for your suggestion. We have now included the resolution details of the videos in Table 9 (Appendix E.1) of the updated manuscript. The average resolution is 1332x1076, and the maximum resolution is 1080x1920. However, some models might preprocess video inputs to its required resolutions.
>
> ----
> > What would happen if corresponding accuracy is zero in the denominator for the three metrics? Might need an epsilon term, or it might just be sufficient to subtract the two accuracies.
>
> Thank you for your suggestion. We now incorporate an epsilon term to prevent division by zero in the updated manuscript.
>
> ----
> > Why might there be a performance plateau beyond 8 frames? Is it in the design of common foundation models? Is it likely due to there being too much information to process (for example, the way the models process more than 8 images) ? Do various models show the plateau behavior exactly at 8 frames? What about at smaller frames?
>
> We hypothesize that the performance plateau is likely due to current foundation models being commonly trained on datasets with a fixed number of frames, such as 8 frames [2] or 16 frames [1]. This training design might limit foundation models’ capability in effectively processing additional temporal information with more frames than they were originally trained on.
>
> We have performed additional analyses with smaller frame counts: 0, 1, 2, 4, 6, and 8 frames. As shown in Figure 5 (Appendix E.4) of the updated manuscript, the performance shows a general increase in this range. However, we also hypothesize that the plateau is also dependent on the difficulty level of the task.
>
> [1] Fei, Jiajun, et al. "Video-ccam: Enhancing video-language understanding with causal cross-attention masks for short and long videos." arXiv preprint arXiv:2408.14023 (2024). \
> [2] Wang, Yi, et al. "Internvideo2: Scaling video foundation models for multimodal video understanding." arXiv preprint arXiv:2403.15377 (2024).
>
> ----
> > How does the performance of zoomed-in views differ when only tested on simulation videos with less noise?
>
> To clarify, *only* human videos have zoomed-in views. *All* simulated human videos are generated from zoomed-in real human videos. This generation from zoomed-in views only is to ensure a more accurate conversion from the real human motions to 3D skinned human models. Thus, Figure 3 (Figure 2(b) of the updated manuscript) presents, given zoomed-in views, the results of real humans compared to simulated humans.

---

> > ### Comment · Reviewer_tp5Z · 2024-11-25
> > **Good rebuttal**
> >
> > I thank the authors for providing the rebuttal. I have carefully read other reviews and the authors' rebuttals.
> >
> > Most of my concerns were addressed, and hence I retain my score to accept the paper.

---

### Official Review · Reviewer_rS5B · 2024-11-04

**Soundness:** 2
**Presentation:** 4
**Contribution:** 3
**Rating:** 5
**Confidence:** 4

**Summary:**

This paper tackles the problem of benchmarking temporal reasoning using multimodal foundation models. First, the paper establishes some principles that temporal reasoning benchmarks must follow in order to test temporal reasoning capabilities, and shows that existing benchmarks fail to cater to these principles. Second, a new manually annotated benchmark, comprising six question types with some counterfactual modifications, is proposed. Third, several existing models are benchmarked demonstrating a big difference between human performance and best model's performance. And finally, some ablation studies are presented that show how one can improve models capabilities on this benchmark.

**Strengths:**

The paper is written well and easy to follow. It addresses an important area in video understanding research, where several benchmarks are known to have shortcomings in actually requiring all the frames of a video [1].
- The benchmark is manually curated in a three-stage process, potentially yielding very high-quality QA annotations
- Several models including proprietary ones lag significantly behind humans. This verifies that the benchmark is indeed difficult for existing models.
- A commendable effort is made in conceptualizing the principles defining how to check whether a benchmark requires temporal reasoning, and its shown that TVBench is way ahead of other benchmarks in adhering to these principles as shown in Tables 1-4
- Some of the ablations / analysis about existing models' capabilities are interesting, i.e. the plateauing at 8 frames compared with humans, exploiting common sense shortcuts etc.

[1]: Revealing Single Frame Bias for Video-and-Language Learning

**Weaknesses:**

The main weakness in my opinion is that the defined principles and ways of measuring benchmarks' adherence to these principles is not very rigorous.

1: Single frame insufficiency:
- Why is the definition limited to a single frame? What about QA pairs that can be answered in just a few frames, say 2 frames? In the abstract, it's argued that such a scenario is bad for temporal reasoning benchmark, however, this principle will consider such a question as good.
- In the second implementation, a single hand-picked frame is compared with uniformly sampled 16 frames. This seems a bit unfair to the 16 frame case. Shouldn't we compare hand-picked 16 frames with hand-picked 1 frame to get a true sense of single frame insufficiency?

2: Frame order sensitivity:
The definition assumes that frame order sensitivity is essential for a good temporal reasoning benchmark. However, a powerful model human can easily figure out the temporal order in a lot of cases. So, I don't know if this needs to be a hard requirement.


3: Frame contribution parity:
Implementation-wise, again why only a single frame is considered? The proposed implementation will not give any signal for a QA pair requiring 2 frames.

For example, a benchmark that simply contains state transition questions requiring exactly 2 frames will get perfect scores on these three principles. However, such a benchmark is not very valuable. In effect, these principles can be considered necessary conditions (second one is a bit questionable), but not sufficient conditions for a benchmark to be considered as requiring temporal reasoning.

Lastly, the way of evaluating these principles using pre-existing models creates a chicken and egg problem, as the models themselves may not be capable of answering the said questions. For example, a very hard benchmark that GPT-4o cannot answer will fail all the principles if we use GPT-4o to evaluate the principles.

A couple of unrelated issues are:
- Many of the proposed question types already are part of existing works, so it’s unclear how this benchmark would be any different in satisfying the principles. Is the difficulty coming from the annotation process and counterfactuals?
- The benchmark may contain a lot of unrealistic questions because of (i) counterfactuals and (ii) simulated data that are used to artificially make the benchmark harder. These may affect the usefulness in real-world applications.

**Questions:**

Audio inputs (speech) are not discussed in this benchmark. What happens if audio (speech) is included as part of video inputs to the models? Will the results on principles and and on TVBench change?

---

> ### Author Response · Authors · 2024-11-25
> **Response to Reviewer rS5B - Part 1**
>
> Thank you for taking the time to review our work and providing such insightful feedback. We would like to address your concerns and questions as follows:
>
> ----
> > The main weakness in my opinion is that the defined principles and ways of measuring benchmarks' adherence to these principles is not very rigorous.
>
> > For example, a benchmark that simply contains state transition questions requiring exactly 2 frames will get perfect scores on these three principles. However, such a benchmark is not very valuable. In effect, these principles can be considered necessary conditions (second one is a bit questionable), but not sufficient conditions for a benchmark to be considered as requiring temporal reasoning.
>
> We’d like to first clarify a potential misunderstanding about the purpose of the principles we have developed. Our goal is _not_ to develop principles that **define** the entire spectrum of what constitutes temporal reasoning. Rather, these principles serve as **diagnostic tools** to reveal loopholes that (1) exist in current benchmarks and (2) are exploited by current foundation models.
>
> Our results in Table 1-4 show that our current principles are sufficient to demonstrate several of these loopholes. By alerting the research community to these loopholes, we believe that our work is both impactful and timely.
>
> We agree that there are likely many other prerequisites for a question to truly “require” temporal reasoning. We hope that, by being the first to pose such principles, our work can inspire others to continue to discover more.
>
> ----
> > 1.1 Single frame insufficiency: Why is the definition limited to a single frame? What about QA pairs that can be answered in just a few frames, say 2 frames? In the abstract, it's argued that such a scenario is bad for temporal reasoning benchmark, however, this principle will consider such a question as good.
>
> As you have suggested, our definition of “single frame insufficiency” serves as a deliberately minimal baseline—a **lower bound** for exposing the lack of temporal reasoning in current benchmarks and models. Specifically, if a model performs equally well (or nearly so) using single frames compared to multiple frames, it strongly suggests that it is not the additional frames that helped with answering and the question does not require temporal reasoning.
>
> This is what our results in Table 1-2 show—existing benchmarks and models don’t even meet this low standard. While our principles may not directly expose all such cases (e.g., two-frame sufficiency), they are still very useful for identifying common deficiencies. Realistically, temporal reasoning should require more frames than 2 frames. Coming up with a higher standard for frame sufficiency will be important once models become more capable of temporal reasoning. Furthermore, the exact number of frames used for this standard may be dependent on the number of separate events depicted in the video.
>
> ----
> > 1.2 Single frame insufficiency: In the second implementation, a single hand-picked frame is compared with uniformly sampled 16 frames. This seems a bit unfair to the 16 frame case. Shouldn't we compare hand-picked 16 frames with hand-picked 1 frame to get a true sense of single frame insufficiency?
>
> Models typically process a video as a set of uniformly-sampled frames. If a model’s performance increases when it is given more samples, one might infer that it is performing temporal reasoning (i.e., that it is reasoning over the relationship between frames). However, there is an alternative explanation: as the number of samples increases, it becomes more likely that one of those samples is highly-informative. Thus, the model may not be considering the relationship between frames at all, but rather, is reasoning over them individually.
>
> The purpose of our test (1 hand-picked frame vs 16 uniformly-sampled) is to detect whether a model may be exploiting this loophole. The hand-picked frame is our approximation of a “best-case” scenario; i.e., a single frame that is maximally representative of the entire video. If a model achieves similar performance in both conditions, then it may be relying on the information within a single frame and is not leveraging all 16 frames available to it.
>
> We have now extended our analysis to study m uniformly-sampled frames (rather than just m=16) to better assess variations in video content and duration. Given the relatively short video duration and homogenous content within each second, it is very likely that a highly informative frame is sampled as the number of sampled frames increases. Our study across m = 1, 8, 16, 24, and 32 demonstrates that all performances stabilize as m approaches 16, as shown in Figure 4 in Appendix E.3 of the updated manuscript. Therefore, our choice of m = 16 is an outcome of experiments rather than an arbitrary one. The results show that m = 16 provides a sufficient window for models to perform temporal reasoning.

---

> ### Author Response · Authors · 2024-11-25
> **Response to Reviewer rS5B - Part 2**
>
> > 2. Frame order sensitivity: The definition assumes that frame order sensitivity is essential for a good temporal reasoning benchmark. However, a powerful model human can easily figure out the temporal order in a lot of cases. So, I don't know if this needs to be a hard requirement.
>
> We hope to clarify this from two aspects. (1) Table 3 shows a significant performance drop on TVBench when frames are shuffled, indicating even advanced models struggle to infer order. (2) While it is true that a sufficiently powerful model or human could infer temporal order in some cases, Figure 1 highlights that the common issue with existing benchmarks lies not in shuffled frames being easily solvable, but in QAs that do not necessitate reasoning in temporal order. Thus, we believe our metric effectively identifies tasks that fail to require reasoning with the correct frame order.
>
> ----
> > 3. Frame contribution parity: Implementation-wise, again why only a single frame is considered? The proposed implementation will not give any signal for a QA pair requiring 2 frames.
>
> We propose through Frame Contribution Parity that each frame should contribute equally to answering the question. Thus, evaluating with the exact number of frames needed to answer the question is unnecessary. By demonstrating a significant performance gain from random to handpicked single frames (even for questions that require 2 frames to answer), we show that the informativeness of frames is uneven.
>
> ----
> > Lastly, the way of evaluating these principles using pre-existing models creates a chicken and egg problem, as the models themselves may not be capable of answering the said questions. For example, a very hard benchmark that GPT-4o cannot answer will fail all the principles if we use GPT-4o to evaluate the principles.
>
> Given the potential misunderstanding that these principles **guarantee** a question requires temporal reasoning, we agree that relying heavily on GPT-4o to qualify good temporal reasoning tasks does create a chicken and egg problem. However, as mentioned above, the goal of these principles is to expose common loopholes in existing benchmarks. GPT-4o only helps in rejecting the bad examples and is not used in creating the valid questions.
>
> We acknowledge that even GPT-4o will fail all the principles on a very hard benchmark. As shown in Table 5, the performances of the SOTA models are still above the baselines, meaning that while our questions are challenging, they are still feasible for models to perform a certain degree of valid reasoning.

---

> ### Author Response · Authors · 2024-11-25
> **Response to Reviewer rS5B - Part 3**
>
> > Many of the proposed question types already are part of existing works, so it’s unclear how this benchmark would be any different in satisfying the principles. Is the difficulty coming from the annotation process and counterfactuals?
>
> Thank you for your question. As illustrated in Table 7 in Appendix A (Table 6 of the updated manuscript), among the six reasoning types included in TVBench, only Visual Cues (4.7%) and Action Counts (19.7%) are sourced from existing datasets. For Visual Cues, we repurpose and reannotate all videos that we sourced from Music-AVQA. For Action Counts, we acknowledge that this is an existing reasoning type within some existing works and we edit and reannotate a part of those videos. In inclusion of the Action Counts reasoning type as one of the six reasoning types, TVBench aims to _comprehensively_ evaluate all aspects of visual temporal reasoning of current models.
>
> ----
> > The benchmark may contain a lot of unrealistic questions because of (i) counterfactuals and (ii) simulated data that are used to artificially make the benchmark harder. These may affect the usefulness in real-world applications.
>
> We hope to respond to this question from two aspects.
>
> (1) There are 233 questions from simulated data and 100 counterfactual questions (not mutually exclusive), which constitute 22.3% of the benchmark QAs. This small portion does not detract from its primary focus on real-world visual temporal reasoning.
>
> (2) While the benchmark does include questions that may be considered “unrealistic” (e.g., counterfactual and questions derived from simulated videos), these questions are intentionally designed to test models’ capabilities in handling out-of-distribution tasks. Such tasks provide valuable insights into models’ robustness and generalizability, which is crucial for real-world applications. To better illustrate our point, we now included two analysis paragraphs in Section 6.3, titled **Models fail to truthfully leverage the visual input while being over-reliant on common sense** and **Models are highly susceptible to noisy information in the input**.
>
> ----
> > Audio inputs (speech) are not discussed in this benchmark. What happens if audio (speech) is included as part of video inputs to the models? Will the results on principles and and on TVBench change?
>
> Thank you for raising this interesting point. Since this work focuses primarily on visual temporal reasoning, audio modality is beyond the scope of this work. All inputs in our evaluation do not include audio. If audio (speech) were to be included, among the six reasoning types, models are likely to take shortcuts in answering questions from Visual Cues, Action Counts, and Velocity & Frequency, leading to an overestimation of model performance on our benchmark.

---

> ### Author Response · Authors · 2024-12-02
> **Looking forward to your feedback**
>
> Dear Reviewer rS5B,
>
> Thanks again for your insightful and thoughtful comments!
>
> With the author-reviewer discussion period ending soon (Dec 2nd at 23:59 AoE), we wanted to take this opportunity to thank you again for your suggestions and to ask whether you had any remaining questions after our responses! We especially hope that you have a chance to review the additional clarifications we provided regarding our metrics.
>
> If we've addressed your concerns, we'd be grateful if you'd consider updating your score!
>
> Best regards,
>
> Authors

---

> ### Comment · Reviewer_rS5B · 2024-12-03
>
> Thanks for your response!
>
> The authors suggest that the principles developed in this paper are diagnostic tools and not comprehensive. If that is the case, it's best to describe them as diagnostic tools in the paper for clarity.
>
> `We hope that, by being the first to pose such principles, our work can inspire others to continue to discover more.`
>
> There are several studies that discuss the drawbacks of existing benchmarks in terms of temporal understanding, spread across different works [e.g. 1, 2, 3].
>
> [1] Revealing Single Frame Bias for Video-and-Language Learning \
> [2] Verbs in Action: Improving verb understanding in video-language models \
> [3] PAXION: Patching Action Knowledge in Video-Language Foundation Models (NeurIPS Spotlight)

---

> ### Author Response · Authors · 2024-12-03
> **Response to Reviewer rS5B's Feedback**
>
> Thank you for your feedback!
>
> > Principles developed in this paper are diagnostic tools and not comprehensive. If that is the case, it's best to describe them as diagnostic tools in the paper for clarity.
>
> We appreciate the reviewer’s suggestions and would like to clarify that the introduction and Section 3 both indicate the principles are intended to evaluate the effectiveness of tasks in visual temporal reasoning, underscoring the diagnostic nature of these principles rather than presenting them as a comprehensive definition of the entire spectrum of visual temporal reasoning. We will further clarify this in our updated manuscript.
>
> > There are several studies that discuss the drawbacks of existing benchmarks in terms of temporal understanding, spread across different works [e.g. 1, 2, 3].\
> [1] Revealing Single Frame Bias for Video-and-Language Learning\
> [2] Verbs in Action: Improving verb understanding in video-language models\
> [3] PAXION: Patching Action Knowledge in Video-Language Foundation Models (NeurIPS Spotlight)
>
> We appreciate the reviewer highlighting these related works. As suggested by *reviewer NXCv*, to strengthen our claims, we have now included a brief literature review for each principle in our updated manuscript. As you suggested, [1][2][3] are also relevant in the context of temporal reasoning. Specifically, [1][2] relate to **Principle 1: Multi-Frame Gain**, considering “static appearance bias”[1] and “bias towards singleframe concepts”[2]. [3] relates to **Principle 2: Frame Order Sensitivity**, considering video reversal, a special case of frame shuffling. We will include these works as parts of our literature review in our updated manuscript.
>
> While these works address individual aspects of temporal reasoning, to the best of our knowledge, our work is the first to unify them as principles with corresponding metrics to: **(1) quantitatively evaluate the effectiveness of visual temporal reasoning tasks**, and **(2) expose loopholes in existing temporal reasoning video understanding benchmarks**. By alerting the research community to these loopholes and by presenting TVBench, we hope that our work can inspire others to propose more rigorous tasks as well as principles and metrics for the advancement of multimodal foundation models.

---

### Author Response · Authors · 2024-12-04
**General Response - Part 2/2**

We greatly appreciate the reviewers' comments and feedback, which have been invaluable in improving our work. We believe we have addressed all the raised concerns, questions, and suggestions raised by the reviewers, including:

- Reviewer rS5B: \
(1) Principles are not very rigorous: We have clarified our principles are intended to serve as a **diagnostic** tool and explained the motivation for each principle at [openreview response link](https://openreview.net/forum?id=fCi4o83Mfs&noteId=cpfjngP43a).\
(2) Question overlap with the existing benchmark: We clarified that although there exists overlap in the *Action Count*, in inclusion of six reasoning types, TVBench aims to _comprehensively_ evaluate all aspects of visual temporal reasoning of current models. Details about QA re-annotation can be found in Section 4.3 and task examples in Table 6.\
(3) Contain unrealistic questions: We have clarified the “unrealistic” questions (i) constitute a small portion of the benchmark, and (ii) are intended to evaluate model capabilities in handling out-of-distribution tasks. We included additional analysis to illustrate this point in the revised Section 6.3, revealing that models (i) **fail to truthfully leverage the visual input while being over-reliant on common sense**, and (ii) **are highly susceptible to noisy information in the input**.

- Reviewer tp5Z:\
(1) Potential ethical concern for video collection: We clarified that all videos sourced from YouTube are licensed under CC 4.0 and more license information can be found in Appendix F. \
(2) Experiments to test language prior: We clarified that our text-only baseline (Random (GPT-4o)) in Table 5 is intended to address this.\
(3) Suggestions including resolution and mathematical stability: We have included video resolution information in Table 9 and added epsilon term in Section 3 in our updated manuscript.\
(4) Analysis with smaller number of frames: We have added model performance across 0, 1, 2, 4, 6, 8 frames in Figure 5 in our updated manuscript.

- Reviewer NXCv:\
(1) Table clarifications: We have added additional clarifications on Table 6 in our updated manuscript.\
(2) Comparison of FPV and TPV requires a more rigorous study: We clarified that theses study intend to suggest general trends in model performance for future research and have carefully rephrased our findings (Appendix B Scenario Analysis). \
(3) References for the principles: We have included references for each principle in our updated manuscript (Section 3).

- Reviewer xcjw: \
(1) 16 frames as the multiple-frame input seems arbitrary: We have studied 1, 8, 16, 24, 32 frames as the multi-frame input, and experiments have shown that as performance stabilizes around 8 frames, 16 frames provide sufficient window for analysis (Figure 4 in Appendix E.3). \
(2)  Insufficient Text-Only Baseline in Table 5: We have clarified our result in pilot study ([openreview response link](https://openreview.net/forum?id=fCi4o83Mfs&noteId=q51rylzyHo)) and have added additional analysis in Section 6.3, revealing that models **lack the basic ability to interpret frames as a continuous sequence**. \
(3) Fairness in benchmark comparison: We have clarified that the selection of 200 QAs for each benchmark is done through proportional random sampling across all reasoning types (if applicable).

---

### Author Response · Authors · 2024-12-04
**General Response - Part 1/2**

We express our sincere gratitude to the reviewers for their insightful comments and constructive feedback.

We appreciate the reviewer’s recognition of key aspects of our work:
- **The proposed benchmark addresses important shortcomings in existing benchmarks**: \
“It addresses an important area in video understanding research” (reviewer rS5B). \
“The paper tackles an important aspect of video understanding, which with benchmarks that are not carefully designed, might be overestimated.” (reviewer tp5Z).
- **The proposed benchmark is rigorously constructed with three quantitative principles**: \
“The benchmark is manually curated in a three-stage process, potentially yielding very high-quality QA annotations” (reviewer rS5B). \
“The dataset seems to consist of well designed and curated data … This is likely due to the rigorous quality check … ” (reviewer tp5Z). \
“I think the approach to the problem and processes are rigorous and effective” (reviewer NXCv). \
“The benchmark is rigorously constructed with three quantitative principles that emphasize the complexity of temporal reasoning tasks” (reviewer xcjw).
- **The experiments and analysis are comprehensive and insightful**:\
“Some of the ablations / analysis about existing models' capabilities are interesting” (reviewer rS5B). \
“Intensive experiments and the analyses provide valuable insights into the performance of multimodal foundation models on video data.” (reviewer tp5z).\
“The benchmark includes a wide range of tasks and scenarios, providing thorough coverage of diverse temporal reasoning challenges” (reviewer xcjw).
- **The paper is well-written and easy to follow**:\
	“The paper is written well and easy to follow.” (reviewer rS5B). \
	“The paper is easy to follow and the key concepts are delivered clearly.” (reviewer NXCv). \
	“Clear and Cohesive Writing: The paper is well-written, presenting its ideas and findings in a clear and organized manner, making it accessible and informative for a broad audience.” (reviewer xcjw).

---

### Meta-Review · Area_Chair_j89L · 2024-12-17

**Metareview:**

This paper proposed a new benchmark to evaluate the visual temporal reasoning of existing models in video understanding task. It targets to resolve limitations in current video understanding benchmarks, which most questions can be answered without visual temporal reasoning. It proposed three quantitative principles (e.g., Multi-Frame Gain, Frame Order Sensitivity, and Frame Information Disparity.) and rigorously construct corresponding metrics based on them. A new benchmark with carefully designed questions are proposed. Several existing MFMs are benchmarked and results show big gap between these MFMs and human performance, exhibiting limitations in these MFMs.

Strength:
1. It addresses an important problem.
2. The metrics are rigorously designed.
3. Experiments are extensive and comprehensive.
4. Paper is well written.

Weakness:
1. Based on the discussion between the reviewer rS5B, the proposed principles serve as diagnostic tools which reflect some aspects of visual temporal reasoning but may not be comprehensive. The authors may want to make this point clear in their paper.

**Additional Comments On Reviewer Discussion:**

Reviewer rS5B pointed out that:
1. There are some issues in the proposed three principals: why only consider to compare with single frame, whether order sensitivity is a reasonable benchmark; why only a single frame is considered in frame contribution.
2. Using GPT-4o to evaluate the three principles creates a chicken and egg problem.
The authors admitted the issues proposed by the reviewers and argued that the proposed benchmark serves as diagnostic tools instead of comprehensive metrics. I agree with the reviewer's comments and the authors response and suggest that the author make this clear in their paper. The proposed benchmark is not comprehensive but it could help reveal some loopholes in existing evaluation metric so it is still a qualified paper.

Reviewers tp5Z and NXCv required some clarifications of the paper and their concerns are addressed during the discussion.

Reviewer xcjw pointed out some issues about the benchmark: 1. Data imbalance across tasks; 2. Fairness in benchmark comparisons; 3. Insufficient text-only baselines. The authors mentioned the difficulty of collecting more questions for some tasks and emphasis the quality of the selected questions to address point 1. To address point 2, the authors mentioned that the 200 questions are sampled through proportional random sampling, which reflect the distribution well. The authors addressed point 3 by giving more details about how the experiments are done.
Reviewer xcjw still has some concerns about point 2 but also mentioned overall, this paper is above acceptance threshold.

---

### Decision · Program_Chairs · 2025-01-22

Accept (Poster)